# Interpretable Transformer Regression for Functional and Longitudinal Covariates

## Abstract

Predicting scalar outcomes from functional data is challenging when measurements are sparse, irregular, and noisy, as in many scientific and clinical longitudinal studies. We propose IDAT, a dual-attention Transformer that operates directly on masked sampling schedules and avoids ad-hoc imputation. IDAT couples (i) time-point attention, which captures local and long-range temporal dynamics together with the response relationship nonparametrically, with (ii) inter-sample attention, which adaptively shares information across subjects with similar trajectories to stabilize estimation under sparsity. These pathways complement one another: time-point attention captures subject-specific dynamics, whereas inter-sample attention leverages population structure to "borrow information" from other subjects, echoing principles from random-effects model in longitudinal analysis. Under a random-effects framework that accounts for irregular sampling and measurement noise, we prove prediction-error bounds and show that IDAT consistently approaches the oracle solution. Across both simulations and real-world applications, IDAT achieves the best overall performance among 19 baselines. Only in the extremely dense case ($> 80\%$ observations) TabPFN (a recent method published in Nature) achieve a slight advantage, while IDAT still significantly outperforms all other baselines in this scenario. The learned attention weights are interpretable, revealing predictive time domains and potential clusters. In conclusion, IDAT, an end-to-end sparsity-aware Transformer architecture, offers improvements both in predictive accuracy and interpretability for scalar-on-function prediction.

## 1 Introduction

Longitudinal data consist of repeated measurements on the same subjects over time, often coupled with time-varying covariates and subject-specific heterogeneity (Diggle et al., 2002; Fitzmaurice et al., 2004; 2009). Such data are ubiquitous in biomedicine, environmental monitoring, and digital health, where dynamic trajectories inform prognosis, treatment response, and risk stratification. It is common to assume that the underneath dynamic trajectory of each subject is a smooth function of time, while the observed longitudinal measurements may be noisy and measured at irregular and subject-specific time points. This adopts a perspective in the field of functional data analysis (FDA) (Wang et al., 2016; Ramsay & Silverman, 2005), where each trajectory is a realization of a latent smooth process observed with noise. When the sampling plan is intensive, a nonparametric approach is typically deployed to model such functional data; whereas a parametric approach, such as a mixed-effect model (Laird & Ware, 1982b) was the norm to model sparse or irregularly sampled functional data, until a nonparametric approach was proposed in (Yao et al., 2005a). Since longitudinal data are discretely sampled functional data (possibly with noise/measurement error), we aim to develop a unified nonparametric approach that handles a broad range of sampling schemes, whether they are intensive, sparse, or in between, including irregular and subject-specific time schedules.

We address the scalar-on-function regression problem, where a functional covariate, possibly observed on a sparse and irregular time schedule with noise, is used to predict a scalar outcome. The method must learn an unspecified functional of the whole trajectories without restrictive parametric forms. Since accurate scalar prediction hinges on correctly capturing the trajectory–outcome relationship in the presence of sparsity and noise, an effective model should therefore (i) accommodate irregular sampling and sparse data, (ii) encode temporal order and dependencies, (iii) borrow

strength across similar subjects, and (iv) learn the trajectory–outcome relationship without restrictive parametric assumptions.

Given that sparse and irregular longitudinal data are the most challenging type of functional data, we introduce the **I**nterpretable **D**ual-**A**ttention **T**ransformer (IDAT), a unified architecture inspired by Transformers (Vaswani et al., 2017) that is tailored to handle high sparsity and irregular sampling. Meanwhile, this approach is broadly applicable to any type of functional/longitudinal data. In contrast to "sparse Transformers" that impose artificially designed sparse attention patterns for computational efficiency (Jaszczur et al., 2021; Correia et al., 2019), our notion of "sparsity" refers to sparsity arising from the sampling scheme. We discretize the time interval into a working grid and use explicit sampling masks to encode each subject's observation pattern. The model combines (i) time-point attention, which serves as a data-adaptive functional encoder capturing smooth trajectory structure, with (ii) inter-sample attention, which learns nearest-neighbor–like weights across subjects via a learned similarity metric. A regressor layer then summarizes the learned representation to deliver end-to-end scalar-on-function prediction.

## 1.1 MAIN CONTRIBUTIONS

**End-to-end scalar-on-function regressors.** We propose a dual-attention Transformer that predicts a scalar outcome from a sparse, irregular longitudinal trajectory in an end-to-end manner. The model uses explicit sampling masks under supervised setup in the training stage, avoiding ad-hoc imputation while respecting each subject's observation pattern. The architecture jointly captures within-trajectory structure and cross-subject similarity, yielding a trainable pipeline without restrictive parametric assumptions. Cross-subject information sharing is essential for longitudinal data due to data sparsity, but is absent in existing longitudinal Transformer designs.

**Interpretable dual-attention Transformer.** The learned attention weights act as nonparametric regression coefficients along two axes. Time-point attention serves as a data-adaptive smoother, aggregating local and long-range temporal information to encode functional structure while propagating signal about $Y$. Inter-sample attention implements learned similarity over subjects, akin to nearest neighbors with a learned metric, stabilizing predictions for sparse and noisy data. Together, the embedding functions as an informative weighting mechanism rather than simple imputation, capable of revealing domain-relevant temporal windows and cluster structure.

**Theoretical and numerical justification.** We derive prediction error bounds and show consistency under sparse, irregular sampling, where standard Transformer theory (which assumes fully observed or densely sampled inputs) does not apply. Extensive simulation and real data studies across a wide range of sparsity levels demonstrate robustness to diverse sampling schemes and superior accuracy relative to 19 baselines (ensemble methods, statistical/functional models, deep learning methods, tabular Transformers, and pre-trained models like TabPFN).

## 1.2 RELATED WORK

Modeling longitudinal data, defined as repeated measurements over time that are often irregularly sampled and of varying length, poses challenges for representation learning and prediction. In biomedical settings (e.g., EHR), recent surveys document a rapid expansion of machine learning and deep learning approaches (Cascarano et al., 2023; Carrasco-Ribelles et al., 2023). Early neural models typically flatten a temporal history into fixed feature vectors for feedforward networks (e.g., cardiovascular risk prediction (Zhao et al., 2019)), thereby discarding ordering information. To retain functional structure, Yao et al. (2021) propose a basis-learning layer in which hidden units act as adaptive basis functions, enabling end-to-end, task-specific basis for fully observed functional data.

Convolutional neural networks (CNNs) capture local temporal structure via 1D convolutions and are competitive for time-series classification (Wang et al., 2017), but modeling long-range dependencies often requires very deep networks or large receptive fields. Recurrent architectures (RNNs/LSTMs) maintain hidden states that aggregate past information and naturally handle variable sequence lengths and missingness patterns, with applications in clinical prognosis (e.g., Alzheimer's disease (Cui et al., 2019; Aghili et al., 2018)) and broader EHR modeling (Lipton et al., 2016). Nonetheless, their inherently sequential computation can be a bottleneck for long sequences.

Transformers (Vaswani et al., 2017) replace the sequential recurrence approach in CNN, RNN, and LSTM by "self-attention" to relate all time points within a sequence, capturing both local and global dependencies and supporting parallel computation. Empirically, reviews report strong performance on longitudinal biomedical tasks (Siebra et al., 2024). Early EHR applications such as BEHRT encoded patient histories as sequences of medical concepts to learn contextualized representations for downstream prediction (Li et al., 2020), while general frameworks demonstrated effectiveness across multivariate time-series classification and regression (Zerveas et al., 2021). The architecture has also been adapted to domain-specific objectives, including survival modeling (Öğretir et al., 2024; Zhang et al., 2025). Beyond prediction, specialized self-attention modules have been proposed for functional data imputation: SAND (Hong et al., 2024) introduces attention weights on derivatives to promote smooth reconstructions under irregular sampling. More broadly, efficient attention variants (e.g., sparse or kernelized forms (Jaszczur et al., 2021; Lou et al., 2024; Correia et al., 2019; Chen et al., 2023)) have been explored to mitigate quadratic time/memory costs on long sequences, though their approximation properties in sparse/irregular regimes require careful validation.

## 2 INTERPRETABLE DUAL-ATTENTION TRANSFORMER (IDAT)

The proposed method, IDAT, is a dual-attention (encoder-only) Transformer for scalar-on-function regression. IDAT (Figure 2) is designed to handle sparse, irregular longitudinal inputs and is applicable to all longitudinal and functional data. Since latent trajectories of functional data are infinite-dimensional, discretization is required for transforming the time domain into grid points. Paired with positional encodings which captures temporal information between the grids to preserve temporal ordering and proximity, we also apply sampling masks to respect the sparsity of the sampling schedule. This produces a tabular-style input embedding, whose columns index time grid locations, so each column at a time grid acts as a feature. The key difference from standard tabular data is the heavy masking induced by sparse and irregular sampling and adjacent columns carry strong temporal dependence through the underlying smooth trajectory.

The dual-attention Transformer block (omit normalization layers) is defined as

$$\mathcal{TB} = \text{FF}_1 \circ A_I \circ \text{FF}_2 \circ A_T. \tag{1}$$

The *time–point attention* $A_T$ operates along the temporal axis within each sample. The *inter-sample attention* $A_I$ applies attention across samples in the batch. The position-wise feed-forward layers $\text{FF}_1, \text{FF}_2$ are two-layer ReLU MLPs. Together, the two attentions reinforce each other: time-point attention captures subject-specific dynamics, while inter-sample attention leverages population structure so trajectories borrow strength from similar subjects when per-subject data are scarce. Stacking $L$ dual-attention Transformer blocks yields the embedding $\mathcal{T} = \mathcal{TB}^{\circ L}$.

$A_T$ operates along each subject's time grid to learn both local and long-range temporal dependencies, encoding trajectory structure while respecting order and smoothness. Unlike imputation methods that reconstruct the entire trajectory without using the outcome, our model is trained with a $Y$-token. During training, attention flows between covariate tokens and the response, providing supervision that turns $A_T$ into a nonparametric weighting scheme over time (low weights mark uninformative windows). At test time the $Y$-token is masked, so predictions rely solely on observed covariates but still benefit from the supervision-shaped representation learned in training. $A_I$ acts across the mini-batch at each time grid, assigning data-adaptive weights to subjects with similar patterns, thereby sharing information across similar subjects and often revealing cluster structure. This modular design couples time-point and inter-sample attentions, allows us to benefit from both within-trajectory encoding and across-subjects contributions. The model prioritizes prediction over imputation, learning attention weights end-to-end for regression rather than pure reconstruction.

Longitudinal data remain fundamentally different from tabular data, even after they have been discretized, due to the within-subject temporal correlations. Existing Transformers for longitudinal data use only temporal attention and ignore cross-subject information sharing. This results in efficiency loss, and the loss could be substantial when each subject has few observations. IDAT leverages the inter-subject attention to improve the performance of transformers for longitudinal data. In addition, longitudinal data are noisy and irregular measurements from a latent process that is often assumed to be smooth. This smoothness assumption is needed for identifiability (Hall et al., 2006; Wang et al., 2016; Li & Hsing, 2010) but often ignored by standard Transformer analyses, which assume dense or fully observed inputs without noise, aka measurement errors. This not only distinguished the

approach of IDAT from standard Transformers' but also has the implication that the theory for longitudinal data is substantially different from existing theory for tabular data. The following sections introduce the setup and key theoretical results to highlight these theoretical differences and establish consistency and generalization guarantees for IDAT in this challenging regime.

## 3 THEORETICAL SETUP

Without loss of generality, we assume all subjects have trajectories in the time interval $I = [0, 1]$ with latent smooth trajectories $X_i(\cdot)$, under contamination of measurement errors and irregular sampling scheme, the observation times for subject $i$ is $\tilde{t}_i = (t_{i1}, \ldots, t_{i,n_i}) \subset I$, and we observe

$$X_i^*(\tilde{t}_i) = X_i(\tilde{t}_i) + \eta_i(\tilde{t}_i), \qquad \eta_i(\tilde{t}_i) \overset{iid}{\sim} N(0, \sigma_X^2). \tag{2}$$

The scalar response is generated from the functional regression model with an unspecified $\mathcal{F}$,

$$Y_i = \mathcal{F}(X_i(\cdot)) + \epsilon_i, \qquad \epsilon_i \overset{iid}{\sim} N(0, \sigma_Y^2). \tag{3}$$

We refer to $\epsilon_i$ as the label noise on $Y_i$, in contrast to the measurement noise $\eta_i$ on $X_i$.

**Discretization, masking and positional encoding.** Despite irregular measurement times and varying $n_i$ across subjects, we align all observations to a shared fixed grid $\tilde{\tau} = (\tau_1, \ldots, \tau_T) \subset I$,

$$M_i(\tau_j) = \mathbf{1}\{\tau_j \in \tilde{t}_i\}, \qquad X_i^*(\tau_j) = X_i^*(t_{i,k})\mathbf{1}\{\tau_j = t_{i,k}\}, \tag{4}$$

and form the length-$T + 1$ vector $D_i = (X_i^*(\tilde{\tau}) \odot M_i(\tilde{\tau}), Y_i) \in \mathbb{R}^{T+1}$ with a mesh size defined by $\Delta = \max_j |\tau_{j+1} - \tau_j|$. Linear embeddings $E_X$ and $E_Y$ from $\mathbb{R}$ to $\mathbb{R}^d$ are applied token–wise along with sinusoidal positional encodings $P(\tilde{\tau})$,

$$\tilde{D}_i = \left[ E_X(X_i^*(\tilde{\tau}) \odot M_i(\tilde{\tau})) + P(\tilde{\tau}), \ E_Y(Y_i) \right] \in \mathbb{R}^{d \times (T+1)}. \tag{5}$$

**Dual–attention Transformer.** Given a batch $\tilde{\mathbf{D}}$ of size $B$, the dual-attention Transformer block is defined in (1). Stacking $L$ dual-attention Transformer blocks yields the embedding $\mathcal{T} = \mathcal{TB}^{\circ L}$.

**Regression layer.** Given the output of dual–attention Transformer $\mathcal{T}(\tilde{D}_i) \in \mathbb{R}^{d \times (T+1)}$, which contains the longitudinal covariate embeddings and the $Y$-token embedding. Each sequence is summarized to a single $d$-vector via a trainable pooling map $\phi : \mathbb{R}^{d \times T} \to \mathbb{R}^d$. The pooled representation is then mapped to a scalar by a MLP $g$ (e.g., two-layer ReLU), yielding the prediction

$$\hat{Y}_i = g\left(\phi([\mathcal{T}(\tilde{D}_i)_{:,(1:T)}])\right). \tag{6}$$

During training, we minimize the loss function $\ell(\hat{Y}_i, g([Z_Y]_i))$ (e.g., MSE for regression), treating the response embedding as an informative target. At test time, the $Y$ token is masked (set to zero) prior to encoding, so predictions depend solely on the covariate sequence.

## 4 KEY THEORETICAL RESULTS

IDAT is a unified approach that works for densely or sparsely recorded functional data, whether the measurement schedule is regular or irregular, and whether the data exhibits clustering or heterogeneity, as long as training and testing data follow the same sampling mechanism and functional relationship. For theoretical analysis, we assume the following setup. Let $\tilde{\mathbf{D}}$ denote the discretized, noisy, masked embedding, and $\tilde{\mathbf{S}}$ the oracle embedding (without measurement and label errors).

**Mesh–size trade-off.** For an $\alpha$-Hölder trajectory, Lemma 3 bounds the error between the embedded observed data $\tilde{\mathbf{D}}$ to the oracle with high probability:

$$\delta_0 = \|\tilde{\mathbf{D}} - \tilde{\mathbf{S}}\|_\infty \lesssim L\Delta^\alpha + \sigma_X \sqrt{\log(BT)} + \sigma_Y \sqrt{\log(B)},$$

where $\Delta = \max_j |\tau_{j+1} - \tau_j|$ is the mesh size. The discretization bias scales as $O(\Delta^\alpha)$: as the grid refines ($\Delta \to 0$), the bias vanishes at rate $\Delta^\alpha$, while noise terms remain controlled. Since $T \asymp \Delta^{-1}$,

time-point attention incurs $O(T^2d)$ time and $O(T^2)$ memory per subject, so halving $\Delta$ quadruples computational cost while improving the bias by only $2^{-\alpha}$. The inter-sample component scales linearly in $T$, so a similar trade-off applies. In practice, $\Delta$ must balance statistical accuracy (smaller $\Delta$) against computational cost (larger $\Delta$). This result is central to establishing consistency: with appropriate smoothness assumptions and grid refinement, the discrete implementation converges to the continuous functional relationship.

In training, the dual-attention block components are Lipschitz: time-point self-attention $A_T$ with constant $L_T$, feed-forward networks $\mathrm{FF}_1, \mathrm{FF}_2$ with constant $L_{\mathrm{FF}}$, and inter-sample attention $A_I$ with constant $L_I$ (Lemmas 4, 8). They also admit uniform approximation on compact sets (Lemmas 5, 6, 7), yielding deterministic approximation errors $\varepsilon_{A_T}$, $\varepsilon_{\mathrm{FF}}$, $\varepsilon_{A_I}$. The dual-attention block $\mathcal{T}_B$ is $L_{\mathcal{T}_B}$-Lipschitz (Lemma 8), ensuring stability: small perturbations in input embeddings propagate in a controlled manner. Inter-sample attention reduces a stochastic embedding error $\varepsilon_{\mathrm{var}}$ by up to a $B^{-1/2}$ factor (Lemma 9) with batch size $B$, showing that larger batches improve variance reduction, though with diminishing returns. Together, these properties ensure that approximation errors accumulate in a controlled way and that the embedding remains stable under discretization and noise, which is essential for establishing consistency.

**Variance reduction by inter–sample attention.** When subject-level heterogeneity is not prominent, queries/keys align across subjects and the inter-sample attention weights are nearly uniform ($\approx 1/B$), yielding an $O_{\mathbb{P}}(1/\sqrt{B})$ reduction in embedding noise (Lemma 9). This variance contraction is orthogonal to deterministic approximation biases, so increasing $B$ stabilizes the embedding without changing those bias terms. MSE improves when the induced averaging does not introduce substantial pooling bias, namely, when the attended neighbors are genuinely similar for the target.

Together, Theorem 1 establishes consistency of $\mathcal{T}$: the IDAT embedding converges to the oracle embedding obtained from a noiseless trajectory observed fully on the time grid.

**Theorem 1** (Consistency of $\mathcal{T}$). *Under the assumptions and notations of Lemmas 3-9, the oracle mapping is $H(\tilde{\mathbf{S}}) = G \circ f_I \circ G \circ f_T(\tilde{\mathbf{S}})$. Let*

$$\varepsilon_{\mathcal{T}\mathcal{B}} := L_{\mathrm{FF}}\big(\varepsilon_{\mathrm{var}} + \varepsilon_{A_I} + L_I\big(\varepsilon_{\mathrm{FF}} + L_{\mathrm{FF}}\varepsilon_{A_T}\big)\big) + \varepsilon_{\mathrm{FF}}.$$

*Then, with probability at least $1 - \delta$,*

$$\big\|\mathcal{T}\mathcal{B}(\tilde{\mathbf{D}}) - H(\tilde{\mathbf{S}})\big\|_\infty \leq \varepsilon_{\mathcal{T}\mathcal{B}}, \qquad \big\|\mathcal{T}(\tilde{\mathbf{D}}) - H(\tilde{\mathbf{S}})\big\|_\infty \leq L_{\mathcal{T}\mathcal{B}}^L (\delta_0 + \varepsilon_{\mathcal{T}\mathcal{B}}) := \varepsilon_{\mathcal{T}}. \quad (7)$$

*In particular, if (i) the mesh shrinks $\Delta \to 0$ so that $\delta_0 \to 0$; (ii) the dual-attention Transformer has sufficient capacity so that $\varepsilon_{\mathrm{FF}}, \varepsilon_{A_T}, \varepsilon_{A_I}, \varepsilon_{\mathrm{var}} \to 0$; and (iii) the block Lipschitz constant is uniformly bounded with training size $n$ and batch size $B$, i.e., $\sup_{n,B} L_{\mathcal{T}\mathcal{B}}(n, B) < \infty$ for fixed depth $L$, then*

$$\big\|\mathcal{T}(\tilde{\mathbf{D}}) - H(\tilde{\mathbf{S}})\big\|_\infty \xrightarrow{\mathbb{P}} 0,$$

*i.e., the dual-attention Transformer $\mathcal{T}$ is consistent for the oracle mapping $H$.*

This result is essential for IDAT as it shows that the dual-attention mechanism can recover functional structure from discretized, sparse, noisy observations. Theorem 1 bounds the error between the IDAT embedding $\mathcal{T}(\tilde{\mathbf{D}})$ and the oracle mapping $H(\tilde{\mathbf{S}})$ by $L_{\mathcal{T}_B}^L(\delta_0 + \varepsilon_{\mathcal{T}_B})$, where $\delta_0$ captures input embedding discretization error and $\varepsilon_{\mathcal{T}_B}$ aggregates approximation errors from dual-attention components, including variance reduction from inter-sample attention. The bound implies consistency under three conditions: (1) mesh refinement ($\Delta \to 0$) to reduce discretization error, (2) sufficient model capacity to drive approximation errors to zero, and (3) a uniformly bounded Lipschitz constant as training size and batch size grow, ensuring stability. Practically, this means using finer time grids (balanced with computational cost), increasing model capacity, and leveraging larger batches for variance reduction (with diminishing $B^{-1/2}$ returns). Theoretically, this addresses identifiability: even with sparse, noisy observations, IDAT converges to the true functional relationship, showing that attention mechanisms can handle sparse functional data (not just dense tabular data) and that dual-attention is sufficient for consistency. This differs from standard Transformer theory, which assumes dense inputs, and demonstrates that with mesh refinement and sufficient capacity, IDAT resolves the identifiability issues inherent in sparse functional data.

Generalization bounds for $\mathcal{T}\mathcal{B}$ and $\mathcal{T}$ are given in Lemmas 10 and 11. Define the predictor with $L_\phi$-Lipschitz pooling function $\phi$ and $L_g$-Lipschitz MLP (excluding the $Y$-token),

$$\hat{Y}^{(L)}(\tilde{D}) = g\big(\phi(\mathcal{T}(\tilde{D})_{:,:,(1:T)})\big).$$

**Theorem 2** (Training MSE generalization and consistency). *Let $\mathcal{T} = \mathcal{T}\mathcal{B}^{\circ L}$ be the $L$-block encoder, and set $p := B\,d\,(T+1)$. Assume (i) Boundedness: $\|\tilde{D}\|_\infty \leq R_{\text{in}}$, $|\widehat{Y}^{(L)}(\tilde{D})| \leq R_{\text{out}}$, and $|Y| \leq M_f$ almost surely with $L_\ell := 2(R_{\text{out}} + M_f)$. (ii) Optimization: a stable SGD regime in which the empirical risk approaches its minimum (within the hypothesis class) with high probability (Hardt et al., 2016). (iii) $\sup_n L_{\mathcal{T}}(n) < \infty$ for fixed depth $L$ and (iv) $p/n \to 0$. Let $\text{MSE}_n^{\text{train}}$ be the training MSE over $n$ samples. Then, for any $\delta \in (0,1)$, with probability at least $1 - \delta$,*

$$\left| \text{MSE}_n^{\text{train}} - \mathbb{E}\left(\widehat{Y}^{(L)}(\tilde{D}) - Y\right)^2 \right| \leq 2\,L_\ell\,L_g\,L_\phi\,L_{\mathcal{T}}\,R_{\text{in}}\sqrt{\frac{2p}{n}} + 3\sqrt{\frac{\ln(2/\delta)}{2n}}.$$

*Moreover, if the regressor approximation error vanishes as capacity grows (Hornik, 1991; Stinchcombe, 1999; Cybenko, 1989; Hornik et al., 1989; Yarotsky, 2017) and $\mathcal{T}$ is consistent (Theorem 1), the population training MSE is consistent.*

Theorem 2 provides a generalization bound connecting training MSE to population error, where $p = Bd(T+1)$ is the effective dimension. The bound has two terms: a complexity term scaling as $\sqrt{p/n}$ (depending on Lipschitz constants and input bounds) and a concentration term scaling as $\sqrt{\ln(1/\delta)/n}$. Under standard conditions, training MSE converges to population MSE. Combined with Theorem 1, this establishes consistency of the population training MSE when approximation error vanishes. Practically, this means balancing model complexity ($p$) with sample size: larger batches, embedding dimensions, or time grids require more data. Theoretically, this provides finite-sample generalization guarantees, the training procedure generalizes well and establishes both consistency and generalization for IDAT.

During training, the final token carries the response embedding to learn the $X \to Y$ relation. By Lemma 12, masking the $Y$-token at test time induces an approximation perturbation scaled by $L_Y = \|E_Y\|_{\text{op}}$. To reduce train-test mismatch, one can randomly mask the $Y$-token during training: draw $d_i \sim \text{Bernoulli}(q)$ and feed $\tilde{Z}_Y = d_i E_Y Y_i$ with a mask-reweighted loss. The training-phase embedding error remains bounded, with the fully masked case serving as a worst-case upper bound.

**Masking $Y$-token.** Lemma 12 bounds the error between the test embedding $\mathcal{T}(\tilde{\mathbf{D}}^*)$ (with masked $Y$-token) and the oracle embedding $H(\tilde{\mathbf{S}})$ as $\|\mathcal{T}(\tilde{\mathbf{D}}^*) - H(\tilde{\mathbf{S}})\|_\infty \leq L_{\mathcal{T}} L_Y (M_f + \sigma_Y \sqrt{2\ln(2N/\delta)}) + \varepsilon_{\mathcal{T}}$, decomposing into (1) a train-test mismatch term scaling with $L_{\mathcal{T}} L_Y$ and response magnitude/noise, and (2) the training embedding error $\varepsilon_{\mathcal{T}}$ from Theorem 1. This shows that test-time predictions remain consistent when training is consistent, with the bound growing logarithmically with test sample size. It also provides a quantitative bound on the train-test gap (perturbation) controlled by the product of Lipschitz constants and expected response magnitude.

Theorem 13 further bounds the pointwise and uniform test errors and controls the population test MSE by the training MSE plus a Rademacher term and an expectation bridge; the empirical test MSE adds an extra concentration term. If the training MSE is consistent and the $Y$-token embedding is scaled so that $L_Y \to 0$, the bridge term vanishes then both population and empirical test MSE are consistent asymptotically (Corollary 14). Details of the theoretical part are listed in the Appendix.

## 5 EXPERIMENTS

We compare our method against a diverse set of 19 baselines that collectively cover statistical, functional, attention-based tabular, deep learning methods and ensemble approaches, each adapted to irregular and sparse longitudinal inputs. The statistical and/or functional baselines comprise ordinary linear regression (LR), functional linear regression (FLR) (Yao et al., 2005a; Cai & Hall, 2006), and functional principal components analysis followed by a regression neural layer (FPCA+NN) (Yao et al., 2005b; Wang et al., 2016).

For deep learning and tabular Transformer methods, we provide mean-imputed inputs on a fixed grid, where each feature corresponds to a time point. Compared methods include SAINT (Somepalli et al., 2021), FTTransformer (Gorishniy et al., 2021), TabNet (Arik & Pfister, 2021), AutoInt (Song et al., 2019), and a vanilla Transformer trained solely on covariates without a $Y$-token followed by a regression neural layer (VT+NN). We also include the most recent state-of-the-art tabular model published in Nature, TabPFN (Hollmann et al., 2022; 2025), a generative Transformer-based foundation model pretrained on millions of synthetic datasets. To assess the value of end-to-end training

relative to decoupled imputation, we also evaluate SAND (Hong et al., 2024) augmented with a prediction multilayer perceptron (SAND+NN)[1], thereby testing how well a learned imputer performs when the regression layer is trained separately. Other deep learning approaches including multilayer perceptron (MLP) and ResNet (He et al., 2016) are considered. As well as AdaFNN (Yao et al., 2021), a basis-specified neural method tailored to completely observed functional data.

Finally, we benchmark strong ensemble and tree-based systems, including AutoGluon (Erickson et al., 2020), which automatically trains, tunes, and stacks diverse models on tabular tasks. Gradient-boosted approaches, including XGBoost (Chen & Guestrin, 2016), LightGBM (Ke et al., 2017), Cat-Boost (Prokhorenkova et al., 2018), and NODE (Popov et al., 2019) are included. We also evaluate xRFM (Beaglehole et al., 2025), a very recent method that combines random-feature (kernel-style) learning with tree-based partitioning. These 19 existing models are compared with our proposed dual-attention method (IDAT) and a variant without inter-sample attention (IDAT w/o $A_I$).

## 5.1 SIMULATION

Without loss of generality, we consider functions on the unit interval $\mathcal{I} = [0, 1]$ generated as follows. In all simulations, the time grid has length $T = 100$. For subject $i$ in group $g \in \{1, \ldots, G\}$, the (noise-free) latent trajectory is

$$X_i(t) = \mu_g(t) + \sum_{k=1}^{20} \left[ a_{i,k}^s \sin(2\pi kt) + a_{i,k}^c \cos(2\pi kt) \right] / k, \qquad t \in \mathcal{I},$$

where $\mu_g$ is a group-specific mean function and $\{a_{i,k}^s, a_{i,k}^c\}$ are Fourier coefficients with smoothness controlled by the decay rate of $k^{-1}$. To induce smooth but heavy-tailed trajectories (beyond the sub-Gaussian assumption), these coefficients are independently drawn from a zero-mean exponential distribution. The response is generated by a functional operator $Y_i = \mathcal{F}(X_i) + \varepsilon_i$ with $\varepsilon_i \overset{iid}{\sim} N(0, 1)$.

**Case I: Functional linear regression.** Set $\mu_g \equiv 0$. Let $\mathcal{F}(X) = \int_0^1 \beta_1(t) X(t) \, dt$ with

$$\beta_1(t) = (3 - 6t) \mathbf{1}\{t \leq 0.5\} + (2t - 1) \mathbf{1}\{t > 0.5\}.$$

**Case II: Nonlinear model.** Set $\mu_g \equiv 0$. Define

$$\mathcal{F}(X) = \int_0^1 \beta_2(t) X(t) \, dt + \left( \int_0^1 \beta_3(t) X(t) \, dt \right)^2,$$

with $\beta_2(t) = (4 - 16t) \mathbf{1}\{t \leq 0.25\}$ and $\beta_3(t) = (4 - 16|t - 0.5|) \mathbf{1}\{0.25 \leq t \leq 0.75\}$. Here, time points $t > 0.75$ are non-informative; the interval just beyond $t = 0.25$ contributes weakly.

**Case III: Cluster analysis.** Let $G = 2$ and with $\mathcal{F}(X) = \int_0^1 0.5 \, t \, X(t) \, dt$,

$$\mu_1(t) = 1 + 4(t - 0.5)\mathbf{1}\{t \geq 0.5\}, \qquad \mu_2(t) = -6(t - 0.5)\mathbf{1}\{t \geq 0.5\}.$$

This scenario is explicitly designed to demonstrate the clustering ability, where group-specific mean shifts create separable subpopulations that inter-sample attention can cluster.

In addition to Cases I-III, we also add measurement error on the functional covariate[2]. We set the signal-to-noise ratio to SNR $= \left( \int_0^1 X_i(t)^2 \, dt \right) / \text{Var}(\eta_i) = 2$. Across the six simulation cases, we evaluate five observation regimes at various levels of sparsity, where the observed percentage of data is provided (in the parentheses): *ssparse* (10%), *vsparse* (20%), *sparse* (50%), *dense* (80%), and *full* (100%). The mean squared errors (MSE) of test samples for all 30 settings are reported in Tables 4–9, evaluating predictive accuracy and computational efficiency across the full spectrum of sampling scenarios. As Table 1 shows, IDAT is the jl:[ What does consistent mean? ] cj:[best-performing] overall (Top1 11/30, Top3 26/30), especially in non-dense regimes ($\leq 50\%$ of the data are observed). TabPFN emerges as the main competitor as sparsity decreases, its pre-trained prior

---

[1]SAND is an imputation method and is not applied when the covariates are 100% observed.

[2]Cases with measurement errors are denoted by Case I*, II* and III*

favors simple, additive, and near-linear relationships, allowing it to fit dominant trends in dense settings. Intuitively, inter–sample attention lowers the variance by borrowing strength from "similar" subjects. However, in the presence of measurement errors, the similarity can be inaccurately estimated and borrowing from mismatched neighbors raises the bias. When the increase in bias outweighs variance reduction, the time–point–only variant (no $A_I$) can prevail, though the dual-attention model still outperforms other baselines. Empirically, IDAT delivers the largest gains in sparse settings with clear cluster structure (Table 3). In practice, we recommend the dual-attention model by default to detect/exploit clustering, with the $A_T$-only variant as a pragmatic fallback if the learned attention maps appear diffuse or uniform across subjects.

Table 1: Overall comparison on the simulation study across 6 cases and 5 sparsity levels. Top1/Top3 count how often a method ranks 1st or within the top 3 in terms of smallest test MSE. Efficiency is measured as average inference time per sample (ms) and model size (#Params in thousands). Preprocessing (imputation) costs are excluded from inference time.

| | Sampling density (# cases) | | | | | | Efficiency measures | |
| | all (30) | | $\leq 50\%$ (18) | | $> 50\%$ (12) | | inference time | #Params |
| Method | Top1 | Top3 | Top1 | Top3 | Top1 | Top3 | (ms/sample) | (1K) |
|---|---|---|---|---|---|---|---|---|
| LR | 3 | 7 | 0 | 1 | 3 | 6 | 0.005 | 0.1 |
| FLR | 0 | 0 | 0 | 0 | 0 | 0 | 19.71 | 0.5 |
| FPCA+NN | 0 | 0 | 0 | 0 | 0 | 0 | 24.433 | 13 |
| TabNet | 0 | 0 | 0 | 0 | 0 | 0 | 0.058 | 12 |
| SAINT | 0 | 0 | 0 | 0 | 0 | 0 | 17.870 | 125000 |
| FTTransformer | 0 | 0 | 0 | 0 | 0 | 0 | 8.501 | 123 |
| AutoInt | 0 | 0 | 0 | 0 | 0 | 0 | 22.179 | 619 |
| TabPFN | 9 | 25 | 4 | 13 | 5 | 12 | 1713.843 | 11000 |
| VT+NN | 0 | 0 | 0 | 0 | 0 | 0 | 28.093 | 333 |
| SAND+NN | 0 | 0 | 0 | 0 | 0 | 0 | 28.030 | 333 |
| MLP | 0 | 0 | 0 | 0 | 0 | 0 | 7.485 | 549 |
| ResNet | 0 | 0 | 0 | 0 | 0 | 0 | 6.597 | 568 |
| AdaFNN | 1 | 7 | 0 | 1 | 1 | 6 | 1.481 | 603 |
| NODE | 0 | 3 | 0 | 2 | 0 | 1 | 14.733 | 6800 |
| CatBoost | 1 | 2 | 0 | 0 | 1 | 1 | 0.067 | 190 |
| XGBoost | 0 | 2 | 0 | 1 | 0 | 1 | 0.014 | 2 |
| LightGBM | 0 | 0 | 0 | 0 | 0 | 0 | 0.021 | 9 |
| AutoGluon | 2 | 10 | 1 | 3 | 1 | 7 | 1.734 | $\approx 2000$ |
| xRFM | 0 | 4 | 0 | 4 | 0 | 0 | 0.070 | 180 |
| IDAT | 11 | 26 | 9 | 17 | 2 | 9 | 3.085 | 180 |
| IDAT w/o $A_I$ | 11 | 24 | 7 | 15 | 4 | 8 | 1.655 | 144 |

## 5.2 Real Data

**National Child Development Study:** We analyze data from the 1958 National Child Development Study (NCDS)[3]. The task is to predict BMI at age 62 from prior BMI trajectories observed at ages 11, 16, 23, 33, 42, 44, 46, 50 and 55. The cohort is relatively homogeneous, including individuals born in Great Britain during a single week in 1958, reducing potential confounding by ethnicity. All models are adjusted for baseline covariates measured at age 7: sex, baseline BMI, and an early-life social adversity index computed as the average of 13 binary indicators (e.g., housing problems, financial hardship, parental divorce, unemployment, illness, disability, or death) (Flèche et al., 2021). We fit sex-stratified models and compare the mean squared error (MSE) and mean absolute error (MAE) of two imputation pipelines for BMI trajectories: (i) mean imputation (a simple but commonly used approach in tabular workflows) and (ii) multiple imputation by chained equations (MICE) (van Buuren & Groothuis-Oudshoorn, 2011). The $n = 4952$ longitudinal BMI

---

[3]University College London, UCL Social Research Institute, Centre for Longitudinal Studies (2024)

series exhibit substantial missingness (mean 25%, range 8–96%), spanning dense to super-sparse regimes. On average, subjects have 6.2 observations (SD 0.9) across the sweeps (Figure 7). Across both imputation settings, IDAT consistently outperforms all competing models, while requiring no imputation or preprocessing and operating directly on sparse and irregular inputs.

**Synthetic HIV Dataset (Health Gym Project):** We use the HIV dataset from the Health Gym project (Kuo et al., 2022), a public collection of synthetic yet realistic clinical datasets, and focus exclusively on the HIV cohort. Since the measurements are monthly and equally spaced, alongside mean imputation and MICE, we also evaluate the last-observation-carried-forward (LOCF) method (Lachin, 2016; Woolley et al., 2009) and report accuracy and F1 score in Table 12. The binary response label indicates whether a patient achieves viral suppression (VL$< 200$ copies/mL) at any time during the prediction window (months 20–30). For each patient, we use VL measurements from the first 20 months as longitudinal covariates and include sex as a baseline covariate. The $n = 8683$ VL series in the feature window is very sparse (mean missingness 65%, range 47–88%). Subjects receive a mean of 5.8 observations (SD 1.8) during the 20-month covariate interval (Figure 9).

Table 2: Performance on real-world tasks. For NCDS BMI (regression), we report MSE/MAE under mean and MICE imputation. For Synthetic HIV (classification), we report F1 under mean, MICE, and LOCF imputation. The best method is in **bold** and the top three are in *italics*. Only a subset of the 19 baselines is shown here for readability; see Tables 11 and 12 for the full comparison.

| Method | NCDS BMI (MSE/MAE) | | | | Synthetic HIV (F1) | | |
|---|---|---|---|---|---|---|---|
| | mean imputed | | MICE | | mean | MICE | LOCF |
| SAINT | 9.7213 | 2.3524 | 9.4844 | 2.3231 | *0.9751* | *0.9745* | 0.9752 |
| TabPFN | *8.5794* | **2.1621** | 8.5164 | **2.1503** | 0.9722 | 0.9734 | *0.9758* |
| MLP | 9.4439 | 2.3142 | 9.2378 | 2.3191 | *0.9751* | 0.9727 | *0.9764* |
| NODE | 9.6511 | 2.2955 | 8.6911 | 2.2528 | 0.9727 | *0.9751* | **0.9769** |
| AutoGluon | 9.0408 | 2.2623 | 8.9595 | 2.2543 | *0.9746* | *0.9745* | 0.9751 |
| IDAT | **8.0061** | *2.1728* | **8.0061** | *2.1728* | **0.9752** | **0.9752** | 0.9752 |
| IDAT w/o $A_I$ | *8.1177* | *2.1966* | *8.1177* | *2.1966* | **0.9752** | **0.9752** | 0.9752 |

## 6 CONCLUSION

Across datasets and simulations with diverse sparsity, our end-to-end dual-attention Transformer IDAT, consistently performs the best when less or equal to $50\%$ of time points are observed, a regime typical of many longitudinal applications, and remains competitive as sparsity decreases. In practice, sparsity varies widely across cohorts, time windows, and variables; thus, robustness across sampling densities is essential. By adaptively leveraging inter-sample attention to borrow strength when data are scarce and emphasizing time-point structure as coverage improves, IDAT offers a unified solution across the full sparsity spectrum, while yielding interpretable dual-attention patterns that clarify when and where each mechanism contributes.

**Domain detection with time–point attention.** Time-point attention learns a data-driven weighting scheme over time, highlighting the segments of a trajectory that are most predictive. As shown in Figures 1a and 8b, when portions of the time axis are not informative for the response, the learned weights shrink toward (near) zero in those intervals, effectively performing time domain selection. This behavior mirrors the known informative window in the simulation and is corroborated by external domain knowledge in the real cohort as shown in Figure 8.

**Clustering with inter–sample attention.** Inter–sample attention acts as a learned, end-to-end nearest-neighbor mechanism: it computes attention across data points (rows) within a batch using a learned similarity and aggregates information from the most relevant samples. This cross-sample sharing of information is particularly helpful for sparse or noisy features. In clustered data, it produces cluster-specific attention profiles that better align with the regression signal. Figure 1b illustrates a two-group setting with different group mean functions: the clusters display clearly distinct

attention patterns[4]. This demonstrates the clustering capability of the dual-attention mechanism. In contrast, removing inter-sample attention (Figure 6) makes profiles of the two clusters much more similar, differing only slightly in the early trajectory. This highlights that time-point and inter-sample attention provide complementary and additive gains in predictive performance.

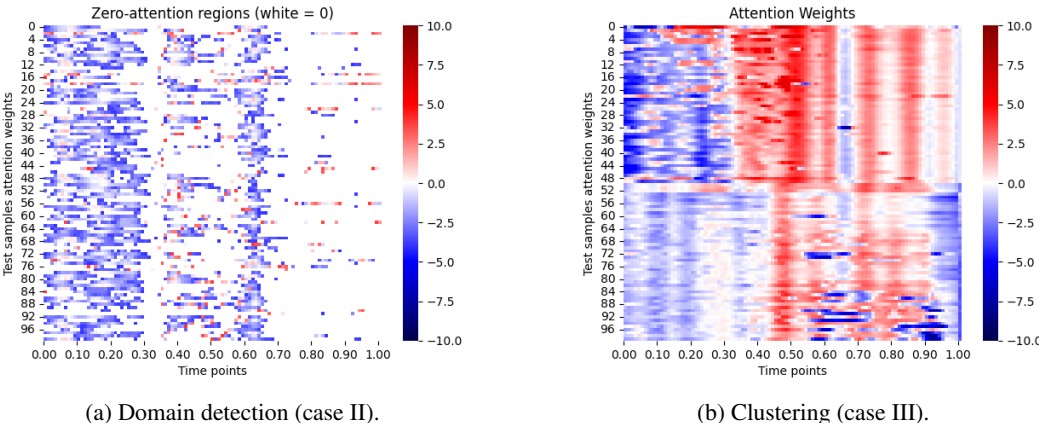

(a) Domain detection (case II).          (b) Clustering (case III).

Figure 1: Interpretable dual-attention. All the attention weights are are scaled by a factor of 1000.

DISCUSSION

In IDAT, temporal patterns benefit from similar subjects identified by inter-sample attention. At the same time, attention better captures subject similarity by leveraging temporal relationships learned in time-point attention. This mutual reinforcement is especially important when per-subject data are scarce. Unlike hierarchical or two-stage architectures, where temporal and relational modeling are decoupled and processed sequentially or in separate stages, IDAT differs from existing attention architectures by jointly modeling temporal and inter-sample dependencies in a single encoder. These hierarchical or two-stage architectures are problematic in sparse and irregular settings, because sparse data makes decoupled modeling unstable. Moreover, IDAT extends beyond the regression setting. On real data (Table 2), replacing the regression layer with a classification layer while keeping dual-attention unchanged shows seamless adaptation to classification. IDAT can also include time-independent covariates via simple concatenation (without positional encodings), allowing the model to jointly learn their relationships with longitudinal covariates and the outcome. For multi-dimensional functional input, an intra-functional attention layer can capture cross-channel relations. To balance the variance reduction and pooling bias under measurement noise, a possible extension is to introduce a learnable, data-adaptive gate $\lambda \in [0, 1]$: $\mathcal{T}B^{(\lambda)} = [\lambda A_T + (1-\lambda)A_I] \circ \text{FF}$.

LIMITATIONS

Absolute sinusoidal positional encodings enable universal approximation on fixed maximum sequence lengths (Yun et al., 2019) but may not extrapolate well beyond the training horizon $T$ or to unseen temporal spacings, and are suboptimal when prediction depends on relative timing (e.g., calendar/seasonal features). Alternative encodings from time-series forecasting or continuous-time/relative encodings (Zhou et al., 2021; Woo et al., 2022) may better capture temporal structure. Practical deployments require the tuning of $T$ and $B$ to balance accuracy and efficiency, larger B and T improve predictions but drive up computation. Computing $A_T$ requires $O(T^2)$ in memory and $O(dT^2)$ computation times per subject; whereas for $A_I$, we need $O(B^2)$ in memory and $O(dB^2)$ computation times for each token. Sparsified or local attentions could possibly be more efficient, yet dual-attention scheme still becomes computationally expensive for very long sequences or large batches. Nevertheless, Table 1 shows IDAT remains relatively fast compared to existing Transformer-based methods.

---

[4]Group 1 assigns negative weights in the first third of the trajectory and near-zero weights in the last third, whereas Group 2 shows near-zero weights early and coherent within-cluster structure thereafter.

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
