# A  APPENDIX

## A.1  MODEL ARCHITECTURE

The proposed architecture IDAT as shown in Figure 2 is designed to handle sparse, irregular longitudinal inputs and is applicable to all longitudinal and functional data. The top panel shows two subjects measured at only a few subject-specific times (blue circles and light-blue crosses) with additional measurement errors. Because the full latent trajectories are infinite-dimensional, we discretize the time domain into grid points, then inject positional encodings and apply sampling masks to tokens for unobserved grid points. The resulting input is a tabular form, whose columns index time grid locations, so each column at a time grid acts as a feature, while positional encodings preserve temporal ordering and proximity. The key difference from standard tabular data is the substantial masking induced by sparse and irregular sampling and strong temporal dependence between adjacent features.

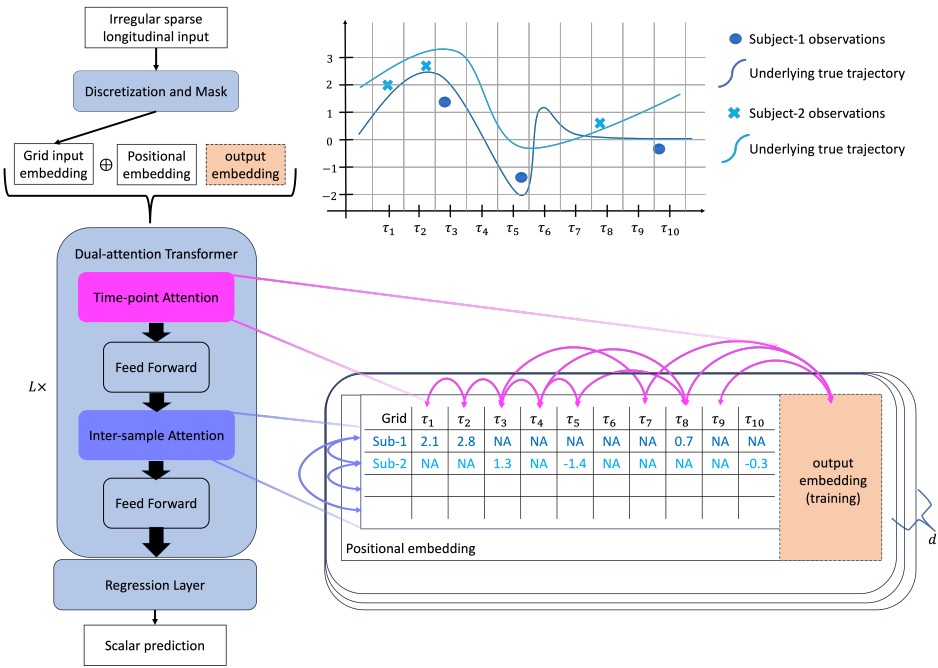

Figure 2: IDAT Model Architecture.

## A.2  SIMULATION

We benchmark our method against a broad set of baselines. Linear regression (LR) is implemented via scikit-learn; FLR and FPCA use the `fdapace`[5] and `fdaconcor`[6] R packages. We select the 99% variance–explained FPCA model, as it outperforms the 90% and 95% FPCA model. TabNet is implemented with `pytorch_tabnet` in PyTorch; we also include `PyTorch` MLP and ResNet. Additional deep tabular models—TabNet, SAINT, FT-Transformer, AutoInt, and NODE—are accessed via the DeepTabular/Mambular [7] stack (Thielmann et al., 2024). TabPFN [8] is employed as in Hollmann et al. (2022). Imputation follows SAND [9] as in Hong et al. (2024). AdaFNN [10] is implemented following Yao et al. (2021). Gradient-boosted baselines (XGBoost, LightGBM, CatBoost)

---

[5] https://CRAN.R-project.org/package=fdapace
[6] https://CRAN.R-project.org/package=fdaconcur
[7] https://github.com/OpenTabular/DeepTabular
[8] https://github.com/PriorLabs/TabPFN
[9] https://github.com/jshong071/SAND/tree/main
[10] https://github.com/jwyyy/AdaFNN/tree/master

use their official Python packages, and AutoGluon (Erickson et al., 2020) results are obtained with `autogluon.tabular`. The xRFM method is implemented with the `xRFM`[11] package in Python.

To mimic longitudinal data, we start with full latent trajectories $X_i(t)$ on a length-100 grid, then subsample grid indices to achieve the target sparsity. In Case I/I* (non-informative sparsity), we subsample uniformly without replacement. In Case II/II* (informative sparsity), we assign sampling weights $w_{i,j} \propto 1+\gamma(|\beta_2(\tau_j)|+|\beta_3(\tau_j)|)$ so time points where the response depends on the trajectory are more likely to be observed. In Case III/III* (cluster-dependent sparsity), weights are $w_{i,j} \propto 1 + \gamma \times [\mathbf{1}(\tau_j \geq 0.5, g = 1) + \mathbf{1}(\tau_j < 0.5, g = 2)]$, so each cluster is sampled more densely in the region where its mean shift differs. Together, these cases cover both non-informative and informative observation schemes. Note that as long as training and testing follow the same sampling plan, IDAT handles both regimes without additional assumptions.

In simulations, we use a 70/20/10 train/validation/test split and randomly mask 10% of target values during training. The model uses 32-dimensional encodings, a 64-unit hidden layer, batch size 128, and a two-layer transformer with 10% dropout. We trained for 500 epochs with learning rate 0.001. Unless stated otherwise, the sample size is $n = 1000$ and functions are sampled on an equally spaced grid of 101 points over $[0, 1]$. Code and scripts will be released for reproducibility upon acceptance. For simulations, we trained on an Apple MacBook Air (Mac15, 12) with an Apple M3 SoC (8-core CPU: 4P+4E), 8GB LPDDR5 unified memory (Hynix), and the integrated Apple GPU via Metal. PyTorch ran on Apple's Metal (MPS) backend; CUDA was not available on this system.

### A.2.1 SIMULATION RESULTS

Table 1 summarizes accuracy and efficiency across 30 simulated cases. IDAT is most consistent overall (Top1: 11/30; Top3: 26/30), followed by its variant without inter–sample attention (*no* $A_I$; Top1: 11/30; Top3: 24/30), with TabPFN ranking third (Top1: 9/30; Top3: 25/30). In noise–free settings the full model dominates, as $A_I$ reduces variance by borrowing strength across similar subjects; under measurement error, similarity can be misestimated and $A_I$ may average over mismatched neighbors, increasing bias—so the time–only variant (*no* $A_I$) can prevail when the bias increase outweighs variance reduction. However, when clustering is apparent in the model, the full model is still outperforms. Empirically, dual-attention yields the largest gains in sparse regimes ($\leq 50\%$ observed; Top1: 9/18, Top3: 17/18), whereas TabPFN is strongest when coverage is denser ($> 50\%$; Top1: 5/12, Top3: 12/12). Notably, among TabPFN's nine Top1 wins, two coincide with LR also being best and four occur when LR is Top3, indicating that TabPFN's prior learned from its pre-training structure captures simple/approximately linear trends well.

In terms of efficiency, IDAT achieves 3.085 ms/sample with 180K parameters, offering a favorable accuracy–latency–size trade–off; TabPFN is substantially larger (11M parameters; 0.047 ms/sample on the server) and slower (1713.843 ms/sample on the local machine; 5.58 ms/sample on the server). AutoGluon ensembles methods such as `WeightedEnsemble L2`, `NeuralNetTorch`, `NeuralNetFastAI`, `LightGBMXT`, and can be fast at inference (1.734 ms/sample) but typically require millions of parameters ($\approx$2M) and still underperform IDAT on Top1/Top3; moreover, achieving competitive accuracy generally entails complexities that make latency comparable to transformer–style models. Classical/GBDT baselines are compact and fast yet rarely Top1/Top3 in this functional setting. Overall, the proposed architecture delivers strong predictive accuracy with modest parameterization and low latency—especially under sparse sampling—while TabPFN is preferable when signal is dense and close to linear. Preprocessing (imputation) times for FPCA/VT/SAND are respectively: Finding FPCA components takes 24.17 $10^{-3}$sec/sample, VT imputation takes 24.79 $10^{-3}$sec/sample, SAND imputation takes 27.79 $10^{-3}$sec/sample. Because mean imputation is computationally negligible, we exclude its runtime from the efficiency comparisons.

Table 3 reports MSE for six simulation scenarios in the very sparse regime (20% observed). Without measurement error, the proposed dual-attention model consistently achieves the lowest MSE. With measurement error, TabPFN slightly leads in the near-linear case I*, while IDAT and its time-point–only variant remain among the top performers; the variant is more noise-robust because it avoids averaging over misidentified neighbors. When clustering is present III*, dual-attention regains a clear advantage, reflecting benefits from cross-subject borrowing.

---

[11]https://arc.net/l/quote/rhehrnrm

Table 3: MSE under different simulation scenarios under very sparse regime (20% observation). The best method is in **bold** and the top three methods are in *italics*. Without measurement error, IDAT consistently outperforms all baselines. With measurement error, in case I* (near-linear ground truth), TabPFN leverages pre-training and slightly edges our models, while IDAT and its time-point–only variant remain among the top performers. The variant (no $A_I$) is more robust to noise because it avoids averaging over misidentified neighbors. When clustering is present (case I*), dual-attention regains a clear advantage.

| Method/MSE | case I | case II | case III | case I* | case II* | case III* |
|---|---|---|---|---|---|---|
| LR | 0.0153 | 0.0108 | 0.0187 | 0.0075 | 0.0069 | 0.0766 |
| FLR | 0.0589 | 0.0039 | 0.0316 | 0.0532 | 0.0066 | 0.1341 |
| FPCA+NN | 0.0406 | 0.0336 | 0.0176 | 0.0561 | 0.0272 | 0.1826 |
| TabNet | 0.0477 | 0.0107 | 0.0316 | 0.0433 | 0.0080 | 0.0659 |
| SAINT | 0.0745 | 0.0109 | 0.2591 | 0.0731 | 0.0108 | 0.2583 |
| FTTransformer | 0.0116 | 0.0036 | 0.0248 | 0.0136 | 0.0037 | 0.0596 |
| AutoInt | 0.0168 | 0.0046 | 0.0295 | 0.0124 | 0.0049 | 0.0666 |
| TabPFN | 0.0083 | *0.0021* | *0.0029* | **0.0067** | *0.0017* | *0.0305* |
| VT+NN | 0.0116 | 0.0168 | 0.0183 | 0.1767 | 0.0757 | 0.0311 |
| SAND+NN | 0.0103 | 0.0104 | 0.0114 | 0.1979 | 0.0856 | 0.0308 |
| MLP | 0.0138 | 0.0031 | 0.0174 | 0.0128 | 0.0029 | 0.0393 |
| ResNet | 0.0128 | 0.0025 | 0.0206 | 0.0147 | 0.0030 | 0.0434 |
| AdaFNN | 0.0127 | 0.0171 | 0.0032 | 0.0091 | 0.0065 | 0.0317 |
| NODE | 0.0126 | 0.0034 | 0.0060 | 0.0111 | 0.0035 | 0.0397 |
| CatBoost | 0.0113 | 0.0041 | 0.0123 | 0.0102 | 0.0029 | 0.0405 |
| XGBoost | 0.0180 | 0.0060 | 0.0287 | 0.0186 | 0.0030 | 0.0525 |
| LightGBM | 0.0189 | 0.0070 | 0.0178 | 0.0152 | 0.0079 | 0.0404 |
| AutoGluon | 0.0116 | 0.0031 | 0.0131 | 0.0074 | 0.0022 | 0.0357 |
| xRFM | *0.0080* | 0.0025 | 0.0128 | 0.0100 | 0.0046 | 0.0342 |
| IDAT | **0.0063** | **0.0008** | **0.0019** | *0.0070* | *0.0015* | **0.0139** |
| IDAT w/o $A_I$ | *0.0077* | 0.0018 | 0.0020 | *0.0069* | **0.0007** | 0.0207 |

In Tables 4, 5, 6, 7, 8, and 9, we report MSE across five sparsity levels for three simulation settings and their measurement error variants, highlighting the best method in **bold** and the top three in *italics*. Although IDAT is designed for sparse observations as it accommodates sampling plan with its masking, it surprisingly achieves competitive—and often state-of-the-art—performance in several dense and fully observed regimes. It consistently attains the best performance in the very- and super-sparse regimes ($< 50\%$ observed). As observations become denser, the transformer-based tabular foundation model TabPFN is often competitive, while AutoGluon remains robust—benefiting from multi-model ensembling and stacking—and performs strongly in several settings. AdaFNN, which adopts a basis-expansion design tailored to fully observed trajectories, performs well when the underlying basis/index structure aligns with the data, but can struggle when the signal departs from an index-model form or under substantial sparsity.

In Figures 3, 4, and 5, we plot log-MSE versus sparsity (x-axis: 0% to 100%). For each setting, the left panel shows results without measurement error and the right panel shows the same setting with measurement error. As expected for transformer-based methods (shown in blue), models that learn end-to-end from $(X, Y)$ are comparatively robust to measurement error, whereas imputation-only variants (SAND+NN and VT+NN) that do not condition on $Y$ are more sensitive. Our model, as well as most of the end-to-end transformer-based models, remain relatively robust to noise.

In Figure 6 (cf. Figure 1b), removing inter-sample attention collapses the separation between clusters: their profiles become much more similar, differing only slightly early in the trajectory. When clustering is present, such averaging across groups can introduce bias in the approximation.

Table 4: MSE under different sparsity level in case I. The best method is in **bold** and the top three methods are in *italics*. IDAT achieves the smallest MSE in sparse regimes and remains competitive as observations become denser; strong competitors include TabPFN and AutoGluon, while AdaFNN is tailored to dense/fully observed data.

| Method/MSE | ssparse 10% | vsparse 20% | sparse 50% | dense 80% | full 100% |
|---|---|---|---|---|---|
| LR | 0.0207 | 0.0153 | 0.0029 | *0.0006* | < **0.0001** |
| FLR | 0.0548 | 0.0589 | 0.0519 | 0.0519 | 0.0546 |
| FPCA+NN | 0.0356 | 0.0406 | 0.0365 | 0.0346 | 0.0335 |
| TabNet | 0.0422 | 0.0477 | 0.0169 | 0.0112 | 0.0064 |
| SAINT | 0.0734 | 0.0745 | 0.0727 | 0.0707 | 0.0757 |
| FTTransformer | 0.0224 | 0.0116 | 0.0050 | 0.0014 | 0.0006 |
| AutoInt | 0.0240 | 0.0168 | 0.0069 | 0.0056 | 0.0026 |
| TabPFN | *0.0157* | 0.0083 | *0.0018* | **0.0005** | *0.0001* |
| VT+NN | 0.0267 | 0.0116 | 0.0065 | 0.0074 | 0.0064 |
| SAND+NN | 0.0256 | 0.0103 | 0.0058 | 0.0063 | 0.0064 |
| MLP | 0.0214 | 0.0138 | 0.0048 | 0.0013 | 0.0011 |
| ResNet | 0.0307 | 0.0128 | 0.0052 | 0.0019 | 0.0009 |
| AdaFNN | 0.0321 | 0.0127 | 0.0030 | *0.0007* | *0.0001* |
| NODE | 0.0199 | 0.0126 | 0.0051 | 0.0012 | 0.0007 |
| CatBoost | 0.0195 | 0.0113 | 0.0034 | 0.0012 | 0.0005 |
| XGBoost | 0.0264 | 0.0180 | 0.0078 | 0.0055 | 0.0034 |
| LightGBM | 0.0266 | 0.0189 | 0.0043 | 0.0036 | 0.0019 |
| AutoGluon | **0.0150** | 0.0116 | 0.0029 | *0.0007* | *0.0002* |
| xRFM | 0.0169 | *0.0080* | 0.0043 | 0.0022 | 0.0025 |
| IDAT | *0.0165* | **0.0063** | *0.0024* | *0.0006* | 0.0003 |
| IDAT w/o $A_I$ | 0.0169 | *0.0077* | **0.0016** | *0.0006* | 0.0003 |

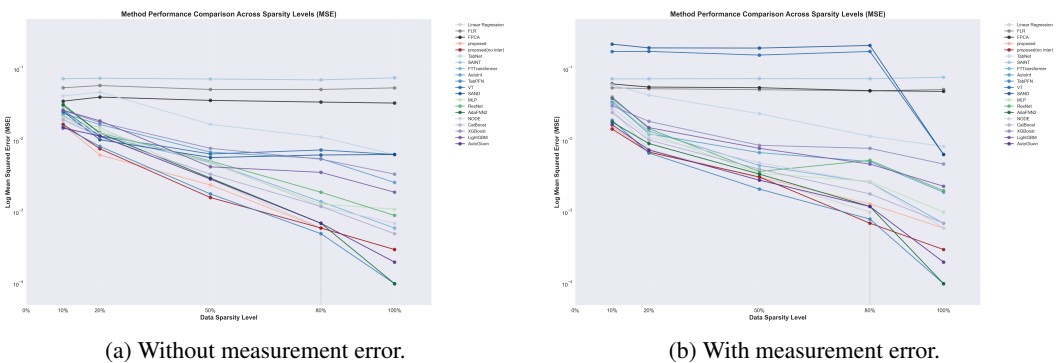

(a) Without measurement error.      (b) With measurement error.

Figure 3: (log) MSE under different sparsity level in case I.

### A.2.2 HYPERPARAMETER SENSITIVITY

IDAT operates on embeddings of shape $B \times d \times (T + 1)$, where $B$ is the batch size and $d$ is the embedding dimension. Under the i.i.d. case, $B$ controls the variance shrinkage of $A_I$, as shown in Lemma 9. The grid length $T$ affects the mesh size and thus the discretization error, as proved in Lemma 3. We use Case I (i.i.d. functional linear regression) in Table 10 to demonstrate how these parameters affect MSE performance.

Matching our theoretical analysis, finer grids (smaller mesh) mitigate discretization bias and reduce MSE, matching our theoretical analysis; whereas coarser grids inflate bias and hence MSE. Larger batch sizes, under this i.i.d. setting, yield stronger variance reduction through inter-sample attention, further lowering MSE.

Table 5: MSE under different sparsity level in case II. The best method is in **bold** and the top three methods are in *italics*. IDAT achieves the smallest MSE in all sparsity regime; strong competitors include TabPFN, AutoGluon and some ensemble models.

| Method/MSE | ssparse 10% | vsparse 20% | sparse 50% | dense 80% | full 100% |
|---|---|---|---|---|---|
| LR | 0.0096 | 0.0108 | 0.0097 | 0.0101 | 0.0073 |
| FLR | 0.0064 | 0.0039 | 0.0080 | 0.0072 | 0.0059 |
| FPCA+NN | 0.0533 | 0.0336 | 0.0320 | 0.0240 | 0.0492 |
| TabNet | 0.0198 | 0.0107 | 0.0018 | 0.0027 | 0.0033 |
| SAINT | 0.0110 | 0.0109 | 0.0107 | 0.0110 | 0.0025 |
| FTTransformer | 0.0066 | 0.0036 | 0.0015 | 0.0018 | 0.0015 |
| AutoInt | 0.0052 | 0.0046 | 0.0062 | 0.0040 | 0.0025 |
| TabPFN | 0.0036 | *0.0021* | *0.0006* | *0.0002* | *0.0002* |
| VT+NN | 0.0145 | 0.0224 | 0.0174 | 0.0135 | 0.0158 |
| SAND+NN | 0.0189 | 0.0172 | 0.0169 | 0.0136 | 0.0158 |
| MLP | 0.0084 | 0.0031 | 0.0036 | 0.0031 | 0.0023 |
| ResNet | 0.0114 | 0.0025 | 0.0035 | 0.0015 | 0.0015 |
| AdaFNN | 0.0281 | 0.0171 | 0.0066 | 0.0043 | 0.0006 |
| NODE | 0.0050 | 0.0034 | *0.0009* | 0.0011 | 0.0005 |
| CatBoost | 0.0031 | 0.0041 | 0.0014 | 0.0004 | *0.0002* |
| XGBoost | *0.0028* | 0.0060 | 0.0011 | 0.0010 | *0.0003* |
| LightGBM | 0.0092 | 0.0070 | 0.0025 | 0.0006 | 0.0012 |
| AutoGluon | 0.0049 | 0.0031 | 0.0010 | *0.0003* | *0.0001* |
| xRFM | 0.0064 | 0.0025 | 0.0017 | 0.0024 | 0.0025 |
| IDAT | **0.0020** | **0.0008** | **0.0003** | *0.0002* | $<$ **0.0001** |
| IDAT w/o $A_I$ | *0.0029* | *0.0018* | **0.0003** | **0.0001** | *0.0002* |

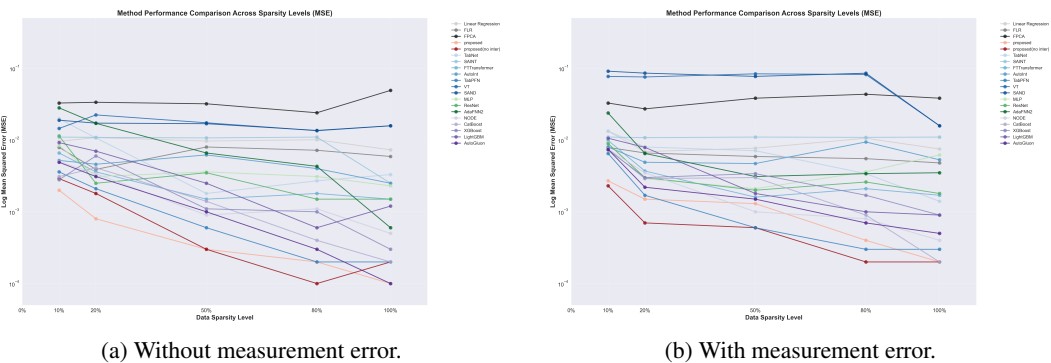

(a) Without measurement error.     (b) With measurement error.

Figure 4: (log) MSE under different sparsity level in case II.

## A.3 REAL DATA

We compare against mean imputation, MICE, and LOCF. LOCF is applied only to the synthetic HIV data; in NCDS the measurement times are highly irregular and widely spaced, making last-observation carry-forward (e.g., for BMI) inappropriate. FPCA-based methods, SAND, and VT already perform imputation internally; thus these baselines—and our imputation-free approach—are unaffected by the external imputation choice. We use a 70/20/10 train/validation/test split and randomly mask 10% of target values during training. To accommodate larger sample sizes and more complex real-data scenarios, we use 128-dimensional encodings, a batch size of 256, and a three-layer transformer with a 10% dropout rate. The server we equipped is a NVIDIA A100 80GB PCIe GPU (driver 535.261.03, CUDA 12.2; MIG enabled) paired with an AMD EPYC 9554P 64-core CPU (128 threads) and 755 GiB system RAM, running Rocky Linux 8.10.

Table 6: MSE under different sparsity level in case III. The best method is in **bold** and the top three methods are in *italics*. IDAT achieves the smallest MSE in sparse regimes and remains competitive as observations become denser; strong competitors include TabPFN and AutoGluon, while AdaFNN is tailored to dense/fully observed data.

| Method/MSE | ssparse 10% | vsparse 20% | sparse 50% | dense 80% | full 100% |
|---|---|---|---|---|---|
| LR | 0.0364 | 0.0187 | 0.0084 | 0.0012 | **0.0002** |
| FLR | 0.0604 | 0.0316 | 0.0291 | 0.0210 | 0.0245 |
| FPCA+NN | 0.0288 | 0.0176 | 0.0144 | 0.0128 | 0.0104 |
| TabNet | 0.0585 | 0.0316 | 0.0075 | 0.0065 | 0.0029 |
| SAINT | 0.2594 | 0.2591 | 0.2584 | 0.2601 | 0.2510 |
| FTTransformer | 0.0440 | 0.0248 | 0.0101 | 0.0047 | 0.0021 |
| AutoInt | 0.0550 | 0.0295 | 0.0147 | 0.0083 | 0.0068 |
| TabPFN | *0.0078* | *0.0029* | **0.0006** | **0.0003** | **0.0002** |
| VT+NN | 0.0409 | 0.0183 | 0.0054 | 0.0053 | 0.0056 |
| SAND+NN | 0.0282 | 0.0114 | 0.0070 | 0.0052 | 0.0056 |
| MLP | 0.0195 | 0.0174 | 0.0060 | 0.0042 | 0.0014 |
| ResNet | 0.0267 | 0.0206 | 0.0085 | 0.0050 | 0.0018 |
| AdaFNN | 0.0169 | 0.0032 | *0.0011* | *0.0005* | **0.0002** |
| NODE | 0.0184 | 0.0060 | 0.0039 | 0.0018 | 0.0008 |
| CatBoost | 0.0358 | 0.0123 | 0.0037 | 0.0026 | 0.0008 |
| XGBoost | 0.0465 | 0.0287 | 0.0083 | 0.0091 | 0.0023 |
| LightGBM | 0.0476 | 0.0178 | 0.0064 | 0.0036 | 0.0011 |
| AutoGluon | 0.0172 | 0.0131 | *0.0019* | 0.0007 | **0.0002** |
| xRFM | 0.0287 | 0.0128 | 0.0053 | 0.0020 | 0.0010 |
| IDAT | **0.0043** | **0.0019** | **0.0006** | *0.0005* | *0.0003* |
| IDAT w/o $A_I$ | *0.0055* | *0.0020* | *0.0011* | *0.0004* | *0.0005* |

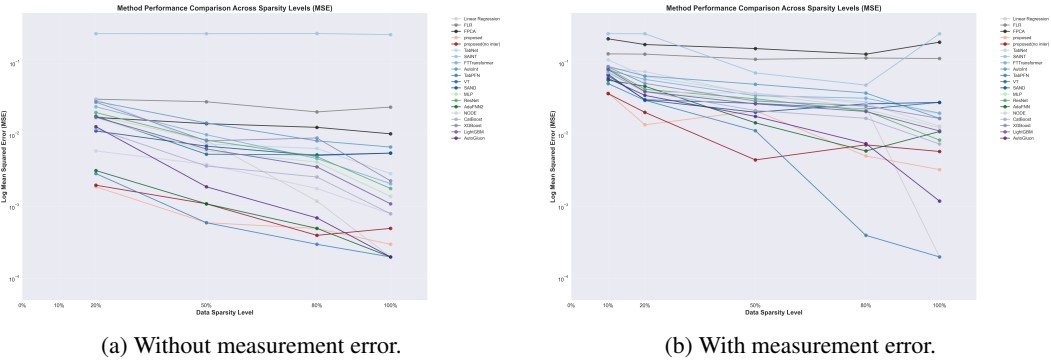

(a) Without measurement error.          (b) With measurement error.

Figure 5: (log) MSE under different sparsity level in case III.

### A.3.1 NATIONAL CHILD DEVELOPMENT STUDY: AGE 62 BMI PREDICTION

The 1958 National Child Development Study (NCDS) is a nationally representative UK birth cohort run by the Center for Longitudinal Studies (CLS) at UCL. Initiated as the Perinatal Mortality Survey, it has followed over 17000 individuals born in a single week in March 1958 across eleven major sweeps at ages 7, 11, 16, 23, 33, 42, 44, 46, 50, 55, and 62. Achieved sample sizes declined from $n = 17415$ at birth (and age 7) to $n = 9790$ (age 55), and $n = 9137$ (age 62). Because all participants were assessed at the same discrete ages, no further time alignment is required. After restricting to participants with valid baseline covariates (measured at age 7) and a valid age-62 BMI, the analytic sample comprises 4952 individuals. Adult BMI shows turning points near ages 65 and 80 (Dahl et al., 2014); thus, predicting BMI at age 62 is reasonable, as the trajectory has not yet crossed a change point.

Table 7: MSE under different sparsity level in case I* (with measurement error). The best method is in **bold** and the top three methods are in *italics*. IDAT achieves the smallest MSE in the super sparse regime and remains competitive as observations become denser; strong competitors include TabPFN and xRFM, while AdaFNN is tailored only to fully observed data.

| Method/MSE | ssparse 10% | vsparse 20% | sparse 50% | dense 80% | full 100% |
|---|---|---|---|---|---|
| LR | 0.0272 | 0.0075 | *0.0030* | *0.0010* | **<0.0001** |
| FLR | 0.0548 | 0.0532 | 0.0515 | 0.0495 | 0.0520 |
| FPCA+NN | 0.0620 | 0.0561 | 0.0549 | 0.0499 | 0.0488 |
| TabNet | 0.0604 | 0.0433 | 0.0238 | 0.0115 | 0.0083 |
| SAINT | 0.0731 | 0.0731 | 0.0736 | 0.0734 | 0.0769 |
| FTTransformer | 0.0332 | 0.0136 | 0.0045 | 0.0026 | 0.0007 |
| AutoInt | 0.0357 | 0.0124 | 0.0068 | 0.0051 | 0.0019 |
| TabPFN | 0.0193 | **0.0067** | **0.0021** | *0.0008* | *0.0001* |
| VT+NN | 0.1758 | 0.1767 | 0.1570 | 0.1763 | 0.0064 |
| SAND+NN | 0.2233 | 0.1979 | 0.1967 | 0.2140 | 0.0064 |
| MLP | 0.0378 | 0.0128 | 0.0035 | 0.0027 | 0.0010 |
| ResNet | 0.0407 | 0.0147 | 0.0037 | 0.0053 | 0.0020 |
| AdaFNN | 0.0182 | 0.0091 | 0.0034 | 0.0012 | *0.0001* |
| NODE | 0.0289 | 0.0111 | 0.0049 | 0.0026 | 0.0006 |
| CatBoost | 0.0246 | 0.0102 | 0.0040 | 0.0018 | 0.0007 |
| XGBoost | 0.0307 | 0.0186 | 0.0086 | 0.0078 | 0.0047 |
| LightGBM | 0.0391 | 0.0152 | 0.0078 | 0.0047 | 0.0023 |
| AutoGluon | 0.0167 | 0.0074 | *0.0028* | 0.0012 | *0.0002* |
| xRFM | *0.0160* | 0.0100 | 0.0042 | 0.0029 | 0.0028 |
| IDAT | *0.0157* | *0.0070* | 0.0031 | 0.0013 | 0.0006 |
| IDAT w/o $A_I$ | **0.0145** | *0.0069* | 0.0031 | **0.0007** | 0.0003 |

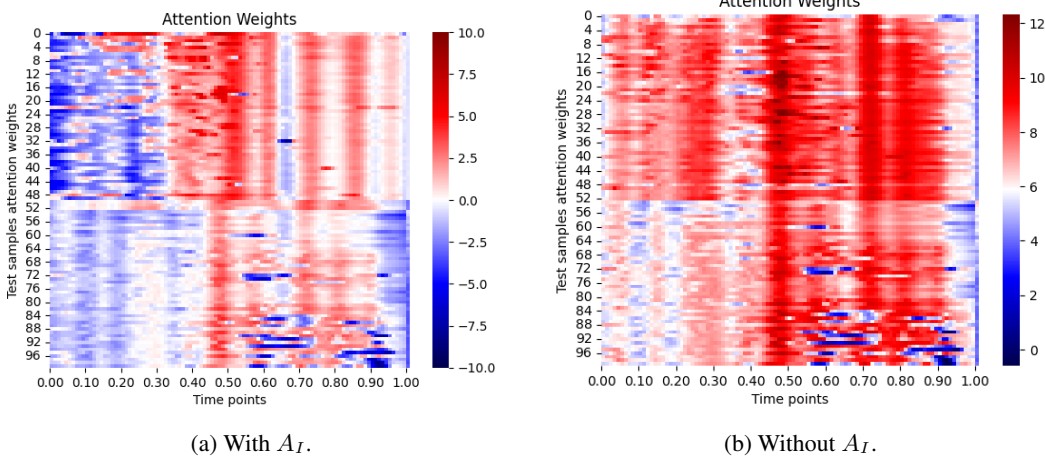

(a) With $A_I$.  (b) Without $A_I$.

Figure 6: Embedding clustering in case III, in comparison with and without $A_I$. All the attention weights are are scaled by a factor of 1000. Removing inter-sample attention collapses the separation between clusters: their profiles become much more similar, differing only slightly early in the trajectory.

Anthropometric measurements included body mass index (BMI) and other measurements such as body fat percentage. The age, sex, and height of the respondent were recorded. Weight was measured in kilograms using Tanita BF-522W scales (upper limit 130 kg; those likely exceeding this were not weighed). BMI was computed as

$$\text{BMI} = \text{weight (kg)} / \left( \text{height (m)}^2 \right).$$

Table 8: MSE under different sparsity level in case II* (with measurement error). The best method is in **bold** and the top three methods are in *italics*. IDAT achieves the smallest MSE in all sparsity regimes; strong competitors include TabPFN and ensemble methods such as NODE and CatBoost.

| Method/MSE | ssparse 10% | vsparse 20% | sparse 50% | dense 80% | full 100% |
|---|---|---|---|---|---|
| LR | 0.0133 | 0.0069 | 0.0077 | 0.0106 | 0.0075 |
| FLR | 0.0079 | 0.0066 | 0.0059 | 0.0055 | 0.0048 |
| FPCA+NN | 0.0328 | 0.0272 | 0.0383 | 0.0435 | 0.0383 |
| TabNet | 0.0133 | 0.0080 | 0.0071 | 0.0034 | 0.0014 |
| SAINT | 0.0110 | 0.0108 | 0.0110 | 0.0109 | 0.0110 |
| FTTransformer | 0.0099 | 0.0037 | 0.0016 | 0.0021 | 0.0017 |
| AutoInt | 0.0080 | 0.0049 | 0.0047 | 0.0094 | 0.0053 |
| TabPFN | 0.0065 | *0.0017* | **0.0006** | *0.0003* | *0.0003* |
| VT+NN | 0.0771 | 0.0757 | 0.0837 | 0.0820 | 0.0158 |
| SAND+NN | 0.0911 | 0.0856 | 0.0771 | 0.0854 | 0.0158 |
| MLP | 0.0091 | 0.0029 | 0.0021 | 0.0036 | 0.0062 |
| ResNet | 0.0089 | 0.0030 | 0.0020 | 0.0026 | 0.0018 |
| AdaFNN | 0.0282 | 0.0065 | 0.0031 | 0.0034 | 0.0035 |
| NODE | 0.0075 | 0.0035 | *0.0010* | 0.0008 | *0.0004* |
| CatBoost | 0.0075 | 0.0029 | 0.0030 | 0.0009 | **0.0002** |
| XGBoost | 0.0073 | 0.0030 | 0.0034 | 0.0017 | 0.0009 |
| LightGBM | 0.0106 | 0.0079 | 0.0018 | 0.0010 | 0.0009 |
| AutoGluon | 0.0074 | 0.0022 | 0.0015 | 0.0007 | 0.0005 |
| xRFM | *0.0058* | 0.0046 | 0.0019 | 0.0027 | 0.0029 |
| IDAT | *0.0027* | *0.0015* | *0.0013* | *0.0004* | **0.0002** |
| IDAT w/o $A_I$ | **0.0023** | **0.0007** | **0.0006** | **0.0002** | **0.0002** |

The longitudinal BMI series exhibit substantial missingness (mean 25%, range 8%–96%), spanning dense to super-sparse regimes; hence a model that remains robust across observation densities is desirable. On average, subjects have 6.2 observations (SD 0.9) across the 10 sweeps (Figure 7). The BMI outcome at age 62 averages 27.95 for females ($n = 2489$) and 28.25 for males ($n = 2463$); values are truncated to $[19.39, 41.14]$. We adjust for birth BMI given its established association with later-life BMI (Parsons et al., 1999; Rogers, 2003; Gillman et al., 2003), and for a childhood stress index defined as the average of 13 binary indicators in the NCDS (e.g., housing problems, financial hardship, parental divorce, unemployment, illness, disability, or bereavement) (Halliday et al., 2014; Stenhammar et al., 2010; Garasky et al., 2009). Because sex differences in BMI are marked across the life course, we formally compared female vs. male BMI distributions at ages 7, 11, 16, 23, 33, 42, 50, 55, and 62 using two-sided Mann–Whitney U tests, obtaining $p$-values of $3.99 \times 10^{-6}$, $9.92 \times 10^{-7}$, $1.93 \times 10^{-22}$, $1.23 \times 10^{-35}$, $6.74 \times 10^{-40}$, $5.60 \times 10^{-42}$, $3.52 \times 10^{-2}$, $1.37 \times 10^{-14}$, and $3.13 \times 10^{-6}$, respectively (all $< 0.05$; age 50 remains significant at $p = 0.035$). Given persistent differences along the entire trajectory, we fit sex-stratified models. Sex differences are well documented in obesity prevalence, maternal factors further contribute to sex-specific BMI trajectories (Gillman et al., 2003). Sex-stratified models thus help mitigate confounding bias in the estimated effects of the primary predictors.

Following CLS guidance on handling missing data, we assessed whether our pre-specified covariates predicted non-response for the outcome (Mostafa et al., 2021; Katsoulis et al., 2024). We found no evidence that these covariates were significant predictors of attrition, suggesting negligible selection bias in the analytic sample. Accordingly, we did not apply explicit non-response adjustments (e.g., inverse probability weighting), while noting that non-response in longitudinal studies can reduce efficiency and, if unaddressed, potentially induce bias.

The detailed MSE/MAE under two imputation methods for the BMI prediction task is provided in Table 11. As shown in Figure 8, we cluster the learned dual–attention weights (unsupervised; $k$ chosen by cross–validated $k$NN) into two groups. The resulting labels are highly associated with $Y$ (age-62 BMI), indicating that inter–sample attention captures prediction–relevant structure. The clusters

Table 9: MSE under different sparsity level in case III* (with measurement error). The best method is in **bold** and the top three methods are in *italics*. IDAT achieves the smallest MSE in sparse regimes and remains competitive as observations become denser; strong competitors include TabPFN and LR in dense/fully observed data.

| Method/MSE | ssparse 10% | vsparse 20% | sparse 50% | dense 80% | full 100% |
|---|---|---|---|---|---|
| LR | 0.0853 | 0.0766 | 0.0367 | 0.0244 | **0.0002** |
| FLR | 0.1353 | 0.1341 | 0.1137 | 0.1184 | 0.1166 |
| FPCA+NN | 0.2192 | 0.1826 | 0.1604 | 0.1339 | 0.1968 |
| TabNet | 0.1120 | 0.2613 | 0.0377 | 0.0291 | 0.0198 |
| SAINT | 0.2585 | 0.2583 | 0.0734 | 0.0497 | 0.2579 |
| FTTransformer | 0.0801 | 0.0596 | 0.0354 | 0.0326 | 0.0202 |
| AutoInt | 0.0903 | 0.0666 | 0.0510 | 0.0385 | 0.0171 |
| TabPFN | 0.0522 | *0.0305* | *0.0115* | **0.0004** | **0.0002** |
| VT+NN | 0.0673 | 0.0311 | 0.0276 | 0.0231 | 0.0285 |
| SAND+NN | 0.0703 | 0.0308 | 0.0208 | 0.0272 | 0.0285 |
| MLP | 0.0769 | 0.0393 | 0.0307 | 0.0228 | 0.0127 |
| ResNet | 0.0846 | 0.0434 | 0.0320 | 0.0217 | 0.0085 |
| AdaFNN | 0.0582 | 0.0479 | 0.0148 | *0.0060* | 0.0112 |
| NODE | 0.0741 | 0.0397 | 0.0383 | 0.0275 | 0.0133 |
| CatBoost | 0.0740 | 0.0405 | 0.0220 | 0.0171 | 0.0075 |
| XGBoost | 0.0884 | 0.0525 | 0.0297 | 0.0255 | 0.0170 |
| LightGBM | 0.0817 | 0.0404 | 0.0272 | 0.0213 | 0.0114 |
| AutoGluon | 0.0617 | 0.0357 | 0.0182 | 0.0076 | *0.0012* |
| xRFM | *0.0417* | 0.0342 | 0.0141 | 0.0075 | 0.0044 |
| IDAT | *0.0386* | **0.0139** | **0.0016** | *0.0051* | *0.0033* |
| IDAT w/o $A_I$ | **0.0378** | *0.0207* | *0.0045* | 0.0073 | 0.0059 |

Table 10: Hyperparameter sensitivity analysis on Simulation Case I illustrating how grid size (mesh resolution) and batch size affect performance.

| Hyperparameters/ Sparsity | ssparse (10%) | vsparse (20%) | sparse (50%) | dense (80%) | full(100%) |
|---|---|---|---|---|---|
| Larger Mesh ($T = 50, B = 128$) | 0.0353 | 0.0253 | 0.0062 | 0.0012 | 0.0004 |
| Baseline ($T = 100, B = 128$) | 0.0165 | 0.0063 | 0.0024 | 0.0006 | 0.0003 |
| Smaller Mesh ($T = 200, B = 128$) | 0.0112 | 0.0038 | 0.0018 | 0.0004 | 0.0002 |
| Smaller Batch ($T = 100, B = 64$) | 0.0232 | 0.0088 | 0.0033 | 0.0007 | 0.0003 |
| Baseline ($T = 100, B = 128$) | 0.0165 | 0.0063 | 0.0024 | 0.0006 | 0.0003 |
| Bigger Batch ($T = 100, B = 256$) | 0.0119 | 0.0042 | 0.0019 | 0.0006 | 0.0003 |

cleanly separate low vs. high BMI at age 62, demonstrating explanatory value and interpretability. Panel (a) reveals two clear attention clusters and a consistently low–weight, non–informative region at the 6th time grid (age-50); panel (b) shows that the attention–derived clusters align with subsequent BMI separation, with the shaded non–informative interval being the only segment where trajectories do not mirror the later outcome.

In the NCDS age–50 sweep, height was not newly measured: healthcare professionals first used the measured height from the age–44 biomedical sweep; if unavailable, the self–reported height from the age–50 main interview. Consequently, the dedicated "height at 50" field appears missing for the vast majority of cases (about 96% in the analyzed population), reducing the informativeness of BMI at age 50. Together with the attention–based time–domain selection, these collection procedures help explain why the age–50 window is identified as non–informative in Figure 8.

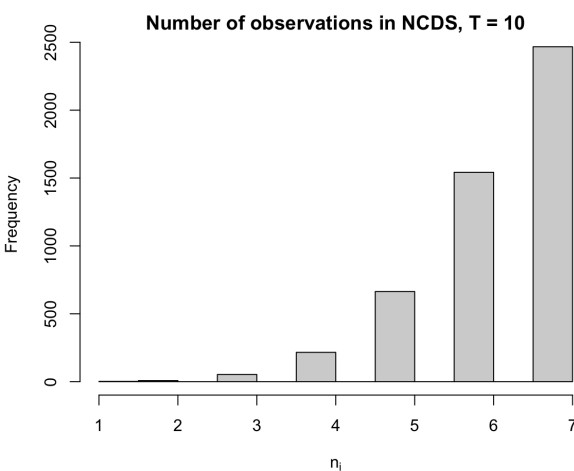

Figure 7: Number of observations in the NCDS study of sample size $n = 4952$.

Table 11: Prediction performance on the NCDS BMI task under two imputation strategies (mean, MICE). The best method is in **bold** and the top three are in *italics*.

| | NCDS BMI | | | |
| | mean imputed | | MICE | |
| Method | MSE | MAE | MSE | MAE |
|---|---|---|---|---|
| LM/GLM | 21.2318 | 3.5864 | 9.1630 | 2.2935 |
| FLR/FGLM | 11.1655 | 2.5682 | 11.1655 | 2.5682 |
| FPCA+NN | 10.2562 | 2.3870 | 10.2562 | 2.3870 |
| TabNet | 9.6078 | 2.3381 | *8.1461* | 2.1843 |
| SAINT | 9.7213 | 2.3524 | 9.4844 | 2.3231 |
| FTTransformer | 9.8154 | 2.3606 | 9.2480 | 2.3276 |
| AutoInt | 8.6933 | 2.2375 | 8.7121 | 2.2607 |
| TabPFN | *8.5794* | **2.1621** | 8.5164 | **2.1503** |
| VT+NN | 19.1824 | 3.3388 | 19.1824 | 3.3388 |
| SAND+NN | 19.1647 | 3.3413 | 19.1647 | 3.3413 |
| MLP | 9.4439 | 2.3142 | 9.2378 | 2.3191 |
| ResNet | 9.3321 | 2.3526 | 8.9923 | 2.2780 |
| AdaFNN | 9.2237 | 2.2716 | 8.6085 | 2.2147 |
| NODE | 9.6511 | 2.2955 | 8.6911 | 2.2528 |
| CatBoost | 9.5655 | 2.3125 | 9.1693 | 2.2969 |
| XGBoost | 10.9604 | 2.4617 | 10.3945 | 2.4074 |
| LightGBM | 9.5160 | 2.3066 | 9.9092 | 2.3795 |
| AutoGluon | 9.0408 | 2.2623 | 8.9595 | 2.2543 |
| xRFM | 11.5200 | 2.5628 | 20.5166 | 2.7625 |
| IDAT | **8.0061** | *2.1728* | **8.0061** | *2.1728* |
| IDAT w/o $A_I$ | *8.1177* | *2.1966* | *8.1177* | *2.1966* |

### A.3.2 SYNTHETIC HIV DATASET (HEALTH GYM): VL/CD4 CLASSIFICATION

Each dataset in the Health Gym HIV dataset (Kuo et al., 2022) project is generated by training a generative adversarial network (GAN) on a corresponding real cohort to reproduce marginal distributions, temporal dynamics, and cross–variable correlations while maintaining a very low re-identification risk. In this study, we use the EuResist–based (Zazzi et al., 2012) real HIV cohort

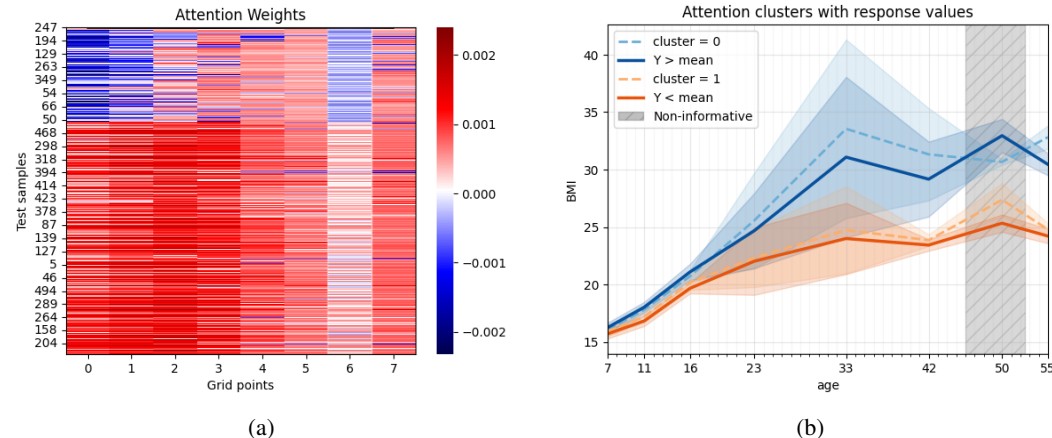

(a)                                    (b)

Figure 8: Meaningful dual-attention structure and outcome alignment across time. (a) Attention weights for test samples. Two clusters are clearly visible, and the masked region (6th time grid, age 50) receives consistently low attention, indicating a non-informative domain. (b) Attention-derived clusters align with lower vs. higher BMI at age 62; the shaded non-informative domain is the only region where cluster trajectories fail to mirror the subsequent BMI separation.

and then sampling fully synthetic patient trajectories from the trained generator. The GAN is optimized to reproduce not only marginal distributions but also temporal dynamics and cross–variable dependencies (e.g., the joint evolution of viral load, CD4, and regimen exposure). Utility is validated by comparing synthetic to real data across distributions, correlations, regimen usage frequencies, and time–series summaries; privacy is audited using best–practice membership– and attribute–disclosure tests, yielding a very low re–identification risk while preserving clinically meaningful patterns.

The EuResist–based source cohort includes individuals who initiated antiretroviral therapy (ART) after 2015 and were treated with the 50 most common regimen combinations (covering 21 drugs). Variables comprise demographics, viral load (VL), CD4 T–cell counts, and regimen indicators (both base combinations and auxiliary drug classes). Records are monthly time series with person–specific therapy durations; to standardize sequence lengths for modeling, trajectories are truncated to the nearest multiple of 10 months. Because the real cohort is sparse/irregular, each laboratory/clinical variable includes a binary "(M)" flag marking whether a measurement was observed at a given month, preserving real–world missingness for GAN training and downstream evaluation.

The HIV synthetic dataset is based on the work of Parbhoo et al. (2017) and Organization et al. (2013). It encodes the core variables needed to model disease monitoring and treatment decisions in HIV: (i) demographics (e.g., sex, age/age group, ethnicity), (ii) ART regimen indicators (base combinations and auxiliary drug classes), (iii) VL, and (iv) CD4 counts. Two additional identifiers (patient ID and month) index the panel structure. This schema supports supervised prediction, policy learning, and descriptive analyses without access to identifiable records.

In terms of scale and structure, the synthetic HIV dataset contains 8916 patients followed at a fixed monthly cadence for 60 months, yielding 534960 panel records ($8916 \times 60$). The time index runs from month 1 to month 60 with no gaps, so each patient contributes a complete 60–step trajectory (in contrast to the variable–length, irregular trajectories in the underlying real cohort). The data schema comprises 15 variables: 3 numeric (e.g., VL, CD4, and a continuous demographic), 5 binary (e.g., regimen or clinical flags), and 5 categorical (e.g., regimen classes, ethnicity), plus patient ID and month index.

Clinical guidelines recommend frequent viral load monitoring until HIV-1 RNA is suppressed below 200 copies/mL, followed by routine testing, establishing VL < 200 copies/mL as the clinical threshold for virologic suppression (Eisinger et al., 2019; JG, 2008). Suppression at this level is clinically meaningful: a systematic review reports essentially zero sexual transmission risk when individuals adhere to antiretroviral therapy and maintain VL< 200 copies/mL (Sabin et al., 2000). Accordingly, predicting whether future VL will be < 200 copies/mL is a decision-relevant end-

point that informs monitoring intervals, adherence support, and risk communication, and is aligned with clinical practice and epidemiologic evidence. The VL series in the feature window is highly sparse (mean missingness 65%, range 47–88%). Subjects receive a mean of 5.8 observations (SD 1.8) during the 20-month covariate interval (Figure 9). The analysis cohort comprises $n = 8683$ synthetic patients meeting inclusion criteria for the two windows. We adjust for sex as a baseline covariate. The detailed accuracy/F1 score under three imputation methods for the VL classification task is provided in Table 12.

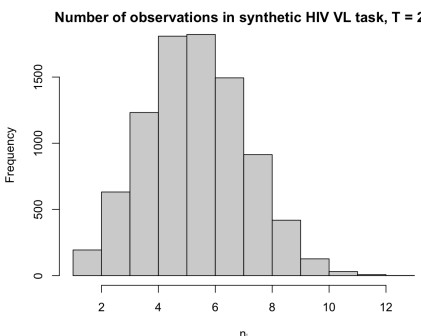 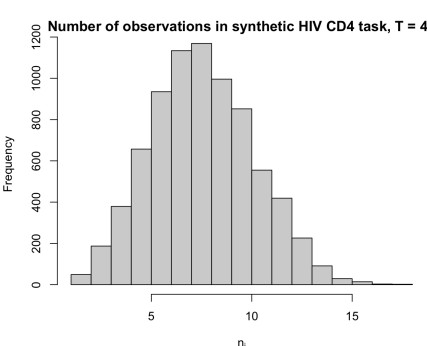

Figure 9: Number of observations in the synthetic HIV study. On the left, VL task with total of $8683$ subjects in $T = 20$ months interval. On the right, CD4 task with total of $7659$ subjects in $T = 40$ months interval.

We would like to report another analysis done on this HIV synthetic dataset in Table 13. AIDS is defined for surveillance by the U.S. CDC (Centers for Disease Control and Prevention, 2025) (and harmonized with WHO (Organization et al., 2013)) as either: (i) CD4 T-cell count $< 200$ cells/mm$^3$, or (ii) the presence of any AIDS-defining illness. In this study we define a binary label indicating whether a patient ever experiences CD4$< 200$ during the evaluation window. For each patient we use the first 40 months of CD4 as the longitudinal covariate vector, and define the outcome as

$$Y = \mathbf{1}\left\{\min_{t \in \{41,\dots,60\}} \text{CD4}_t < 200\right\},$$

i.e., whether CD4 ever drops below 200 in the last 20 months (prediction window). The CD4 series in the feature window is highly sparse (mean missingness 76%, range 49–93%). On average, subjects have 8 (SD 2.6) observations over the 40-month covariate window (Figure 9). The analysis cohort comprises $n = 7659$ synthetic patients meeting inclusion criteria for the two windows. We adjust for sex as a baseline covariate. With only a $6\%$ minority rate, the task is highly imbalanced. Dual-attention can struggle because both its objective and mechanics amplify majority signals while diluting minority cues. Inter-sample attention forms softmax-weighted averages across the batch; with few minority examples per batch, these averages are pulled toward dense majority neighborhoods, washing out rare patterns. If a $Y$-token is included during training for classification, the encoder can "cheat" by attending to that token instead of learning from covariates; at test time the $Y$-token is masked, creating a distribution shift where the learned shortcut disappears and generalization degrades. Finally, imbalanced batches mean the model rarely attends to minority neighbors, preventing stable minority prototypes and yielding poor F1 score and accuracy.

Ensemble methods perform strongly on the classification task, reflecting robustness under relatively short grids and stable outcomes. In this setting, LOCF is reasonable because the observed grid is small and VL varies modestly over successive months, while MICE, although a state-of-the-art imputation strategy, mainly boosts competing models without closing the gap to our approach. Across regression and classification, MICE tends to improve baselines, yet none match the proposed dual-attention Transformer. For classification, especially under severe imbalance, a more tailored class-balanced design, such as class-balanced batching, logit-adjusted losses and validation-based thresholding, can further improve performance and better capture minority cases.

Table 12: Classification performance on the synthetic HIV VL task under three imputation strategies (mean, MICE, LOCF). We report accuracy (ACC) and F1; the best score is in **bold** and the top three are in *italics*. IDAT matches the top methods, and LOCF performs comparably given stable VL trajectories.

| | Synthetic HIV – VL | | | | | |
| | mean imputed | | MICE | | LOCF | |
| Method | ACC | F1 | ACC | F1 | ACC | F1 |
|---|---|---|---|---|---|---|
| LM/GLM | *0.9505* | *0.9746* | **0.9517** | *0.9751* | 0.9448 | 0.9716 |
| FLR/FGLM | 0.9486 | 0.9736 | 0.9486 | 0.9736 | 0.9486 | 0.9736 |
| FPCA+NN | 0.9413 | 0.9698 | 0.9413 | 0.9698 | 0.9413 | 0.9698 |
| TabNet | 0.9482 | 0.9734 | 0.9494 | 0.9740 | 0.9517 | 0.9752 |
| SAINT | **0.9517** | *0.9751* | 0.9505 | 0.9745 | 0.9517 | 0.9752 |
| FTTransformer | 0.9459 | 0.9721 | *0.9494* | 0.9739 | *0.9528* | 0.9757 |
| AutoInt | 0.9425 | 0.9703 | *0.9494* | 0.9740 | *0.9528* | *0.9758* |
| TabPFN | 0.9459 | 0.9722 | 0.9482 | 0.9734 | *0.9528* | *0.9758* |
| VT+NN | 0.9252 | 0.9611 | 0.9252 | 0.9611 | 0.9252 | 0.9611 |
| SAND+NN | 0.9321 | 0.9647 | 0.9321 | 0.9647 | 0.9321 | 0.9647 |
| MLP | **0.9517** | *0.9751* | 0.9471 | 0.9727 | *0.9540* | *0.9764* |
| ResNet | 0.9448 | 0.9715 | **0.9517** | *0.9751* | 0.9517 | 0.9752 |
| AdaFNN | 0.9321 | 0.9649 | 0.9321 | 0.9649 | 0.9321 | 0.9649 |
| NODE | 0.9471 | 0.9727 | **0.9517** | *0.9751* | **0.9551** | **0.9769** |
| CatBoost | 0.9379 | 0.9679 | 0.9448 | 0.9715 | 0.9517 | 0.9751 |
| XGBoost | 0.9333 | 0.9652 | 0.9471 | 0.9727 | 0.9505 | 0.9745 |
| LightGBM | 0.9379 | 0.9678 | 0.9448 | 0.9715 | 0.9482 | 0.9733 |
| AutoGluon | *0.9505* | *0.9746* | *0.9505* | 0.9745 | 0.9517 | 0.9751 |
| xRFM | 0.9413 | 0.9698 | 0.9413 | 0.9698 | 0.9413 | 0.9698 |
| IDAT | **0.9517** | **0.9752** | **0.9517** | **0.9752** | 0.9517 | 0.9752 |
| IDAT w/o $A_I$ | **0.9517** | **0.9752** | **0.9517** | **0.9752** | 0.9517 | 0.9752 |

## A.4 THEORY FOR SCALAR-ON-FUNCTION REGRESSION: NONPARAMETRIC RANDOM-EFFECT MODELS

For subject $i$ in a batch of size $B$, the longitudinal measurements follows a random effect model (Laird & Ware, 1982a; Mu et al., 2008),

$$X_i^*(\tilde{t}_i) = \mu(\tilde{t}_i) + b_i(\tilde{t}_i) + \eta_i(\tilde{t}_i), \tag{8}$$

with $\tilde{t}_i = (t_{i1}, \ldots, t_{in_i}) \subset [0,1]$, where $n_i$ may differ among subjects. The irregular samples $X_i^*(\tilde{t}_i)$ are then viewed as a noisy finite-dimensional samples from the continuous trajectory $X_i(\cdot)$. The effect $\mu$ is the population mean effect which is shared among all subjects and captures the general trajectory trend, while the subject-specific effect $b_i \sim subG(0, \mathcal{B})$ captures the effect on subjects and is distributed independently of each other and of the measurement noise $\eta_i \sim N(0, \sigma_X^2)$. We impose smoothness on both $\mu$ and $b_i$ via the Hölder class $C^\alpha([0,1]; \mathbb{R})$. For $\alpha \in (0,1]$, define

$$C^\alpha([0,1]; \mathbb{R}) = \left\{ g : [0,1] \to \mathbb{R} : \|g\|_{C^\alpha} := \|g\|_\infty + [g]_{C^\alpha} < \infty \right\},$$

where $\|g\|_\infty = \sup_{t \in [0,1]} |g(t)|$, $[g]_{C^\alpha} = \sup_{s \neq t} \frac{|g(t) - g(s)|}{|t-s|^\alpha}$. In particular, $\alpha = 1$, $C^1([0,1])$ is exactly the space of Lipschitz functions in the $\ell_\infty$ norm. The response $Y_i \in \mathbb{R}$ is generated by a functional regression operator $\mathcal{F}$ acting on the entire trajectory:

$$Y_i = \mathcal{F}(X_i(\cdot)) + \epsilon_i, \ \epsilon_i \sim N(0, \sigma_Y^2). \tag{9}$$

We assume $\mathcal{F} : C^\alpha([0,1]; \mathbb{R}) \to [-M_f, M_f] \subset \mathbb{R}$ is bounded and $L_f$-Lipschitz in supremum norm, namely

$$|\mathcal{F}(X) - \mathcal{F}(X')| \leq L_f \|X - X'\|_\infty, \quad \|X - X'\|_\infty = \sup_{t \in [0,1]} |X(t) - X'(t)|.$$

Table 13: Classification performance on the synthetic HIV CD4 task under three imputation strategies (mean, MICE, LOCF). We report accuracy (ACC) and F1; the best score is in **bold** and the top three are in *italics*. Tree/ensemble methods and TabPFN attain the highest ACC, while FPCA+NN yields the best F1, with rankings broadly stable across imputations.

| | Synthetic HIV – CD4 | | | | | |
| | mean imputed | | MICE | | LOCF | |
| Method | ACC | F1 | ACC | F1 | ACC | F1 |
|---|---|---|---|---|---|---|
| LM/GLM | 0.8158 | 0.3095 | 0.8327 | 0.2182 | 0.8210 | 0.1039 |
| FLR/FGLM | 0.8184 | 0.1026 | 0.8184 | 0.1026 | 0.8184 | 0.1026 |
| FPCA+NN | 0.8405 | **0.7648** | 0.8405 | **0.7648** | *0.8405* | **0.7648** |
| TabNet | 0.8132 | 0.2258 | 0.8495 | 0.4579 | 0.8366 | 0.3942 |
| SAINT | 0.8249 | 0.1176 | 0.8171 | 0.3092 | 0.8262 | 0.1625 |
| FTTransformer | *0.8508* | *0.5022* | 0.8405 | 0.4279 | 0.8353 | *0.4641* |
| AutoInt | 0.8457 | 0.4138 | 0.8353 | 0.3981 | 0.8327 | 0.3385 |
| TabPFN | **0.8586** | 0.4631 | **0.8599** | 0.4653 | **0.8508** | *0.4700* |
| VT+NN | 0.7951 | 0.0366 | 0.7951 | 0.0366 | 0.7951 | 0.0366 |
| SAND+NN | 0.8080 | 0.2885 | 0.8080 | 0.2885 | 0.8080 | 0.2885 |
| MLP | 0.8379 | *0.4681* | 0.8482 | 0.4293 | 0.8340 | 0.3725 |
| ResNet | 0.8470 | 0.3516 | 0.8379 | 0.3590 | 0.8379 | 0.3961 |
| AdaFNN | 0.7406 | 0.1736 | 0.7406 | 0.1736 | 0.7406 | 0.1736 |
| NODE | *0.8495* | 0.3763 | 0.8431 | 0.3388 | 0.8379 | 0.3781 |
| CatBoost | 0.8521 | 0.3936 | *0.8534* | *0.4744* | *0.8457* | 0.4516 |
| XGBoost | *0.8495* | 0.4579 | 0.8444 | 0.4643 | 0.8353 | 0.4356 |
| LightGBM | 0.8482 | 0.4179 | *0.8521* | *0.4722* | 0.8392 | 0.4414 |
| AutoGluon | 0.8470 | 0.4327 | 0.8508 | 0.4279 | 0.8288 | 0.4211 |
| xRFM | 0.8145 | 0.3129 | 0.8145 | 0.3129 | 0.8145 | 0.3129 |
| IDAT | 0.8171 | 0.3092 | 0.8171 | 0.3092 | 0.8171 | 0.3092 |
| IDAT w/o $A_I$ | 0.8171 | 0.3092 | 0.8171 | 0.3092 | 0.8171 | 0.3092 |

To accommodate irregular sampling to a Transformer model, we fix the ordered grids $\tilde{\tau} = \{\tau_j\}_{j=1}^T \subset [0,1]$. In the sparse regime—where each trajectory contributes only a few time-points relative to $T$—one may take $\{\tau_j\}$ to be the union of all distinct observation times (optionally augmented by equispaced points to guarantee full coverage). In the dense regime it is customary to partition $[0,1]$ into $T$ bins, and aggregate local measurements via averaging or interpolation. We define the mesh size $\Delta = \max_{1 \le j < T} |\tau_{j+1} - \tau_j|$ and $\rho = \min_{1 \le j < T} |\tau_{j+1} - \tau_j|$, we will carefully show how the discretization incurs an approximation error of order $\Delta^\alpha$ relative to the true continuous trajectory in Lemma 3.

In the training phase, we align each subject's irregular observations to the fixed grid $\tilde{\tau} = (\tau_1, \ldots, \tau_T)$ by introducing a binary mask $M_i(\tilde{\tau}) \in \{0,1\}^T$ and a zero-padded trajectory vector $X_i^*(\tilde{\tau}) \in \mathbb{R}^T$. Concretely, for each $j = 1, \ldots, T$ we define

$$M_i(\tau_j) = \begin{cases} 1, & \exists\, k \text{ such that } \tau_j = t_{i,k}, \\ 0, & \text{otherwise}, \end{cases} \qquad X_i^*(\tau_j) = \begin{cases} X_i^*(t_{i,k}), & \tau_j = t_{i,k}, \\ 0, & M_i(\tau_j) = 0. \end{cases}$$

Thus the elementwise (Hadamard) product $X_i(\tilde{\tau}) \odot M_i(\tilde{\tau})$ retains the observed values and zeros out unsampled entries. Finally, we concatenate this length-$T$ vector with the scalar response $Y_i$ to obtain

$$D_i = \big( X_i^*(\tilde{\tau}) \odot M_i(\tilde{\tau}),\ Y_i \big) \in \mathbb{R}^{T+1},$$

which is then passed to the embedding along with the positional information. In order for the model to make use of the order of the time sequence, we must inject some information about the relative or absolute position of the observation in the time trajectory. We apply two learned linear maps with embedding dimension $d$,

$$E_X : \mathbb{R} \to \mathbb{R}^d, \qquad E_Y : \mathbb{R} \to \mathbb{R}^d,$$

pointwise across the sequence and then add sinusoidal positional encodings $P(\tilde{\tau})$.

$$\tilde{D}_i = \big(E_X \cdot [X_i^*(\tilde{\tau}) \odot M_i(\tilde{\tau})] + P(\tilde{\tau}),\ E_Y \cdot Y_i\big) \in \mathbb{R}^{d\times(T+1)},$$

for dimension $i \in [d]$ and $j \in [T]$, we define the relative position as $p = \sum_{k \leq j} \tau_k/\rho$, then

$$P^{(2i)}(\tau_j) = \sin(p \cdot 10000^{-2i/d}),\ P^{(2i+1)}(\tau_j) = \cos(p \cdot 10000^{-2i/d}).$$

Intuitively, the geometric progression of wavelengths ensures each timepoint is mapped to a unique location in $\mathbb{R}^d$, which underpins the universal approximation property over sequences of bounded length.

Consider a mini-batch embedding of size $B$, we define $\tilde{\mathbf{D}} = \{\tilde{D}_i\}_{i=1}^B \in \mathbb{R}^{B\times d\times(T+1)}$ as a training batch. The dual-attention Transformer block $\mathcal{TB} : \mathbb{R}^{B\times d\times(T+1)} \to \mathbb{R}^{B\times d\times(T+1)}$ is defined by

$$\mathcal{TB}(\tilde{\mathbf{D}}) = \mathrm{FF}_1 \circ A_I \circ \mathrm{FF}_2 \circ A_T(\tilde{\mathbf{D}}) = \mathrm{FF}_1\Big(A_I\big(\mathrm{FF}_2(A_T(\tilde{\mathbf{D}}))\big)\Big) \in \mathbb{R}^{B\times d\times(T+1)}, \qquad (10)$$

where $\mathrm{FF}_1, \mathrm{FF}_2 : \mathbb{R}^{B\times d\times(T+1)} \to \mathbb{R}^{B\times d\times(T+1)}$ position-wise feed-forward network (e.g. a two-layer ReLU MLP with hidden dimension $d_{\mathrm{FF}}$). WLOG, we assume $\mathrm{FF}_1 = \mathrm{FF}_2 = \mathrm{FF}$. For analytical simplicity we omit layer normalization, while preserving the basic architecture of the Transformer.

We next introduce the dual-attention mechanism, which combines classical self-attention over time points with an inter-sample attention across the batch. Let $\sigma[\cdot]$ denote the column-wise softmax operator, which maps any real matrix to a column-stochastic matrix (nonnegative entries, each column summing to one). Both attention modules employ $H$ distinct heads of size $d_A$. For each head $h$, the time-point projections satisfy $W_Q^h, W_K^h, W_V^h \in \mathbb{R}^{d_A\times d}$ and $W_O^h \in \mathbb{R}^{d\times d_A}$, while the inter-sample projections satisfy $V_Q^h, V_K^h, V_V^h \in \mathbb{R}^{d\times d_A}$ and $V_O^h \in \mathbb{R}^{d_A\times d}$. This design follows Yun et al. (2019), augmented by inter-sample attention to borrow strength across subjects.

**Time-Point Attention.** Applied independently to each sample $i = 1, \dots, B$. Let $\tilde{D}_i \in \mathbb{R}^{d\times(T+1)}$ be the input embedding of the $i$th subject. For head $h$ we form

$$Q_i^h = W_Q^h\,\tilde{D}_i, \quad K_i^h = W_K^h\,\tilde{D}_i, \quad V_i^h = W_V^h\,\tilde{D}_i, \quad W_Q^h, W_K^h, W_V^h \in \mathbb{R}^{d_A\times d}.$$

We then compute the attention weights

$$S_i^h = \sigma\big[(K_i^h)^\top Q_i^h\big] \in \mathbb{R}^{(T+1)\times(T+1)},$$

and update the sequence by

$$A_T(\tilde{D}_i) = \tilde{D}_i + \sum_{h=1}^H W_O^h\big(V_i^h S_i^h\big) \in \mathbb{R}^{d\times(T+1)}, \quad W_O^h \in \mathbb{R}^{d\times d_A}.$$

**Inter-Sample Attention.** Operates across the batch at each time index $t = 1, \dots, T+1$. Denote $\tilde{D}^t = \tilde{\mathbf{D}}_{:,:,t} \in \mathbb{R}^{B\times d}$. For head $h$ set

$$Q_t^h = \tilde{D}^t U_Q^h, \quad K_t^h = \tilde{D}^t U_K^h, \quad V_t^h = \tilde{D}^t U_V^h, \quad U_Q^h, U_K^h, U_V^h \in \mathbb{R}^{d\times d_A}.$$

The batch-wise attention weights are

$$S_t^h = \sigma\big[Q_t^h\,(K_t^h)^\top\big] \in \mathbb{R}^{B\times B},$$

and the updated features

$$A_I(\tilde{D}^t) = \tilde{D}^t + \sum_{h=1}^H \big(S_t^h V_t^h\big) U_O^h \in \mathbb{R}^{B\times d}, \quad V_O^h \in \mathbb{R}^{d_A\times d}.$$

Reassembling over all $t$ yields $A_I(\tilde{\mathbf{D}}) \in \mathbb{R}^{B\times d\times(T+1)}$.

Composing $L$ Transformer blocks defines the full dual-attention Transformer embedding

$$\mathcal{T}(\tilde{\mathbf{D}}) = \underbrace{\mathcal{TB} \circ \cdots \circ \mathcal{TB}}_{L\ \text{times}}(\tilde{\mathbf{D}}) = \mathcal{TB}^{\circ L}(\tilde{\mathbf{D}}) : \mathbb{R}^{B\times d\times(T+1)} \to \mathbb{R}^{B\times d\times(T+1)}. \qquad (11)$$

**Regressor Layer.** After learning an embedding from the dual-attention mechanism, let the dual–attention encoder output

$$\mathcal{T}(\tilde{D}_i) \in \mathbb{R}^{d \times (T+1)}, \qquad \mathcal{T}(\tilde{D}_i) = \begin{bmatrix} Z_X \mid Z_Y \end{bmatrix}_i,$$

where $Z_X = \mathcal{T}(\tilde{D}_i)_{:,:,1:T} \in \mathbb{R}^{d \times T}$ contains the token-wise representations of the longitudinal covariates and $Z_Y = \mathcal{T}(\tilde{D}_i)_{:,:,T+1} \in \mathbb{R}^d$ is the embedding of the response token. Define $\phi \colon \mathbb{R}^{d \times T} \to \mathbb{R}^d$ is any deterministic pooling operator that aggregates the first length-$T$ sequence of $d$-dimensional vectors from $\mathcal{T}(\tilde{D}_i)$ into a single $d$-dimensional summary. Common choices include mean pooling or attention pooling, which is designed to learn the nonparametric regression weight between the embedding. In any case $\phi$ is a fixed, deterministic function—once its parameters are trained, it introduces no additional randomness at inference. We assume $\phi$ is Lipschitz, with constant $L_\phi$, so that for any two sequences $Z, Z'$,

$$\|\phi(Z) - \phi(Z')\|_\infty \leq L_\phi \|Z - Z'\|_\infty.$$

We take $g \colon \mathbb{R}^d \to \mathbb{R}$ to be a two-layer ReLU network of the form

$$g(z) = W^{(2)} \sigma\big(W^{(1)} z + b^{(1)}\big) + b^{(2)},$$

where

$$W^{(1)} \in \mathbb{R}^{d_{\text{FF}} \times d}, \quad b^{(1)} \in \mathbb{R}^{d_{\text{FF}}}, \quad W^{(2)} \in \mathbb{R}^{1 \times d_{\text{FF}}}, \quad b^{(2)} \in \mathbb{R}.$$

Here $\sigma(x) = \max\{0, x\}$ acts element-wise. This network is Lipschitz in the sup-norm with constant

$$L_g = \|W^{(2)}\|_{\text{op}} \|W^{(1)}\|_{\text{op}},$$

since each linear map has operator norm equal to its largest singular value and ReLU is 1-Lipschitz. The model prediction is therefore $\hat{Y}_i = g\big(\phi([Z_X]_i)\big)$. During training we minimise a loss $\ell\big(\hat{Y}_i, g([Z_Y]_i)\big)$, treating the response embedding $Z_Y$ as an informative target. We train $g$ so that it uniformly approximates (Hornik, 1991; Stinchcombe, 1999; Cybenko, 1989; Hornik et al., 1989; Yarotsky, 2017) the true functional on the oracle embedding range; that is, for $z = \phi([Z_X]_i)$,

$$\sup_{\|z\|_\infty \leq R} \big|g(z) - f(s_i)\big| \leq \varepsilon_g.$$

### A.4.1 Overview

The theoretical development proceeds as follows. We first embed irregular, noisy longitudinal measurements on a fixed grid and control the resulting input–embedding discrepancy (Lemma 3). In training phase, the components of the dual–attention block (time–point self–attention $A_T$, position–wise feed–forward FF, and inter–sample attention $A_I$) are Lipschitz (Lemmas 4, 8) and admit uniform approximation on compact sets (Lemma 5,6,7), yielding a deterministic *approximation bias*. Inter–sample attention further reduces stochastic embedding variance by up to a $B^{-1/2}$ factor (Lemma 9). Stacking $L$ blocks yields the training phase embedding error $\varepsilon_{\mathcal{T}}$ (Theorem 1). *Generalization bounds* for a single block and for $L$ stacked blocks are given in Lemma 10 and 11, and the train MSE deviation appears in Theorem 2. Combining these with the approximation bounds from Theorem 1 shows that the training MSE is controlled by (i) input discretization, (ii) model approximation and variance terms, and (iii) a Rademacher generalization term. Under standard structural risk minimization scaling: refining the grid so $\Delta \to 0$, increasing capacity so $\varepsilon_{\text{FF}}, \varepsilon_{A_T}, \varepsilon_{A_I}, \varepsilon_g \to 0$, enforcing norm control so the encoder Lipschitz $L_{\mathcal{T}}$ remains bounded and letting $p = B d (T+1)$ grow with $n$ so that $\sqrt{p/n} \to 0$ (or an analogous spectral complexity term $\to 0$), the generalization term vanishes and the training MSE is consistent.

During the testing phase, Theorem 13 bounds pointwise and uniform test prediction error via a decomposition into embedding error, regressor approximation error, discretization bias, and label noise. It then controls the population test MSE via the training MSE plus a Rademacher complexity term and an expectation bridge that propagates the test-phase perturbation from the missing $Y$-token; the empirical test MSE further adds a standard test concentration term. As shown in Corollary 14, the test MSE is consistent if: (i) the mesh shrinks $\Delta \to 0$, (ii) the training MSE is consistent, (iii) the bridge term vanishes with $L_Y \to 0$ or $Y$-token masking, and (iv) $N \to \infty$ so the test concentration term vanishes; under these structural risk minimization-style conditions, both population and empirical test MSE converge to the Bayes risk.

### A.4.2 TRAINING PHASE.

**Lemma 3** (Input Embedding Discretization Error). *Under the random effect model with assumption that $\mu, b_i \in C^\alpha([0,1]; \mathbb{R})$ with Hölder constant $L/2$, $\sup_{t \in [0,1]} |\mu(t)| = \|\mu\|_\infty$, and $b_i \sim \mathcal{GP}(0, B)$ with $\sigma_b^2 = \sup_{t \in [0,1]} B(t,t)$. The measurement noise satisfies $\eta_i(\tau_j) \overset{\text{i.i.d.}}{\sim} N(0, \sigma_X^2)$, and the label noise $\epsilon_i \overset{\text{i.i.d.}}{\sim} N(0, \sigma_Y^2)$. Let $\tilde{D}_i = \big(E_X\big(X_i^*(\tilde{\tau}) \odot M_i(\tilde{\tau})\big) + P(\tilde{\tau}), E_Y Y_i\big)$ and $\tilde{S}_i = \big(E_X\big(s_i(\tilde{\tau})\big) + P(\tilde{\tau}), E_Y f(s_i)\big)$, where $s_i(\tau_j) = \mu(\tau_j) + b_i(\tau_j)$. Then with probability at least $1 - \delta$,*

$$\delta_0 = \|\tilde{\mathbf{D}} - \tilde{\mathbf{S}}\|_\infty$$

$$\leq (L_X + L_Y L_f)\Big(L\,\Delta^\alpha + \sigma_X \sqrt{2\ln\tfrac{2BT}{\delta}} + \|\mu\|_\infty + \sigma_b\sqrt{2\ln\tfrac{2BT}{\delta}}\Big) + L_Y \sigma_Y \sqrt{2\ln\frac{2B}{\delta}},$$

*where $\Delta = \max_j |\tau_{j+1} - \tau_j|$ is the mesh size, $L_X = \|E_X\|_{\text{op}}$ and $L_Y = \|E_Y\|_{\text{op}}$.*

**Remark 1:** The true response is generated by the functional

$$Y_i = F\big(X_i(\cdot)\big) + \epsilon_i, \ \mathcal{F} : C^\alpha([0,1]; \mathbb{R}) \to \mathbb{R}, \ \|\epsilon_i\| \sim N(0, \sigma_Y^2).$$

We never observe $X_i$ continuously, only at grid points $\{\tau_j\}_{j=1}^T$. Write $s_i(\tau_j) = \mu(\tau_j) + b_i(\tau_j)$, and extend $s_i$ to a continuous function $\tilde{s}_i \in C^\alpha([0,1]; \mathbb{R})$ (e.g. by linear interpolation). We then define the oracle scalar

$$f(s_i) := \mathcal{F}(\tilde{s}_i).$$

Since $\mathcal{F}$ is $L_f$–Lipschitz in the supremum norm, with mesh size $\Delta$ we have

$$\big|\mathcal{F}(X_i(\cdot)) - f(s_i)\big| = \big|\mathcal{F}(X_i(\cdot)) - \mathcal{F}(\tilde{s}_i)\big| \leq L_f\|X_i - \tilde{s}_i\|_\infty \leq L_f\,\Delta^\alpha.$$

Thus $f(s_i)$ approximates the true functional $\mathcal{F}(X_i(\cdot))$ with a discretization error of order $\Delta^\alpha$.

**Remark 2:** In the sequel, we establish high-probability, non-asymptotic bounds showing that our learned Transformer embedding $\mathcal{T}(\tilde{D})$ approximates the oracle embedding $H(\tilde{S})$ up to some error terms. One term is regarding to discretization error of order $\Delta^\alpha$, arising from sampling an $\alpha$–Hölder trajectory on a grid of mesh size $\Delta$. While reducing $\Delta$ tightens the discretization bound, it also increases the sequence length $T$ and hence computational cost. In practical applications, $\Delta$ must therefore be chosen to balance statistical accuracy against the available computational budget.

*Proof.* Fix $i \in [B]$ and $j \in [T]$. If $M_i(\tau_j) = 1$, then $X_i^*(\tau_j) = s_i(t_{ik}) + \eta_i(t_{ik}) = \mu(t_{ik}) + b_i(t_{ik}) + \eta_i(t_{ik})$ for some $t_{ik}$ with $|t_{ik} - \tau_j| \leq \Delta$. Hence by continuity assumption on $\mu$ and $b_i$,

$$\big|X_i^*(\tau_j) - s_i(\tau_j)\big| \leq \big|s_i(t_{ik}) - s_i(\tau_j)\big| + |\eta_i(\tau_j)| \leq L\,\Delta^\alpha + |\eta_i(\tau_j)|.$$

If $M_i(\tau_j) = 0$ then $X_i^*(\tau_j) \odot M_i(\tau_j) = 0$ and $\big|0 - s_i(\tau_j)\big| = |s_i(\tau_j)|$. Therefore for every $i, j$,

$$\big|X_i^*(\tau_j)M_i(\tau_j) - s_i(\tau_j)\big| \leq |s_i(\tau_j)| \leq L\,\Delta^\alpha + |\eta_i(\tau_j)| + |s_i(\tau_j)|.$$

By standard Gaussian-maxima bounds and a union bound over $i$ and $j$,

$$\max_{i,j} |\eta_i(\tau_j)| \leq \sigma_X \sqrt{2\ln\frac{2BT}{\delta}}, \quad \max_{i,j} |b_i(\tau_j)| \leq \sigma_b \sqrt{2\ln\frac{2BT}{\delta}},$$

each with probability $\geq 1 - \delta/2$. Hence with probability $\geq 1 - \delta$,

$$\max_{i,j}\big|X_i^*(\tau_j)M_i(\tau_j) - s_i(\tau_j)\big| \leq L\,\Delta^\alpha + \sigma_X\sqrt{2\ln\frac{2BT}{\delta}} + \|\mu\|_\infty + \sigma_b\sqrt{2\ln\frac{2BT}{\delta}}.$$

Applying $E_X$ (operator-norm $L_X$) to each token and noting that the positional encoding $P(\tau_j)$ cancels, we get

$$\max_{i,j}\big\|E_X(X_i^* M_i) - E_X(s_i)\big\|_2 \leq L_X\Big(L\,\Delta^\alpha + \sigma_X\sqrt{2\ln\frac{2BT}{\delta}} + \|\mu\|_\infty + \sigma_b\sqrt{2\ln\frac{2BT}{\delta}}\Big).$$

Finally, for the $Y$-token, since $f$ is $L_f$–Lipschitz in the sup-norm,

$$\big|Y_i - f(s_i)\big| = \big|f(X_i) - f(s_i)\big| + |\epsilon_i| \leq L_f \,\|X_i - s_i\|_\infty + |\epsilon_i|.$$

Then with probability at least $1 - \frac{\delta}{2}$, $\max_{i,j}\big|X_i(\tau_j) M_i(\tau_j) - s_i(\tau_j)\big|$ yields the same upper bound as $L\,\Delta^\alpha + \sigma_X\sqrt{2\ln\frac{2BT}{\delta}} + \|\mu\|_\infty + \sigma_b\sqrt{2\ln\frac{2BT}{\delta}}$. A Gaussian-maxima bound and union bound over $i = 1, \ldots, B$ give $\max_{1 \leq i \leq B} |\epsilon_i| \leq \sigma_Y\sqrt{2\ln\frac{2B}{\delta}}$ with probability $\geq 1 - \frac{\delta}{2}$. Hence with probability at least $1 - \delta$,

$$\max_{1 \leq i \leq B}\big|Y_i - f(s_i)\big| \leq L_f\Big(L\,\Delta^\alpha + \sigma_X\sqrt{2\ln\frac{2BT}{\delta}} + \|\mu\|_\infty + \sigma_b\sqrt{2\ln\frac{2BT}{\delta}}\Big) + \sigma_Y\sqrt{2\ln\frac{2B}{\delta}}.$$

Finally, applying the linear embedding $E_Y$ (with operator norm $L_Y$) to each scalar yields

$$\max_{1 \leq i \leq B}\big\|E_Y(Y_i) - E_Y\big(f(s_i)\big)\big\|_2$$

$$\leq L_Y\Big(L_f\big(L\,\Delta^\alpha + \sigma_X\sqrt{2\ln\frac{2BT}{\delta}} + \|\mu\|_\infty + \sigma_b\sqrt{2\ln\frac{2BT}{\delta}}\big) + \sigma_Y\sqrt{2\ln\frac{2B}{\delta}}\Big).$$

Combining these bounds yields the stated result. $\qquad\square$

**Lemma 4** (Lipschitz Continuity of the Feed-Forward Network)**.** *Let*

$$\mathrm{FF} : \mathbb{R}^{B \times d \times (T+1)} \;\longrightarrow\; \mathbb{R}^{B \times d \times (T+1)}$$

*be the position-wise two-layer ReLU network defined by, for each sample $i \in [B]$ and time-index $j \in [T+1]$,*

$$\mathrm{FF}(Z)_{i,j} = W_2\Big[\max\{0,\, W_1 Z_{i,j}\}\Big] + b_2, \quad W_1 \in \mathbb{R}^{d_{\mathrm{FF}} \times d},\; W_2 \in \mathbb{R}^{d \times d_{\mathrm{FF}}},\; b_2 \in \mathbb{R}^d,$$

*where the ReLU ($\max\{0,x\}$) acts element-wise on $x \in \mathbb{R}^{d_{\mathrm{FF}}}$, and $Z_{i,j} \in \mathbb{R}^d$ denotes the embedding of the $j$th token of the $i$th sample. Then $\mathrm{FF}$ is Lipschitz continuous in both the sup-norm and any $L_p$ norm, with constant*

$$L_{\mathrm{FF}} = \|W_2\|_{\mathrm{op}} \,\|W_1\|_{\mathrm{op}},$$

*where the operator norm of a matrix $W \in \mathbb{R}^{m \times n}$ is $\|W\|_{\mathrm{op}} = \sup_{x \in \mathbb{R}^n \setminus \{0\}} \|W x\|_2 / \|x\|_2$.*

*Proof.* Observe that: $x \mapsto W_1 x$ is $\|W_1\|_{\mathrm{op}}$-Lipschitz in $\ell_2$. The ReLU activation $x \mapsto \max\{0,x\}$ is 1-Lipschitz in $\ell_2$. $x \mapsto W_2 x + b_2$ is $\|W_2\|_{\mathrm{op}}$-Lipschitz in $\ell_2$. Composing these shows each token map $Z_{i,j} \mapsto \mathrm{FF}(Z)_{i,j}$ is $L_{\mathrm{FF}}$-Lipschitz. Applying this position-wise yields, for any $Z, Z' \in \mathbb{R}^{B \times d \times (T+1)}$, with $\|Z\|_\infty = \max_{i,j} \|Z_{i,j}\|_2$,

$$\|\mathrm{FF}(Z) - \mathrm{FF}(Z')\|_\infty \leq L_{\mathrm{FF}} \,\|Z - Z'\|_\infty.$$

$\qquad\square$

**Lemma 5** (Approximation Power of the Feed-Forward Network)**.** *Let $R > 0$ and let*

$$G \colon [-R, R]^{B \times d \times (T+1)} \;\to\; \mathbb{R}^{B \times d \times (T+1)}$$

*be any position-wise operator satisfying the Lipschitz condition*

$$\| G(Z) - G(Z')\|_\infty \leq L_G \,\| Z - Z'\|_\infty \quad \forall Z, Z' \text{ with } \|Z\|_\infty, \|Z'\|_\infty \leq R.$$

*Then for every $\varepsilon_{\mathrm{FF}} > 0$ there exists a two-layer ReLU network $\mathrm{FF} : \mathbb{R}^{B \times d \times (T+1)} \to \mathbb{R}^{B \times d \times (T+1)}$ applied position-wise (i.e. independently to each $(i,j)$ token) with hidden width*

$$d_{\mathrm{FF}} = \big\lceil (2\,L_G\,R/\varepsilon_{\mathrm{FF}})^d \big\rceil$$

*such that $\sup_{\|Z\|_\infty \leq R} \big\|\mathrm{FF}(Z) - G(Z)\big\|_\infty \leq \varepsilon_{\mathrm{FF}}$.*

*Proof Sketch.* This lemma follows classical universal approximation results (Hornik, 1991; Stinchcombe, 1999; Cybenko, 1989; Hornik et al., 1989; Yarotsky, 2017). Since $G$ acts independently on each $d$-dimensional token $Z_{i,j} \in [-R, R]^d$ and is $L_G$-Lipschitz in the sup-norm, the classical two-layer ReLU universal approximation construction on $[-R, R]^d$ yields a network $h : \mathbb{R}^d \to \mathbb{R}^d$ of width $\lceil (2L_G R/\varepsilon_{\mathrm{FF}})^d \rceil$ satisfying $\sup_{\|x\|_\infty \leq R} \|h(x) - G(Z)_{i,j}\|_\infty \leq \varepsilon_{\mathrm{FF}}$. Applying $h$ to each token position-wise produces the desired FF. $\qquad\square$

**Lemma 6** (Uniform Approximation by Multi-Head Time-point Attention)**.** *Let* $X = \{U \in \mathbb{R}^{d \times (T+1)} : \|U\|_\infty \leq R\}$ *be a compact subset in the sup–norm, and let*

$$f_{\mathrm{T}} : X \longrightarrow \mathbb{R}^{d \times (T+1)}$$

*be any $L_{\mathrm{temp}}$-Lipschitz continuous mapping. Then for every $\varepsilon_{\mathrm{A_T}} > 0$ there exist $H \geq H_0(\varepsilon)$ and $d_A \geq d_0(\varepsilon)$, and weight matrices $\{W_Q^h, W_K^h, W_V^h \in \mathbb{R}^{d_A \times d}, W_O^h \in \mathbb{R}^{d \times d_A}\}_{h=1}^H$ such that the multi-head time-point attention operator satisfies*

$$\sup_{U \in X} \|A_T(U) - f_{\mathrm{T}}(U)\|_\infty \leq \varepsilon_{\mathrm{A_T}}.$$

*Remark: In particular, if $\tilde{\mathbf{D}}, \tilde{\mathbf{S}} \in \mathbb{R}^{B \times d \times (T+1)}$ satisfy $\|\tilde{D}_i\|_\infty \leq R$ and $\|\tilde{S}_i\|_\infty \leq R$ for all $i$, then with $\|\tilde{\mathbf{D}} - \tilde{\mathbf{S}}\|_\infty$ bounded in Lemma 3,*

$$\|A_T(\tilde{\mathbf{D}}) - f_{\mathrm{T}}(\tilde{\mathbf{S}})\|_\infty \leq \varepsilon_{\mathrm{A_T}} + L_{\mathrm{T}} \|\tilde{\mathbf{D}} - \tilde{\mathbf{S}}\|_\infty = \varepsilon_{\mathrm{A_T}} + L_{\mathrm{T}} \delta_0.$$

*This is the total embeddind error if we only apply time-point attention mechanism to get embedding.*

*Proof Sketch.* The time-point attention is identical to the classical multi-head self-attention. The time-point attention operator $A_T$ is realized by first projecting the input $U$ via the matrices $W_Q^h, W_K^h, W_V^h$, then computing the scaled dot-products $(K^h)^\top Q^h$ and normalizing each row by a softmax to obtain $S^h$. These attention weights are used to re-weight the values $V^h$, the results are linearly combined by $W_O^h$, and finally a residual connection adds the original $U$. Each of these steps—linear projection, softmax-normalization, weighted summation, and residual addition—is a continuous map on the compact domain $X$. By the universal-approximation theorem for Transformer self-attention (Yun et al., 2019; Takeshita & Imaizumi, 2025; Kajitsuka & Sato, 2023), networks of this form with sufficiently many heads $H$ and head-dimension $d_A$ can uniformly approximate any continuous target mapping $f_{\mathrm{T}}$ on $X$ to within $\varepsilon$. The residual connection preserves this approximation while enhancing expressivity, yielding the claimed bound. The final residual ensures exact interpolation of the noiseless input when $\tilde{\mathbf{D}} = \tilde{\mathbf{S}}$, and the Lipschitz continuity of $f_{\mathrm{T}}$ propagates any input mismatch $\|\tilde{\mathbf{D}} - \tilde{\mathbf{S}}\|_\infty$ to an additional error of at most $L_{\mathrm{T}}\|\tilde{\mathbf{D}} - \tilde{\mathbf{S}}\|_\infty$, yielding the stated bound. $\qquad\square$

**Lemma 7** (Uniform Approximation by Multi-Head Inter-Sample Attention)**.** *Let*

$$Y = \{ Z \in \mathbb{R}^{B \times d \times (T+1)} : \|Z\|_\infty \leq R \}$$

*be the closed sup-norm ball of radius $R$, and let*

$$f_{\mathrm{I}} : Y \longrightarrow \mathbb{R}^{B \times d \times (T+1)}$$

*be any $L_{\mathrm{I}}$–Lipschitz continuous mapping across the batch dimension. Then for every $\varepsilon_{A_I} > 0$ there exist integers $H$ and $d_A$ (depending on $\varepsilon, R, B, d, T, L_{\mathrm{I}}$) and weight matrices*

$$\{ V_Q^h, V_K^h, V_V^h \in \mathbb{R}^{d \times d_A}, V_O^h \in \mathbb{R}^{d_A \times d} \}_{h=1}^H$$

*such that the multi-head inter-sample attention operator satisfies*

$$\sup_{Z \in Y} \|A_I(Z) - f_{\mathrm{I}}(Z)\|_\infty \leq \varepsilon_{A_I}.$$

*Remark: Consequently, if $\tilde{\mathbf{D}}, \tilde{\mathbf{S}} \in \mathbb{R}^{B \times d \times (T+1)}$ satisfy $\|\tilde{D}_i\|_\infty \leq R$ and $\|\tilde{S}_i\|_\infty \leq R$ for all $i$, then with $\|\tilde{\mathbf{D}} - \tilde{\mathbf{S}}\|_\infty$ bounded in Lemma 3, then*

$$\|A_I(\tilde{\mathbf{D}}) - f_{\mathrm{I}}(\tilde{\mathbf{S}})\|_\infty \leq \varepsilon_{A_I} + L_{\mathrm{I}} \|\tilde{\mathbf{D}} - \tilde{\mathbf{S}}\|_\infty = \varepsilon.$$

*This is the total embedding error if we ignore the time-point embedding.*

*Proof Sketch.* Inter-sample attention at each time-step $t$ is exactly self-attention over the "batch-axis" vectors $Z_{:,t,:} \in \mathbb{R}^{B \times d}$. As in Lemma 6, one shows that a finite number of heads $H$ and head-dimension $d_A$ suffice to uniformly approximate any continuous, Lipschitz mapping $f_I$ on the compact set $Y$. The argument parallels that for time-point attention: each step (linear projections, scaled-dot-product plus softmax, weighted summation, residual addition) is continuous, so by the universal-approximation property of multi-head self-attention (Yun et al., 2019) one can achieve error at most $\varepsilon$. The final residual ensures exact interpolation of the noiseless input when $\tilde{\mathbf{D}} = \tilde{\mathbf{S}}$, and the Lipschitz continuity of $f_I$ propagates any input mismatch $\|\tilde{\mathbf{D}} - \tilde{\mathbf{S}}\|_\infty$ to an additional error of at most $L_I \|\tilde{\mathbf{D}} - \tilde{\mathbf{S}}\|_\infty$, yielding the stated bound. $\square$

**Lemma 8** (Lipschitz Continuity of Dual-Attention Transformer Block). *Let*

$$\mathcal{TB} = \mathrm{FF} \circ A_I \circ \mathrm{FF} \circ A_T \; : \; \mathbb{R}^{B \times d \times (T+1)} \; \to \; \mathbb{R}^{B \times d \times (T+1)}$$

*be the dual-attention Transformer block. Suppose (1) The position-wise MLP* $\mathrm{FF}$ *is* $L_{\mathrm{FF}}$*–Lipschitz in the sup-norm. (2) The time-point attention* $A_T$ *is* $L_T$*–Lipschitz in the sup-norm. (3) The inter-sample attention* $A_I$ *is* $L_I$*–Lipschitz in the sup-norm. Then* $\mathcal{TB}$ *is Lipschitz continuous in the sup-norm with constant* $L_{\mathcal{TB}} = L_{\mathrm{FF}}^2 L_T L_I$, *i.e. for all* $X, X'$, $\left\| \mathcal{TB}(X) - \mathcal{TB}(X') \right\|_\infty \leq L_{\mathrm{FF}}^2 L_T L_I \left\| X - X' \right\|_\infty$. *Note that the Transformer of layer* $L$ *is naturally* $L_{\mathcal{TB}}^L$*–Lipschitz.*

*Proof.* By assumption $\mathrm{FF}$, $A_T$, and $A_I$ satisfy $\|\mathrm{FF}(U) - \mathrm{FF}(U')\|_\infty \leq L_{\mathrm{FF}} \|U - U'\|_\infty$, $\|A_T(V) - A_T(V')\|_\infty \leq L_T \|V - V'\|_\infty$, $\|A_I(W) - A_I(W')\|_\infty \leq L_I \|W - W'\|_\infty$. Now set $U = A_T(X)$, $U' = A_T(X')$, $V = \mathrm{FF}(U)$, $V' = \mathrm{FF}(U')$, $W = A_I(V)$, $W' = A_I(V')$. Then for any $X, X' \in \mathbb{R}^{B \times d \times (T+1)}$,

$$
\begin{aligned}
\|\mathcal{TB}(X) - \mathcal{TB}(X')\|_\infty &= \left\| \mathrm{FF}\big(A_I(\mathrm{FF}(A_T(X)))\big) - \mathrm{FF}\big(A_I(\mathrm{FF}(A_T(X')))\big) \right\|_\infty \\
&\leq L_{\mathrm{FF}} \left\| A_I(V) - A_I(V') \right\|_\infty \\
&\leq L_{\mathrm{FF}} L_I \left\| V - V' \right\|_\infty \\
&= L_{\mathrm{FF}} L_I \left\| \mathrm{FF}(U) - \mathrm{FF}(U') \right\|_\infty \\
&\leq L_{\mathrm{FF}}^2 L_I \left\| U - U' \right\|_\infty \\
&= L_{\mathrm{FF}}^2 L_I \left\| A_T(X) - A_T(X') \right\|_\infty \\
&\leq L_{\mathrm{FF}}^2 L_I L_T \left\| X - X' \right\|_\infty,
\end{aligned}
$$

establishing the stated Lipschitz constant. $\square$

**Lemma 9** (Variance Reduction by Inter-Sample Attention). *Let*

$$Z_i = \mathrm{FF}\big(A_T(\tilde{D}_i)\big), \qquad m_i = \mathrm{FF}\big(A_T(\tilde{S}_i)\big),$$

*be the feed-forward outputs of the time-point attention on the observed embedding* $\tilde{D}_i$ *and the oracle embedding* $\tilde{S}_i$. *For each time-step* $t = 1, \dots, T+1$, *write*

$$Z^t = \big[Z_1^t; \dots; Z_B^t\big] \in \mathbb{R}^{B \times d}, \qquad m^t = \big[m_1^t; \dots; m_B^t\big] \in \mathbb{R}^{B \times d},$$

*where each row decomposes as*

$$Z_k^t = m_k^t + \gamma_k^t, \quad \gamma_{k,j}^t \overset{iid}{\sim} N(0, \sigma_{\mathrm{enc}}^2), \quad k \in \{1, \dots, B\}, \; j \in \{1, \dots, d\},$$

*under the assumption that* $\|m^t\|_\infty \leq M_m$, *the map* $\beta^t$ *is* $L_\beta$*–Lipschitz and the FF network propagates the Gaussian embedding noise from* $A_T(\tilde{D}_i)$ *with Lipschitz constant absorbed into* $\sigma_{\mathrm{enc}}$. *Define the inter-sample attention weights at time* $t$ *by*

$$\beta_{i,k}^t = \frac{\exp\big((Q_i^t \cdot K_k^t)/\sqrt{d_A}\big)}{\sum_{\ell=1}^B \exp\big((Q_i^t \cdot K_\ell^t)/\sqrt{d_A}\big)},$$

*where* $Q^t = Z^t W_Q$, $K^t = Z^t W_K \in \mathbb{R}^{B \times d_A}$. *Then the inter-sample attention output for subject* $i$ *at* $t$ *is* $\big[A_I(Z^t)\big]_{i,:} = \sum_{k=1}^B \beta_{i,k}^t Z_k^t$. *For any* $\delta \in (0,1)$, *with probability at least* $1 - \delta$,

$$\max_{i=1,\dots,B} \big\| A_I(Z^t)_i - A_I(m^t)_i \big\|_\infty \; \leq \; \sigma_{\mathrm{enc}} \max_{i=1,\dots,B} \sqrt{\sum_{k=1}^B (\beta_{i,k}^t)^2} \; \sqrt{2 \ln \frac{2 B d}{\delta}},$$

*which is of order between* $O_{\mathbb{P}}\left(\frac{\sigma_{\text{enc}}}{\sqrt{B}}\sqrt{2\ln\frac{2Bd}{\delta}}\right)$ *and* $O_{\mathbb{P}}\left(\sigma_{\text{enc}}\sqrt{2\ln\frac{2Bd}{\delta}}\right)$.

*Taken consideration across all time-step, we have*

$$\left\|A_I\big(\text{FF}(A_T(\tilde{\mathbf{D}}))\big) - A_I\big(\text{FF}(A_T(\tilde{\mathbf{S}}))\big)\right\|_\infty$$

$$\leq\ \sigma_{\text{enc}}\max_{i\in[B]}\sqrt{\sum_{k=1}^{B}(\beta_{i,k}^t)^2}\ \sqrt{2\ln\frac{2B(T+1)d}{\delta}} + \min\{L_\beta M_m, L_I\}\ \|\text{FF}(A_T(\tilde{\mathbf{D}})) - \text{FF}(A_T(\tilde{\mathbf{S}}))\|_\infty$$

$$=\ \varepsilon_{\text{var}}.$$

*In particular, if the subject-specific effect is absent so that* $\beta_{i,k}^t = 1/B$, *then the bound is*

$$\frac{\sigma_{\text{enc}}}{\sqrt{B}}\ \sqrt{2\ln\frac{2B(T+1)d}{\delta}} + \min\{L_\beta M_m,\ L_I\}\ \|\text{FF}(A_T(\tilde{\mathbf{D}})) - \text{FF}(A_T(\tilde{\mathbf{S}}))\|_\infty.$$

**Remark 1:** *In the general random-effects case, attention implements a non-uniform weighted average. As long as the learned weights* $\beta_{i,k}^t$'s *are reasonably spread over multiple subjects (typically because similar* $h_k^t$ *get grouped), we can obtain a variance reduction of order* $O_{\mathbb{P}}(1/\sqrt{B}) \leq O_{\mathbb{P}}(\sqrt{\sum_{k=1}^B (\beta_{i,k}^t)^2}) \leq O_{\mathbb{P}}(1)$. *In particular, if subject-specific effect is absent, all queries are identical so that* $\beta_{i,k}^t = 1/B$, *then inter-sample attention is equivalent to simple averaging, and we could achieve the* $1/\sqrt{B}$ *variance reduction.*

$$\max_{i,j}\left|\big[A_I(Z^t) - A_I(m^t)\big]_{i,j}\right|\ \leq\ \frac{\sigma_{\text{enc}}}{\sqrt{B}}\ \sqrt{2\ln\frac{2Bd}{\delta}}.$$

*By contrast, without inter-sample attention each coordinate error* $\gamma_{k,j}^t$ *satisfies* $\max_{k,j}|\gamma_{k,j}^t| \leq \sigma_{\text{enc}}\sqrt{2\ln(2Bd/\delta)}$ *with the same probability. Thus inter-sample attention achieves a* $O_{\mathbb{P}}(1/\sqrt{B})$ *reduction in the dominant noise term when the attention weights are uniform.*

**Remark 2:** *(Effective Embedding-Noise Standard Deviation.) Let* $\eta_i(\tau_j) \overset{iid}{\sim} N(0, \sigma_X^2)$ *and* $\epsilon_i \overset{iid}{\sim} N(0, \sigma_Y^2)$. *Define*

$$\tilde{D}_i = \big(E_X(X_i(\tilde{\tau})\odot M_i(\tilde{\tau})) + P(\tilde{\tau}),\ E_Y(Y_i)\big),\quad \tilde{S}_i = \big(E_X(s_i(\tilde{\tau})) + P(\tilde{\tau}),\ E_Y(f(s_i))\big),$$

*with* $s_i(\tau_j) = \mu(\tau_j) + b_i(\tau_j)$. *Write* $L_X = \|E_X\|_{\text{op}}$ *and* $L_Y = \|E_Y\|_{\text{op}}$. *Then after embedding, measurement noise becomes* $E_X\eta_i(\tau_j) \sim N(0, L_X^2\sigma_X^2)$ *and label noise becomes* $E_Y\epsilon_i \sim N(0, L_Y^2\sigma_Y^2)$. *Passing through a* $L_T$-*Lipschitz time-point attention and an* $L_{\text{FF}}$-*Lipschitz feedforward network multiplies each variance by* $(L_T L_{\text{FF}})^2$. *Hence the effective embedding-noise standard deviation is*

$$\sigma_{\text{enc}} = L_{\text{FF}}\,L_T\,\max\{L_X\,\sigma_X,\ L_Y\,\sigma_Y\}.$$

*Proof.* Define the error

$$U_{i,j} = \sum_{k=1}^{B}\beta_{i,k}^t\,\gamma_{k,j}^t.$$

For fixed $i$ and $j$, $D_{i,j}$ is Gaussian with mean zero and variance $\sigma_{\text{enc}}^2\sum_{k=1}^B(\beta_{i,k}^t)^2$. Hence

$$\Pr\big(|D_{i,j}| > u\big)\ \leq\ 2\exp\left(-\frac{u^2}{2\,\sigma_{\text{enc}}^2\sum_k(\beta_{i,k}^t)^2}\right).$$

By a union bound over all $i\in[B]$ and $j\in[d]$,

$$\Pr\left(\max_{i,j}|D_{i,j}| > u\right)\ \leq\ 2\,B\,d\,\exp\left(-\frac{u^2}{2\,\sigma_{\text{enc}}^2\max_i\sum_k(\beta_{i,k}^t)^2}\right).$$

Setting

$$u = \sigma_{\text{enc}} \max_i \sqrt{\sum_k (\beta_{i,k}^t)^2} \sqrt{2 \ln \frac{2\,B\,d}{\delta}}$$

makes the right-hand side $\leq \delta$. This yields the first displayed bound. The uniform-weight case follows by substituting $\beta_{i,k}^t = 1/B$. Finally, for i.i.d. $\gamma_{k,j}^t \sim N(0, \sigma_{\text{enc}}^2)$, a standard Gaussian-maxima bound gives $\max_{k,j} |\gamma_{k,j}^t| \leq \sigma_{\text{enc}} \sqrt{2 \ln(2\,B\,d/\delta)}$, completing the comparison.

For each row $i$,

$$A_I(Z^t)_i - A_I(m^t)_i = \underbrace{\sum_{k=1}^B \beta_{i,k}^t(Z^t)\,\gamma_k^t}_{=:U_i} + \underbrace{\sum_{k=1}^B \big(\beta_{i,k}^t(Z^t) - \beta_{i,k}^t(m^t)\big)\,m_k^t}_{=:V_i}.$$

Uniform bound across all time-steps $U_{i,j}$ gives the additional $(T+1)$ in the first term of $U_i$. Lipschitz continuity of the weight map gives $\|V_i\|_\infty \leq M_m \|\beta^t(Z^t) - \beta^t(m^t)\|_1 \leq M_m\, L_\beta\, \|Z^t - m^t\|_\infty$,

$\square$

**Theorem** (Theorem 1, revisit, Training-phase embedding error and consistency of $\mathcal{T}$). *Let $\tilde{\mathbf{D}} \in \mathbb{R}^{B \times d \times (T+1)}$ be the observed, masked embedding and let $\tilde{\mathbf{S}} \in \mathbb{R}^{B \times d \times (T+1)}$ be the "oracle" embedding obtained from the noiseless, fully-observed trajectories $s_i = \mu(\tilde{\tau}) + b_i(\tilde{\tau})$ and noiseless responses $f(s_i)$. The corresponding fully observed positional embedding would be $\tilde{s}_i = (E_X s_i + P(\tilde{\tau}), E_Y f(s_i))$, with all the masks are defined as 1, implying fully observed. We define $\tilde{\mathbf{S}} = \{\tilde{S}_i\}_{i=1}^B \in \mathbb{R}^{B \times d \times (T+1)}$, then the ideal block-wise mapping by*

$$H(\tilde{\mathbf{S}}) = G \circ f_I \circ G \circ f_T(\tilde{\mathbf{S}}) = G\Big(f_I\big(G(f_T(\tilde{\mathbf{S}}))\big)\Big).$$

*For the approximation errors, by Lemma 5, the position-wise feed-forward network* FF *satisfies*

$$\|\mathrm{FF}(Z) - G(Z)\|_\infty \leq \varepsilon_{\mathrm{FF}} \quad \text{for all } \|Z\|_\infty \leq R.$$

*Lemma 6 guarantees*

$$\|A_T(Z) - f_T(Z)\|_\infty \leq \varepsilon_{A_T} \quad \text{for all } \|Z\|_\infty \leq R,$$

*and Lemma 7 guarantees*

$$\|A_I(Z) - f_I(Z)\|_\infty \leq \varepsilon_{A_I} \quad \text{for all } \|Z\|_\infty \leq R.$$

*Then with the stochastic error we obtained in Lemma 9, we have that for any $\delta \in (0,1)$, with probability at least $1 - \delta$,*

$$\|A_I(\mathrm{FF}(A_T(\tilde{\mathbf{D}}))) - A_I(\mathrm{FF}(A_T(\tilde{\mathbf{S}})))\|_\infty \leq \varepsilon_{\mathrm{var}}.$$

*With the assumption that $G$, $f_T$, and $f_I$ are Lipschitz continuous in the sup-norm with constants $L_{\mathrm{FF}}$, $L_T$, and $L_I$, respectively. Then the a single dual-attention block for any $\delta \in (0,1)$,*

$$\mathcal{TB}(\tilde{\mathbf{D}}) = \mathrm{FF}\big(A_I\big(\mathrm{FF}(A_T(\tilde{\mathbf{D}}))\big)\big),$$

*obeys the embedding error bound with probability at least $1 - \delta$,*

$$\big\|\mathcal{TB}(\tilde{\mathbf{D}}) - H(\tilde{\mathbf{S}})\big\|_\infty \leq L_{\mathrm{FF}}\Big(\varepsilon_{\mathrm{var}} + \varepsilon_{A_I} + L_I\big(\varepsilon_{\mathrm{FF}} + L_{\mathrm{FF}}\,\varepsilon_{A_T}\big)\Big) + \varepsilon_{\mathrm{FF}} = \varepsilon_{\mathcal{TB}}. \qquad (12)$$

*Consequently, with Lipschitz continuity of $\mathcal{TB}$ showed in Lemma 8 composing $L$ blocks yields embedding error of the dual-attention transformer*

$$\big\|\mathcal{T}(\tilde{\mathbf{D}}) - H(\tilde{\mathbf{S}})\big\|_\infty \leq L_{\mathcal{TB}}^L\,(\delta_0 + \varepsilon_{\mathcal{TB}}) = \varepsilon_{\mathcal{T}}, \qquad (13)$$

*with probability at least $1 - \delta$ and input embedding discretization error $\delta_0$ derived in Lemma 3.*

*Proof.* Employ the following decomposition

$$\|\mathcal{TB}(\tilde{\mathbf{D}}) - H(\tilde{\mathbf{S}})\|_\infty = \left\|\mathrm{FF}\big(A_I(\mathrm{FF}(A_T(\tilde{\mathbf{D}})))\big) - G\big(f_I(G(f_T(\tilde{\mathbf{S}})))\big)\right\|_\infty$$

$$\leq \underbrace{\|\mathrm{FF}(A_I(\mathrm{FF}(A_T(\tilde{\mathbf{D}})))) - \mathrm{FF}(A_I(\mathrm{FF}(A_T(\tilde{\mathbf{S}}))))\|_\infty}_{\text{(I) inter-sample embedding stochastic error}}$$

$$+ \underbrace{\|\mathrm{FF}(A_I(\mathrm{FF}(A_T(\tilde{\mathbf{S}})))) - G(A_I(\mathrm{FF}(A_T(\tilde{\mathbf{S}}))))\|_\infty}_{\text{(II) FF bias}}$$

$$+ \underbrace{\|G(A_I(\mathrm{FF}(A_T(\tilde{\mathbf{S}})))) - G\big(f_I(\mathrm{FF}(A_T(\tilde{\mathbf{S}})))\big)\|_\infty}_{\text{(III) inter-sample attention approximation error}}$$

$$+ \underbrace{\|G\big(f_I(\mathrm{FF}(A_T(\tilde{\mathbf{S}})))\big) - G\big(f_I(G(A_T(\tilde{\mathbf{S}})))\big)\|_\infty}_{\text{(IV) FF bias}}$$

$$+ \underbrace{\|G\big(f_I(G(A_T(\tilde{\mathbf{S}})))\big) - G\big(f_I(G(f_T(\tilde{\mathbf{S}})))\big)\|_\infty}_{\text{(V) time-point attention approximation error}}.$$

By Lemma 9 and continuity assumption, $(I) \leq L_{\mathrm{FF}}\varepsilon_{var}$. By Lemma 5, (II) and (IV) can be bounded by $\varepsilon_{\mathrm{FF}}$ and $L_{\mathrm{FF}}L_I\varepsilon_{\mathrm{FF}}$, respectively, with Lipchitz assumption. (III) and (V) can be bounded with Lemma 6 and 7 by $L_{\mathrm{FF}}\varepsilon_{A_I}$ and $L_{\mathrm{FF}}^2 L_I\varepsilon_{A_T}$. Rearranging yields the stated bound. Define the sequence $\delta_k = \|\mathcal{TB}^{\circ k}(\tilde{D}) - H(\tilde{S})\|_\infty$, $k = 0, 1, \ldots, L$. Since $\delta_0 = \|\tilde{D} - H(\tilde{S})\|_\infty$ and $H$ is fixed, we use only the block-level bound and Lipschitz property (Lemma 8):

$$\delta_1 = \|\mathcal{TB}(\tilde{D}) - H(\tilde{S})\|_\infty \leq \varepsilon_{\mathcal{TB}},$$

and for $k \geq 1$,

$$\delta_{k+1} = \|\mathcal{TB}(\mathcal{TB}^{\circ k}(\tilde{D})) - H(\tilde{S})\|_\infty$$
$$\leq \|\mathcal{TB}(\mathcal{TB}^{\circ k}(\tilde{D})) - \mathcal{TB}(H(\tilde{S}))\|_\infty + \|\mathcal{TB}(H(\tilde{S})) - H(\tilde{S})\|_\infty$$
$$\leq L_{\mathcal{TB}}\,\delta_k + \varepsilon_{\mathcal{TB}}.$$

Unrolling this recursion gives

$$\delta_L \leq L_{\mathcal{TB}}^L \delta_0 + \sum_{j=0}^{L-1} L_{\mathcal{TB}}^j \varepsilon_{\mathcal{TB}} = L_{\mathcal{TB}}^L \delta_0 + \frac{L_{\mathcal{TB}}^L - 1}{L_{\mathcal{TB}} - 1}\varepsilon_{\mathcal{TB}} \leq L_{\mathcal{TB}}^L (\delta_0 + \varepsilon_{\mathcal{TB}}).$$

Note that $\delta_0$ is related to the embedding error from discretization and is bounded under the random effect model by Lemma 3. □

**Remark:** (Oracle Approximation by a Dual-Attention Transformer) Theorem 1 directly implies that oracle approximation from dual-attention transformer is achievable. Let $X = \{Z \in \mathbb{R}^{B \times d \times (T+1)} : \|Z\|_\infty \leq R\}$ be compact. Let the oracle operator be

$$H = G \circ f_I \circ G \circ f_T : X \to \mathbb{R}^{B \times d \times (T+1)},$$

where $G$, $f_T$, and $f_I$ are continuous and $L_{\mathrm{FF}}$-, $L_T$-, $L_I$–Lipschitz on $X$, respectively. Then for any $\varepsilon > 0$ there exist: a position-wise two-layer ReLU network FF, a multi-head time-point attention $A_T$, a multi-head inter-sample attention $A_I$, with sufficiently large hidden width and number of heads/head-dimension (as guaranteed by Lemmas 5, 6, and 7), for any fixed depth $L \in \mathbb{N}$ stacks of the dual-attention block $\mathcal{T} = \mathcal{TB}^{\circ L}$ with $\mathcal{TB} = \mathrm{FF} \circ A_I \circ \mathrm{FF} \circ A_T$. $\mathcal{T}$ satisfies the uniform oracle-approximation bound

$$\sup_{\tilde{\mathbf{D}} \in X} \|\mathcal{T}(\tilde{\mathbf{D}}) - H(\tilde{\mathbf{S}})\|_\infty \leq \varepsilon_{\mathcal{T}}.$$

**Remark:** In particular, if the mesh shrinks $\Delta \to 0$ so that $\delta_0 \to 0$, the approximation and stochastic terms vanish $\varepsilon_{\text{FF}}, \varepsilon_{A_T}, \varepsilon_{A_I}, \varepsilon_{\text{var}} \to 0$ with increasing capacity, and $\sup_B L_{\mathcal{TB}}(B) < \infty$ (the block Lipschitz constant remains uniformly bounded as batch size $B$ grows for fixed $L$), then

$$\left\| \mathcal{T}(\tilde{\mathbf{D}}) - H(\tilde{\mathbf{S}}) \right\|_\infty \longrightarrow 0,$$

i.e., the daul-attention Transformer embedding $\mathcal{T}$ is consistent for the oracle mapping $H$.

**Lemma 10** (Generalization error for a dual–attention block). *Let* $\mathcal{TB} = \text{FF} \circ A_I \circ \text{FF} \circ A_T : \mathbb{R}^{B \times d \times (T+1)} \to \mathbb{R}^{B \times d \times (T+1)}$ *be* $L_{\mathcal{TB}}$*–Lipschitz in* $\| \cdot \|_\infty$ *(Lemma 8). Let* $\Psi : \mathbb{R}^{B \times d \times (T+1)} \to [0,1]$ *be* $L_\Psi$*–Lipschitz in* $\| \cdot \|_\infty$*. Assume inputs satisfy* $\|\tilde{D}\|_\infty \le R_{\text{in}}$ *almost surely and set* $p := B\, d\, (T+1)$*. Then the class* $\mathcal{G} := \{\Psi \circ \mathcal{TB}\}$ *obeys*

$$\hat{\mathfrak{R}}_n(\mathcal{G}) := \mathbb{E}_\sigma \left[ \sup_{g \in \mathcal{G}} \frac{1}{n} \sum_{i=1}^n \sigma_i\, g(\tilde{D}_i) \right] \le L_\Psi L_{\mathcal{TB}}\, R_{\text{in}} \sqrt{\frac{2\,p}{n}}.$$

*Consequently, for* $[0,1]$*–valued* $\Psi \circ \mathcal{TB}$*, with probability at least* $1 - \delta$*,*

$$\mathbb{E}[\Psi(\mathcal{TB}(\tilde{D}))] \le \frac{1}{n} \sum_{i=1}^n \Psi(\mathcal{TB}(\tilde{D}_i)) + 2\, L_\Psi L_{\mathcal{TB}}\, R_{\text{in}} \sqrt{\frac{2\,p}{n}} + 3\sqrt{\frac{\ln(2/\delta)}{2n}}.$$

*Proof.* $\Psi \circ \mathcal{TB}$ is $L_\Psi L_{\mathcal{TB}}$–Lipschitz in $\| \cdot \|_\infty$, hence also $L_\Psi L_{\mathcal{TB}}$–Lipschitz in $\| \cdot \|_2$. By Lemma 8, for all $X, X'$, $\|\mathcal{TB}(X) - \mathcal{TB}(X')\|_\infty \le L_{\mathcal{TB}} \|X - X'\|_\infty$. Since $\Psi$ is $L_\Psi$–Lipschitz in $\| \cdot \|_\infty$,

$$|\Psi(\mathcal{TB}(X)) - \Psi(\mathcal{TB}(X'))| \le L_\Psi \|\mathcal{TB}(X) - \mathcal{TB}(X')\|_\infty \le L_\Psi L_{\mathcal{TB}} \|X - X'\|_\infty.$$

Because $\|v\|_\infty \le \|v\|_2$ for all $v \in \mathbb{R}^p$, this also implies $|\Psi(\mathcal{TB}(X)) - \Psi(\mathcal{TB}(X'))| \le L_\Psi L_{\mathcal{TB}} \|X - X'\|_2$; i.e., the same Lipschitz constant works in $\ell_2$ (no dimension factor is introduced when passing from $\ell_\infty$ to $\ell_2$). From $\|\tilde{D}_i\|_\infty \le R_{\text{in}}$ we get $\|\tilde{D}_i\|_2 \le \sqrt{p}\, R_{\text{in}}$ for each $i$. It is a standard fact (Bartlett & Mendelson, 2002; Shalev-Shwartz & Ben-David, 2014) that for an $L$–Lipschitz (in $\ell_2$) function class on a set with $\max_i \|\tilde{D}_i\|_2 \le R_2$,

$$\hat{\mathfrak{R}}_n \le L\, R_2 \sqrt{\frac{2}{n}}.$$

Applying this with $L = L_\Psi L_{\mathcal{TB}}$ and $R_2 = \sqrt{p}\, R_{\text{in}}$ yields $\hat{\mathfrak{R}}_n(\mathcal{G}) \le L_\Psi L_{\mathcal{TB}} \left( \sqrt{p}\, R_{\text{in}} \right) \sqrt{2/n} = L_\Psi L_{\mathcal{TB}}\, R_{\text{in}} \sqrt{2p/n}$. Finally, apply the standard Rademacher generalization inequality for $[0,1]$–valued functions: for all $g \in \mathcal{G}$, with probability at least $1 - \delta$,

$$\mathbb{E}[g(\tilde{D})] \le \frac{1}{n} \sum_{i=1}^n g(\tilde{D}_i) + 2 \hat{\mathfrak{R}}_n(\mathcal{G}) + 3\sqrt{\frac{\ln(2/\delta)}{2n}},$$

which gives the stated bound. $\square$

**Lemma 11** (Generalization error for an $L$-stack dual–attention Transformer). *Let* $\mathcal{TB} = \text{FF} \circ A_I \circ \text{FF} \circ A_T : \mathbb{R}^{B \times d \times (T+1)} \to \mathbb{R}^{B \times d \times (T+1)}$ *be* $L_{\mathcal{TB}}$*–Lipschitz in* $\| \cdot \|_\infty$ *(Lemma 8) and define the $L$-block Transformer* $\mathcal{T} := \mathcal{TB}^{\circ L}$*,* $\mathcal{T}$ *is* $L_{\mathcal{T}}$*–Lipschitz in* $\| \cdot \|_\infty$ *with* $L_{\mathcal{T}} = L_{\mathcal{TB}}^L$*. Let* $\Psi : \mathbb{R}^{B \times d \times (T+1)} \to [0,1]$ *be* $L_\Psi$*–Lipschitz in* $\| \cdot \|_\infty$*. Assume inputs satisfy* $\|\tilde{D}\|_\infty \le R_{\text{in}}$ *almost surely and set* $p := B\, d\, (T+1)$*. Then the class* $\mathcal{G}_L := \{\Psi \circ \mathcal{T}\}$ *obeys*

$$\hat{\mathfrak{R}}_n(\mathcal{G}_L) := \mathbb{E}_\sigma \left[ \sup_{g \in \mathcal{G}_L} \frac{1}{n} \sum_{i=1}^n \sigma_i\, g(\tilde{D}_i) \right] \le L_\Psi L_{\mathcal{T}}\, R_{\text{in}} \sqrt{\frac{2\,p}{n}} = L_\Psi L_{\mathcal{TB}}^L\, R_{\text{in}} \sqrt{\frac{2\,p}{n}}.$$

*Consequently, for* $[0,1]$*–valued* $\Psi \circ \mathcal{T}$*, with probability at least* $1 - \delta$*,*

$$\mathbb{E}[\Psi(\mathcal{T}(\tilde{D}))] \le \frac{1}{n} \sum_{i=1}^n \Psi(\mathcal{T}(\tilde{D}_i)) + 2\, L_\Psi L_{\mathcal{TB}}^L\, R_{\text{in}} \sqrt{\frac{2\,p}{n}} + 3\sqrt{\frac{\ln(2/\delta)}{2n}}.$$

*Proof.* By Lemma 8, $\mathcal{TB}$ is $L_{\mathcal{TB}}$–Lipschitz in $\|\cdot\|_\infty$. Therefore the composition $\mathcal{T} = \mathcal{TB}^{\circ L}$ is $L_\mathcal{T}$–Lipschitz with $L_\mathcal{T} = L_{\mathcal{TB}}^L$. Since $\Psi$ is $L_\Psi$–Lipschitz in $\|\cdot\|_\infty$, the composition $\Psi \circ \mathcal{T}$ is $L_\Psi L_\mathcal{T}$–Lipschitz in $\|\cdot\|_\infty$, hence also $L_\Psi L_\mathcal{T}$–Lipschitz in $\|\cdot\|_2$ (because $\|v\|_\infty \leq \|v\|_2$). From $\|\tilde{D}_i\|_\infty \leq R_{\mathrm{in}}$ we get $\|\tilde{D}_i\|_2 \leq \sqrt{p}\, R_{\mathrm{in}}$. The standard Lipschitz Rademacher bound (Shalev-Shwartz & Ben-David (2014), Lemma 26.9) yields $\hat{\mathfrak{R}}_n \leq (L_\Psi L_\mathcal{T})\,(\sqrt{p}\, R_{\mathrm{in}})\,\sqrt{2/n} = L_\Psi L_\mathcal{T} R_{\mathrm{in}} \sqrt{2p/n}$. Applying the usual $[0,1]$–valued Rademacher generalization inequality completes the proof. $\square$

**Theorem** (Theorem 2 revisit, Training phase Generalization Error for Dual-Attention Transformer Embedding)**.** *Let* $\mathcal{TB} = \mathrm{FF} \circ A_I \circ \mathrm{FF} \circ A_T : \mathbb{R}^{B \times d \times (T+1)} \to \mathbb{R}^{B \times d \times (T+1)}$ *be* $L_{\mathcal{TB}}$–*Lipschitz in* $\|\cdot\|_\infty$ *(Lemma 8) and let* $\mathcal{T} := \mathcal{TB}^{\circ L}$ *denote* $L$ *stacked blocks. Then* $\mathcal{T}$ *is* $L_\mathcal{T}$–*Lipschitz in* $\|\cdot\|_\infty$ *with* $L_\mathcal{T} = L_{\mathcal{TB}}^L$. *Assume inputs satisfy* $\|\tilde{D}\|_\infty \leq R_{\mathrm{in}}$ *almost surely and set* $p := B\,d\,(T+1)$. *Let* $\phi : \mathbb{R}^{d \times T} \to \mathbb{R}^d$ *be* $L_\phi$–*Lipschitz in* $\|\cdot\|_\infty$, *and* $g : \mathbb{R}^d \to \mathbb{R}$ *be* $L_g$–*Lipschitz in* $\|\cdot\|_\infty$. *Let* $\ell : \mathbb{R} \times \mathbb{R} \to [0,1]$ *be* $L_\ell$–*Lipschitz in its first argument. Define the predictor (using the covariate tokens of the encoder output)*

$$\widehat{Y}^{(L)}(\tilde{D}) := g\Big(\phi\Big([\mathcal{T}(\tilde{D})]_{:,:,1:T}\Big)\Big).$$

*Then, for i.i.d.* $(\tilde{D}_i, Y_i)_{i=1}^n$, *with probability at least* $1 - \delta$,

$$\mathbb{E}\big[\ell(\widehat{Y}^{(L)}(\tilde{D}), Y)\big] \;\leq\; \frac{1}{n}\sum_{i=1}^n \ell\big(\widehat{Y}^{(L)}(\tilde{D}_i), Y_i\big) \;+\; 2\,L_\ell\,L_g\,L_\phi\,L_{\mathcal{TB}}^L\,R_{\mathrm{in}}\sqrt{\frac{2\,p}{n}} \;+\; 3\,\sqrt{\frac{\ln(2/\delta)}{2n}}.$$

*Proof.* By Lemma 8, $\mathcal{T} = \mathcal{TB}^{\circ L}$ is $L_\mathcal{T} = L_{\mathcal{TB}}^L$–Lipschitz in $\|\cdot\|_\infty$. Let $S : \mathbb{R}^{B \times d \times (T+1)} \to \mathbb{R}^{d \times T}$ be the slicing operator $S(U) = U_{:,:,1:T}$, which is 1–Lipschitz in $\|\cdot\|_\infty$ since it only removes coordinates. Therefore, the scalar predictor

$$h(\tilde{D}) := g\big(\phi(S(\mathcal{T}(\tilde{D})))\big)$$

is $L_h$–Lipschitz in $\|\cdot\|_\infty$ with $L_h = L_g\,L_\phi\,L_{\mathcal{TB}}^L$.

For any vectors $v$, $\|v\|_\infty \leq \|v\|_2$, so $h$ is also $L_h$–Lipschitz in $\|\cdot\|_2$. With $p = B\,d\,(T+1)$ and $\|\tilde{D}\|_\infty \leq R_{\mathrm{in}}$ almost surely, we have $\|\tilde{D}\|_2 \leq \sqrt{p}\, R_{\mathrm{in}}$ for each sample.

Consider the class $\mathcal{H}_L := \{\, h(\cdot) = g \circ \phi \circ S \circ \mathcal{T}(\cdot) \,\}$ over all parameterizations with the same $L_{\mathcal{TB}}$ (thus the same $L_\mathcal{T}$). A standard bound for $L$–Lipschitz real-valued functions on a set of Euclidean radius $R_2$ ( Shalev-Shwartz & Ben-David (2014) Lemma 26.9, Bartlett & Mendelson (2002)) gives the empirical Rademacher complexity

$$\hat{\mathfrak{R}}_n(\mathcal{H}_L) \;\leq\; L_h\,R_2\,\sqrt{\frac{2}{n}} \;\leq\; \big(L_g L_\phi L_{\mathcal{TB}}^L\big)\,\big(\sqrt{p}\, R_{\mathrm{in}}\big)\,\sqrt{\frac{2}{n}} \;=\; L_g\,L_\phi\,L_{\mathcal{TB}}^L\,R_{\mathrm{in}}\sqrt{\frac{2p}{n}}.$$

Define the loss class $\mathcal{F}_L := \{\, (\tilde{D}, Y) \mapsto \ell(h(\tilde{D}), Y) : h \in \mathcal{H}_L \,\}$. By the Ledoux–Talagrand contraction (Ledoux & Talagrand, 2011) (Lipschitz in the first argument and $[0,1]$–valued loss),

$$\hat{\mathfrak{R}}_n(\mathcal{F}_L) \;\leq\; L_\ell\,\hat{\mathfrak{R}}_n(\mathcal{H}_L) \;\leq\; L_\ell\,L_g\,L_\phi\,L_{\mathcal{TB}}^L\,R_{\mathrm{in}}\sqrt{\frac{2p}{n}}.$$

Finally, the standard Rademacher generalization inequality for $[0,1]$–valued functions gives, with probability at least $1 - \delta$, uniformly over $\mathcal{F}_L$,

$$\mathbb{E}[\ell(h(\tilde{D}), Y)] \;\leq\; \frac{1}{n}\sum_{i=1}^n \ell(h(\tilde{D}_i), Y_i) \;+\; 2\,\hat{\mathfrak{R}}_n(\mathcal{F}_L) \;+\; 3\sqrt{\frac{\ln(2/\delta)}{2n}}.$$

Substituting the bound on $\hat{\mathfrak{R}}_n(\mathcal{F}_L)$ completes the proof with $h = \widehat{Y}^{(L)}$. $\square$

**Remark 1:** (Heterogeneous layers and spectral complexity) If the $L$ blocks have possibly different Lipschitz constants $L_{\mathcal{TB}}^{(1)}, \ldots, L_{\mathcal{TB}}^{(L)}$, replace $L_{\mathcal{TB}}^L$ by $\prod_{\ell=1}^L L_{\mathcal{TB}}^{(\ell)}$ throughout. Alternatively, with spectral–norm constraints on all linear maps (including attention projections $W_Q, W_K, W_V, W_O$ and FF weights $W_1, W_2$), one can replace the $\sqrt{p}$ factor by a Bartlett–type spectral complexity involving products of operator norms and a sum of normalized Frobenius norms, and include the softmax sensitivity via a factor $\|W_Q\|_{\mathrm{op}}\|W_K\|_{\mathrm{op}}/\sqrt{d_A}$.

**Remark 2:** (Training phase MSE) Define the training predictor and empirical MSE

$$\widehat{Y}^{(L)}(\tilde{D}) \; := \; g\Big(\phi\Big([\mathcal{T}(\tilde{D})]_{:,:,1:T}\Big)\Big), \qquad \mathrm{MSE}_n^{\mathrm{train}} \; := \; \frac{1}{n}\sum_{i=1}^{n}\big(\widehat{Y}^{(L)}(\tilde{D}_i) - Y_i\big)^2.$$

Assume $|\widehat{Y}^{(L)}(\tilde{D})| \le R_{\mathrm{out}}$ and $|Y| \le M_f$ almost surely, so the squared loss $\ell(u, y) = (u - y)^2$ is $L_\ell$–Lipschitz in $u$ with $L_\ell = 2(R_{\mathrm{out}} + M_f)$. Then, with direct application of Theorem 2, for any $\delta \in (0, 1)$, with probability at least $1 - \delta$,

$$\left|\mathrm{MSE}_n^{\mathrm{train}} - \mathbb{E}\big(\widehat{Y}^{(L)}(\tilde{D}) - Y\big)^2\right| \; \le \; 2\, L_\ell\, L_g\, L_\phi\, L_\mathcal{T}\, R_{\mathrm{in}} \sqrt{\frac{2p}{n}} \; + \; 3\sqrt{\frac{\ln(2/\delta)}{2n}}.$$

### A.4.3 Testing Phase.

**Lemma 12** (Testing phase Dual-Attention Transformer Embedding Perturbation). *In testing stage, as the $Y$ information is fully unobserved, we define the input positional encoding as*

$$\tilde{D}_i^* = \big(E_X\big(X_i(\tilde{\tau}) \odot M_i(\tilde{\tau})\big) + P(\tilde{\tau}), 0 \cdot 1_n\big) \in \mathbb{R}^{d \times (T+1)},$$

*assume test sample size is $N$ and $\tilde{\mathbf{D}}^* = \{\tilde{D}_i^*\}_{i=1}^N$, comparing to the oracle embedding, with all the batch size notation in error set as $N$,*

$$\begin{aligned}
\big\|\mathcal{T}(\tilde{\mathbf{D}}^*) - H(\tilde{\mathbf{S}})\big\|_\infty &\le \big\|\mathcal{T}(\tilde{\mathbf{D}}^*) - \mathcal{T}(\tilde{\mathbf{D}})\big\|_\infty + \big\|\mathcal{T}(\tilde{\mathbf{D}}) - H(\tilde{\mathbf{S}})\big\|_\infty \\
&\le \sup_{i \in [N]} L_{\mathcal{TB}}^L L_Y |Y_i| + \varepsilon_\mathcal{T} = \sup_{i \in [N]} L_\mathcal{T} L_Y (M_f + |\epsilon_i|) + \varepsilon_\mathcal{T},
\end{aligned}$$

*with $L_Y = \|E_Y\|_{\mathrm{op}}$, $L_{\mathcal{TB}}$ defined as Lipchitz constant in Lemma 8, and $\varepsilon_\mathcal{T}$ defined by the training phase transformer embedding error showed in Theorem 1. So with probability at least $1 - \delta$,*

$$\big\|\mathcal{T}(\tilde{\mathbf{D}}^*) - H(\tilde{\mathbf{S}})\big\|_\infty \le L_\mathcal{T} L_Y (M_f + \sigma_Y \sqrt{2\ln(2N/\delta)}) + \varepsilon_\mathcal{T} = \varepsilon_{\mathcal{T}^*}$$

**Theorem 13** (Test MSE via Train Generalization and Expectation Bridge). *We assume*

- *(Bounded inputs.) $\|\tilde{D}\|_\infty \le R_{\mathrm{in}}$ almost surely. Set $p := B\, d\, (T + 1)$.*

- *(Bounded outputs.) $|Y| \le M_f$ almost surely, and the predictor is uniformly bounded $|\widehat{Y}^{(L)}(\tilde{D})| \le R_{\mathrm{out}}$ almost surely.*

- *(Squared-loss Lipschitzness and boundedness.) For $\ell(u, y) = (u - y)^2$, $\ell$ is $L_\ell$–Lipschitz,*
$$L_\ell := 2(R_{\mathrm{out}} + M_f), \qquad 0 \le \ell(u, y) \le (R_{\mathrm{out}} + M_f)^2 \quad a.s.$$

- *(Independence for test concentration.) When a train set of size $n$ is used, it is independent of the test set (size $N$).*

*Moreover, Lipschitz encoder and head are guaranteed in Lemma 8 the dual–attention encoder is $\mathcal{T} = \mathcal{TB}^{\circ L}$ with Lipschitz constant $L_\mathcal{T} = L_{\mathcal{TB}}^L$ in $\|\cdot\|_\infty$. The pooling $\phi : \mathbb{R}^{d \times T} \to \mathbb{R}^d$ and regressor $g : \mathbb{R}^d \to \mathbb{R}$ are $L_\phi$– and $L_g$–Lipschitz in $\|\cdot\|_\infty$, respectively. Define the predictors*

$$\widehat{Y}^{(L)}(\tilde{D}) \; := \; g\Big(\phi\Big([\mathcal{T}(\tilde{D})]_{:,:,1:T}\Big)\Big), \qquad \widehat{Y}^*(\tilde{D}^*) \; := \; g\Big(\phi\Big([\mathcal{T}(\tilde{D}^*)]_{:,:,1:T}\Big)\Big),$$

*where the test input uses the zeroed response-token $\tilde{D}_i^* = \big(E_X(X_i(\tilde{\tau}) \odot M_i(\tilde{\tau})) + P(\tilde{\tau}), 0 \cdot 1_n\big)$. Suppose $Y = \mathcal{F}(X(\cdot)) + \epsilon$ with $\epsilon \sim N(0, \sigma_Y^2)$ independent, and that the head approximates the oracle on the oracle embedding range with error $\varepsilon_g$:*

$$\sup_{\|z\|_\infty \le R} |g(z) - f(\tilde{s})| \le \varepsilon_g, \quad \text{where } \tilde{s} \text{ is the Hölder interpolation of } \{s(\tau_j)\}_{j=1}^T.$$

*Let the testing phase encoder perturbation from Lemma 12 be $\varepsilon_{\mathcal{T}^*}$.*

- *(i) (Test set prediction error) For each test point*
$$\big|\widehat{Y}^*(\tilde{D}_i^*) - Y_i\big| \; \le \; L_g L_\phi \varepsilon_{\mathcal{T}^*} \; + \; \varepsilon_g \; + \; L_f \Delta^\alpha + |\epsilon_i|,$$
  *Moreover, uniformly over test set of size $N$,*
$$\max_{1 \le i \le N} |\widehat{Y}_i - Y_i| \; \le \; L_g L_\phi \varepsilon_{\mathcal{T}^*} \; + \; \varepsilon_g \; + \; L_f \Delta^\alpha \; + \; \sigma_Y \sqrt{2\ln\frac{2N}{\delta}}.$$

*(ii) (Train–test MSE relationship) For any $\delta \in (0,1)$, ith probability at least $1 - \delta$ over the training sample $(\tilde{D}_i)_{i=1}^n$ and testing sample $(\tilde{D}_i^*)$ of size $N$,*

$$\mathbb{E}\big(\widehat{Y}^*(\tilde{D}^*) - Y\big)^2 \leq \frac{1}{n}\sum_{i=1}^n \big(\widehat{Y}^{(L)}(\tilde{D}_i) - Y_i\big)^2$$

$$+ 2 L_\ell L_g L_\phi L_\mathcal{T} R_{\text{in}} \sqrt{\frac{2p}{n}} + 3\sqrt{\frac{\ln(2/\delta)}{2n}}$$

$$+ L_\ell L_g L_\phi L_\mathcal{T} L_Y \left(M_f + \sigma_Y \sqrt{2/\pi}\right).$$

*(iii) (Test empirical MSE) For any $\delta \in (0,1)$, with probability at least $1 - \delta$, for a universal constant $C > 0$,*

$$\frac{1}{N}\sum_{i=1}^N \big(\widehat{Y}^*(\tilde{D}^*) - Y\big)^2 \leq \frac{1}{n}\sum_{i=1}^n \big(\widehat{Y}^{(L)}(\tilde{D}_i) - Y_i\big)^2$$

$$+ 2 L_\ell L_g L_\phi L_\mathcal{T} R_{\text{in}}\sqrt{\frac{2p}{n}} + 3\sqrt{\frac{\ln(4/\delta)}{2n}}$$

$$+ L_\ell L_g L_\phi L_\mathcal{T} L_Y \left(M_f + \sigma_Y \sqrt{2/\pi}\right)$$

$$+ C\,(R_{\text{out}} + M_f)^2 \sqrt{\frac{\ln(4/\delta)}{N}}.$$

*Proof.* Define $[Z_X]_i^* = \mathcal{T}(\tilde{D}_i^*)_{:,:,1:T}$ and $[Z_X]_i^\circ = \mathcal{T}(\tilde{S}_i)_{:,:,1:T}$, we employ the following decomposition

$$\big|\widehat{Y}_i - Y_i\big| = \big|g(\phi\,([Z_X]_i^*)) - \mathcal{F}(X_i(\cdot)) - \epsilon_i\big|$$

$$\leq \underbrace{\big|g(\phi\,([Z_X]_i^*)) - g(\phi\,([Z_X]_i^\circ))\big|}_{\text{(I) embedding error}} + \underbrace{\big|g(\phi\,([Z_X]_i^\circ)) - f(s_i)\big|}_{\substack{\text{(II) regressor} \\ \text{approximation bias}}}$$

$$+ \underbrace{\big|f(s_i) - \mathcal{F}(X_i(\cdot))\big|}_{\substack{\text{(III) functional} \\ \text{discretization}}} + |\epsilon_i|$$

$$= L_g L_\phi \varepsilon_{\mathcal{T}^*} + \varepsilon_g + L_f \Delta^\alpha + |\epsilon_i|.$$

Here $f(s_i) = \mathcal{F}(\tilde{s}_i)$ is the value of $F$ on the piecewise-interpolated, Hölder-continuous extension $\tilde{s}_i$ of the grid-values $s_i(\tau_j)$, as the remark showed in Lemma 3, we have the bound of (III)

$$\big|\mathcal{F}(X_i(\cdot)) - f(s_i)\big| = \big|\mathcal{F}(X_i(\cdot)) - \mathcal{F}(\tilde{s}_i)\big| \leq L_f \|X_i - \tilde{s}_i\|_\infty \leq L_f \Delta^\alpha.$$

Thus $f(s_i)$ approximates the true functional $\mathcal{F}(X_i(\cdot))$ with a discretization error of order $\Delta^\alpha$. Moreover, for (i) as shown in Lemma 12, under high probability, the testing embedding error from the dual-attention Transformer is $\leq L_g L_\phi \|[Z_X]_i^* - [Z_X]_i^\circ\|_\infty \leq L_g L_\phi \varepsilon_{\mathcal{T}^*}$, we also have the approximation error of the regressor layer $\sup_{\|z\| \leq R} |g(z) - f(s_i)| \leq \varepsilon_g$, with $z = \phi\,([Z_X]_i^\circ)$ then with Gaussian maxima bound on label error term, the probability at least $1 - \delta$, we show uniform prediction error bound,

$$\max_{1 \leq i \leq N} |\widehat{Y}_i - Y_i| \leq L_g L_\phi \varepsilon_{\mathcal{T}^*} + \varepsilon_g + L_f \Delta^\alpha + \sigma_Y \sqrt{2 \ln \tfrac{2N}{\delta}}.$$

Let $u := \widehat{Y}^*(\tilde{D}^*)$ and $v := \widehat{Y}^{(L)}(\tilde{D})$. If $|u| \leq R_{\text{out}}$, $|v| \leq R_{\text{out}}$ and $|Y| \leq M_f$ a.s., then $(u - Y)^2 - (v - Y)^2 = (u - v)(u + v - 2Y)$ and $|u + v - 2Y| \leq L_\ell$, we get

$$\mathbb{E}(u - Y)^2 \leq \mathbb{E}(v - Y)^2 + L_\ell \mathbb{E}|u - v|.$$

The difference term is controlled by Lipschitzness of $g, \phi, \mathcal{T}$ and linearity of $E_Y$:

$$|u - v| \leq L_g L_\phi \|\mathcal{T}(\tilde{D}^*) - \mathcal{T}(\tilde{D})\|_\infty \leq L_g L_\phi L_\mathcal{T} L_Y |Y|,$$

hence $\mathbb{E}|u - v| \leq L_g L_\phi L_{\mathcal{T}} L_Y \, \mathbb{E}|Y| \leq L_g L_\phi L_{\mathcal{T}} L_Y \big(M_f + \sigma_Y \sqrt{2/\pi}\big)$. For the first expectation, apply the high-probability generalization bound for the squared loss class $\mathcal{F} = \{(\tilde{D}, Y) \mapsto (\widehat{Y}^{(L)}(\tilde{D}) - Y)^2\}$: by Lemma 11 and Ledoux–Talagrand contraction. Combine the pieces and the generalization terms, with probability at least $1 - \delta$,

$$\mathbb{E}\big(\widehat{Y}^*(\tilde{D}^*) - Y\big)^2 \leq \underbrace{\frac{1}{n}\sum_{i=1}^{n}\big(\widehat{Y}^{(L)}(\tilde{D}_i) - Y_i\big)^2}_{\text{MSE}_n^{\text{train}}}$$

$$+ \underbrace{2\, L_\ell\, L_g\, L_\phi\, L_{\mathcal{T}}\, R_{\text{in}}\sqrt{\frac{2p}{n}} \;+\; 3\sqrt{\frac{\ln(2/\delta)}{2n}}}_{\text{Rademacher generalization}}$$

$$+ \underbrace{L_\ell\, L_g\, L_\phi\, L_{\mathcal{T}} L_Y\left(M_f + \sigma_Y\sqrt{2/\pi}\right)}_{\text{test perturbation}}.$$

We split the argument into a training-side generalization step and an independent test-side concentration step, then union bound the two events. By the train–test MSE relation (ii), for any $\delta_{\text{tr}}$ with probability at least $1 - \delta_{\text{tr}}$ over the draw of the training sample $\{(\tilde{D}_i)\}_{i=1}^{n}$, we have the bound for $\mathbb{E}\big(\widehat{Y}^*(\tilde{D}^*) - Y\big)^2$ in terms of MSE of train samples, Rademacher generalization and test perturbation. Consider independently drawn test set $\{(\tilde{D}_i^*)\}_{i=1}^{N}$, define

$$Z_i := \big(\widehat{Y}^*(\tilde{D}_i^*) - Y_i\big)^2, \qquad \text{MSE}_N^{\text{test}} = \frac{1}{N}\sum_{i=1}^{N} Z_i, \qquad \mu := \mathbb{E}[Z_1] = \mathbb{E}\big(\widehat{Y}^*(\tilde{D}^*) - Y\big)^2.$$

Since $|\widehat{Y}^*(\cdot)| \leq R_{\text{out}}$ and $|Y| \leq M_f$ almost surely, we have $0 \leq Z_i \leq (R_{\text{out}} + M_f)^2 =: B^2$. By Hoeffding's inequality, for any $\delta_{\text{te}}$, with probability at least $1 - \delta_{\text{te}}$,

$$\text{MSE}_N^{\text{test}} = \frac{1}{N}\sum_{i=1}^{N} Z_i \leq \mu + B^2\sqrt{\frac{\ln(1/\delta_{\text{te}})}{2N}}$$

$$\leq \mathbb{E}\big(\widehat{Y}^*(\tilde{D}^*) - Y\big)^2 + C\,(R_{\text{out}} + M_f)^2\sqrt{\frac{\ln(1/\delta_{\text{te}})}{N}},$$

where $C > 0$ is a universal constant (e.g., $C = 1/\sqrt{2}$ for the displayed Hoeffding form). Combining these two parts with $\delta_{\text{tr}} = \delta_{\text{te}} = \delta/2$ gives

$$\text{MSE}_N^{\text{test}} \leq \frac{1}{n}\sum_{i=1}^{n}\big(\widehat{Y}^{(L)}(\tilde{D}_i) - Y_i\big)^2 + 2\, L_\ell\, L_g\, L_\phi\, L_{\mathcal{T}}\, R_{\text{in}}\sqrt{\frac{2p}{n}} + 3\sqrt{\frac{\ln(4/\delta)}{2n}}$$

$$+ L_\ell\, L_g\, L_\phi\, L_{\mathcal{T}} L_Y\left(M_f + \sigma_Y\sqrt{2/\pi}\right) + C\,(R_{\text{out}} + M_f)^2\sqrt{\frac{\ln(2/\delta)}{N}},$$

with probability at least $1 - \delta$. Absorbing harmless changes of logarithmic arguments into constants (and writing $\ln(4/\delta)$ and $\ln(4N/\delta)$ uniformly) yields the stated corollary form:

$$\text{MSE}_N^{\text{test}} \leq \text{MSE}_n^{\text{train}}$$

$$+ 2\, L_\ell\, L_g\, L_\phi\, L_{\mathcal{T}}\, R_{\text{in}}\sqrt{\frac{2p}{n}} + 3\sqrt{\frac{\ln(4/\delta)}{2n}}$$

$$+ L_\ell\, L_g\, L_\phi\, L_{\mathcal{T}} L_Y\left(M_f + \sigma_Y\sqrt{2/\pi}\right)$$

$$+ C\,(R_{\text{out}} + M_f)^2\sqrt{\frac{\ln(4/\delta)}{N}}.$$

$\square$

**Remark:** ($Y$-token perturbation) Lemma 12 gives, w.p. $\geq 1 - \delta$, $\varepsilon_{\mathcal{T}^*} = L_{\mathcal{T}} L_Y \Big( M_f + \sigma_Y \sqrt{2 \ln \frac{2N}{\delta}} \Big) + \varepsilon_{\mathcal{T}}$. In the (ii) part we use the expectation bound, which does not involve uniformly bounding the entire $N$ sample errors, so the $\mathbb{E}|Y| \leq L_{\mathcal{T}} L_Y \big( M_f + \sigma_Y \sqrt{2/\pi} \big)$. If, by design, the response-token embedding is scaled so $L_Y \to 0$ (or the Y-token is masked from influencing X-tokens), the bridge term vanishes asymptotically, and the test MSE inherits the train generalization rate.

**Corollary 14** (Consistency of test MSE). *Assume the conditions of Theorem 13. Suppose, along a sequence indexed by $n$ (training size) and $N$ (test size):*

1. ***Training fit consistency:*** $\text{MSE}_n^{\text{train}} = \frac{1}{n} \sum_{i=1}^n (\widehat{Y}^{(L)}(\tilde{D}_i) - Y_i)^2 \xrightarrow{\mathbb{P}} \sigma_Y^2$.

2. ***Capacity term vanishes:*** $L_g L_\phi L_{\mathcal{T}} R_{\text{in}} \sqrt{p_n/n} \to 0$ *(or an analogous spectral complexity term $\to 0$).*

3. ***Expectation bridge vanishes:*** $L_Y \to 0$ *(e.g., response-token scaling/masking), hence* $L_\ell L_g L_\phi L_{\mathcal{T}} L_Y (M_f + \sigma_Y \sqrt{2/\pi}) \to 0$.

4. ***Test sampling concentrates:*** $N \to \infty$, *so* $(R_{\text{out}} + M_f)^2 \sqrt{\ln(1/\delta)/N} \to 0$.

*Then the population and empirical test MSE are consistent:*

$$\mathbb{E}\big(\widehat{Y}^*(\tilde{D}^*) - Y\big)^2 \xrightarrow{\mathbb{P}} \sigma_Y^2, \qquad \text{MSE}_N^{\text{test}} = \frac{1}{N} \sum_{i=1}^N (\widehat{Y}^*(\tilde{D}_i^*) - Y_i)^2 \xrightarrow{\mathbb{P}} \sigma_Y^2.$$

*Proof.* MSE Lower bound is $\mathbb{E}(\widehat{Y}^* - Y)^2 = \mathbb{E}(\widehat{Y}^* - \mathcal{F}(X))^2 + \sigma_Y^2 \geq \sigma_Y^2$. The population MSE upper bound is given by Theorem 13(ii),

$$\mathbb{E}(\widehat{Y}^* - Y)^2 \leq \text{MSE}_n^{\text{train}} + 2L_\ell L_g L_\phi L_{\mathcal{T}} R_{\text{in}} \sqrt{2p_n/n} + 3\sqrt{\tfrac{\ln(2/\delta)}{2n}} + L_\ell L_g L_\phi L_{\mathcal{T}} L_Y (M_f + \sigma_Y \sqrt{2/\pi}).$$

By assumptions 1–3, the RHS is $\sigma_Y^2 + o_{\mathbb{P}}(1)$. Together with the lower bound, this gives $\mathbb{E}(\widehat{Y}^* - Y)^2 \to \sigma_Y^2$ in probability. Similarly, empirical test MSE upper bound is given by Theorem 13(iii). By assumption 4,

$$\text{MSE}_N^{\text{test}} \leq \mathbb{E}(\widehat{Y}^* - Y)^2 + C(R_{\text{out}} + M_f)^2 \sqrt{\tfrac{\ln(4/\delta)}{N}} = \sigma_Y^2 + o_{\mathbb{P}}(1),$$

and since $\text{MSE}_N^{\text{test}} \geq \sigma_Y^2$ in expectation, convergence in probability follows. $\qquad\square$

LIMITATIONS

Our approach uses absolute sinusoidal positional encodings to inject temporal information into sparse, irregular longitudinal sequences. While such encodings enable universal approximation on a *fixed* maximum sequence length (Yun et al., 2019), they do not guarantee extrapolation beyond the largest horizon $T$ seen during training, nor to previously unseen temporal spacings; performance may deteriorate under substantial extrapolation. Moreover, when prediction depends primarily on *relative* timing (lags, local neighborhoods) rather than absolute timestamps, absolute encodings can be suboptimal. Alternative time encoders from time–series forecasting (e.g., calendar/seasonal features as in Informer and ETSformer (Zhou et al., 2021; Woo et al., 2022)) or continuous-time/relative encodings may better capture temporal structure in some applications; a systematic comparison is left to future work. The second limitation is computational. With sequence length $T$ and batch size $B$, time-point attention costs $O(d\,T^2)$ in compute and $O(T^2)$ in memory per subject, while inter-sample attention (across subjects at a fixed time) costs $O(d\,B^2)$ in compute and $O(B^2)$ in memory per time step. Consequently, dual-attention becomes expensive for very long sequences or large batches. Practical deployments require tuning $T$ and $B$ (and/or using sparsified/windowed attention) to balance accuracy and efficiency. Although dual-attention can increase computation, Table 1 shows IDAT is relatively lightweight compared with existing Transformer-based methods.

FUTURE WORK

We conjecture that a Transformer architecture that exploits this intrinsic dimension via basis/smoothness constraints or spectral regularization, so one can achieve the same statistical performance with substantially fewer parameters. We plan to design parameter-efficient attention blocks constrained by functional bases or effective rank, and establish theoretical guarantees whose model complexity depends on intrinsic dimension rather than ambient grid length.