# OpenReview forum: "Interpretable Transformer Regression for Functional and Longitudinal Covariates"
_ICLR.cc/2026/Conference — Submitted to ICLR 2026_

### Official Review · Reviewer_63go · 2025-10-15

**Soundness:** 2
**Presentation:** 2
**Contribution:** 1
**Rating:** 2
**Confidence:** 3

**Summary:**

The paper proposes the Interpretable Dual-Attention Transformer (IDAT), which is a model for predicting scalar outcomes from sparse, irregular, and noisy longitudinal data. Specifically, IDAT integrates time-point attention to capture temporal structure and inter-sample attention to share information across similar subjects. The methods is trained end-to-end without imputation.

**Strengths:**

- **Sound approach:** IDAT unifies temporal (time-point) and cross-sample (inter-sample) attention mechanisms. This design allows the model to capture both individual trajectory dynamics and shared structures across subjects, and it seems to work well in practice.

- **Mesh size insight:** Personally, I found th mesh size tradeoff provides helpful insights how to balance computational efficiency with prediction bias.

**Weaknesses:**

- The paper is very **straight forward**. The paper basically uses a grid discretization for functional data input.

- **Dual attention** = known parts, new combo: Time-point self-attention and cross-sample (nearest-neighbor-like) pooling are both established ideas. IDAT simply combines them in one encoder and trains them  end-to-end with masks.

- **Important parts of the paper (e.g., ALL results on simulated data in Sec. 4.1, all Lemmas) are relegated to the appendix.** I understand that not all results can always make it to the main body of the paper. However, here it seems the authors have moved too much important parts to the supplements.

- **Performance is very regime-specific:** The strongest gains are reported when only < 50% of time points are observed, which is fair since this is the regime the method is designed for. However, as sparsity decreases, it seems like the advantage will narrow outside of the very sparse regime

- **Standard consistency and Lipschitz arguments:** The main results are architectural Lipschitz/approximation lemmas and two theorems: encoder consistency and a training-MSE generalization bound under standard assumptions (boundedness, stable SGD, $p/n \to 0$ ).
These are mathematically sound but completely expected for modern attention+MLP networks. My issue here is that these insights do not provide information /  do not translate into design guidance (e.g., how to pick layers, heads, grid size beyond asymptotics)

- **Variance reduction claim:** Inter-sample attention reduces embedding noise by $O(B^{−1/2})$ when subjects are similar. However, this is a mini-batch averaging effect. The result does not quantify when the bias from pooling outweighs variance gains (of note, the paper concedes this trade-off empirically).

- **Presentation:** the paper would benefit a lot from improving presentation and writing. As mentioned earlier, many components that appear important are moved to the supplements for no obvious reason.

**Questions:**

- How does IDAT fundamentally differ from existing attention architectures that combine temporal and relational modeling (such as TabNet, Set Transformers, or hierarchical temporal attention models)? What isthe novelty beyond combining two known mechanisms?

- Why is inter-sample attention considered novel?

- The paper provides consistency and Lipschitz-type generalization proofs, but these are standard for attention networks. Specifically, what new theoretical insight do these results provide about dual-attention?

- The variance reduction lemma assumes similarity among samples. Can the authors provide any conditions or diagnostics for when inter-sample attention introduces bias instead, for example in in heterogeneous populations?

- Regarding the variance reduction claims, how does the train-test mismatch around the Y-token affect consistency in practice?

- The interpretability claim rests on “weights acting as regression coefficients.” Is it possible to somehow interpret these weights quantitatively or only qualitatively as saliency maps?

- Sinusoidal positional encodings do not extrapolate well. How does IDAT handle unseen time grids? Would it be posssible to incorporate relative or learned encodings?

---

> ### Author Response · Authors · 2025-11-23
> **Response to Reviewer 63go**
>
> We thank the reviewer for their detailed review and thoughtful feedback, which has been valuable in helping us improve our paper presentation. We are grateful for the opportunity to improve our paper based on this feedback.
>
> **Response to Weakness 1 and 2:**
>
> We acknowledge that the individual components of IDAT (time-point self-attention and cross-sample pooling) are indeed established ideas. However, these designs are new to the longitudinal data, which are fundamentally different from tabular data. The novelty is not in inventing new attention primitives, but in how we tailor and integrate them for sparse functional data:
>
> - Functional vs. tabular structure: Even after discretization, longitudinal trajectories retain a continuous-time structure, so adjacent grid entries represent nearby time points, not independent features. This allows us to control discretization error via smoothness assumptions and requires time-aware positional encodings that reflect actual temporal distances, unlike tabular transformers where feature order is arbitrary with increment dependent positional encodings.
> - Population-aware stabilization: Inter-sample attention is not just mini-batch averaging; it implements the ``borrowing strength'' principle from longitudinal statistics. The attention weights adaptively down-weight dissimilar subjects so that, even in sparse regimes, embeddings are stabilized by functionally similar neighbors, mirroring random effects model or empirical Bayes shrinkage, but inside the transformer encoder. This aspect, along with the theoretical property, has never been investigated.
> - Grid integration and masking: Discretization is not mere preprocessing, sharing a common grid is what allows the model to learn which time regions are informative across subjects, while the mask mechanism encodes the sampling schedule.
>
> Each component of IDAT, while inspired by prior transformer ideas, is deliberately redesigned for the end-to-end challenge of predicting or classifying from irregular, sparse functional inputs, which is particularly difficult because each subject provides so little information. Moreover, IDAT is an unifying approach, it works for densely or sparsely recorded functional data whether the measure schedule is regular or irregular.
>
> **Response to Weakness 3:**
>
> We want to thank the reviewer for the suggestion.  In the revision we will:
>
> 1. Move the main theoretical guarantees (consistency/identifiability, variance reduction, discretization error) into the body with concise statements and intuitive explanations.
> 2. Bring the key simulation data results into the main text so readers can immediately see how IDAT compares to impute-predict pipelines (including the state-of-the-art tabular pretrained model TabPFN) across the sparsity spectrum.
>     This will ensure that the core contributions, including statistical guarantees tailored to longitudinal data and empirical evidence under severe sparsity are visible.
>
> **Response to Weakness 4:**
>
> IDAT’s largest absolute gains appear in the very sparse regime, which is exactly the difficult regime we set out to fix, but two clarifications are important:
>
> 1. Sparse regime is normal for longitudinal data: **In many longitudinal cohorts, less than half of observation is actually the norm, not a corner case.** This makes impute-predict pipelines unreliable, as imputation error would struggle and there are barely enough to infer trends. Real datasets like the HIV VL data we demonstrated has mean missingness 65\%, so excelling in this regime is practically necessary.
> 2. Competitive outside the sparse regime: Even when observations become denser, IDAT remains near the top. **Across 30 benchmark cases, IDAT ranks in the top three 26 times among the other 19 baseline models**, including TabPFN and other dense-regime methods. The gap does narrow when sparsity drops, this is as expected when competitors are designed for dense inputs; but IDAT never collapses, it stays competitive while retaining its advantage on the sparse subjects that standard models handle poorly. Real datasets like NCDS (8–96\% missingness) and HIV (47–88\%) routinely contain subjects spanning the entire sparsity spectrum, IDAT provides reliable predictions across the full spectrum of sparsity encountered in practice.

---

> ### Author Response · Authors · 2025-11-23
> **Response to Reviewer 63go**
>
> **Response to Weakness 5:**
>
> While consistency and Lipschitz-type generalization proofs are indeed
> standard for attention networks. What we aimed to highlight is that, in
> sparse functional settings, these "expected" results carry different
> consequences, as identifiability under sparse sampling can be an issue.
> In our data each subject has very few observations ($n_i \approx 5$), so
> there are infinitely many smooth trajectories consistent with those
> points unless we control discretization error and leverage smoothness
> assumptions. The encoder-consistency guarantee (Theorem 1) shows that,
> even when observations are sparse, masked, and corrupted by measurement
> error, IDAT still converges to the true underlying trajectory on the
> grids. **This is something that is not automatic in this regime because
> sparsely sampled functional data cannot be uniquely identified without
> smoothness assumptions [1,2,3]**. This is
> qualitatively different from standard transformer analyses, which
> typically assume fully observed or densely sampled inputs.
>
> **Response to Weakness 6:**
>
> We agree that when subjects are truly i.i.d. the bound reduces to the familiar mini-batch averaging effect—that is precisely what we note in remarks of Lemma 9. The first remark states that if the attention weights are uniform, we recover the the most variance reduction. When the weights are not uniform but adaptively concentrate on functionally similar subjects, so the same averaging principle holds as long as the weights remain sufficiently spread. The second remark then makes explicit how measurement/label noise propagate through the encoder via the embedding operator norms and Lipschitz constants, so you can see exactly how the embedding-noise enters the bound. The variance guarantee hinges on the learned weights aligning with functional similarity, when they do not, the bias–variance trade-off behaves exactly as we observe empirically.
>
> **Response to Weakness 7:**
>
> Thanks for the candid feedback. We will reorganize the paper so that all core components, such as the key theoretical results, main simulation outcomes, and interpretability analysis appear in the main text with concise explanations.
>
>
> [1] Hall, P., Müller, H. G., & Wang, J. L. (2006). Properties of principal component methods for functional and longitudinal data analysis.
>
> [2] Wang, J. L., Chiou, J. M., & Müller, H. G. (2016). Functional data analysis. Annual Review of Statistics and its application, 3(1), 257-295.
>
> [3] Li, Y., & Hsing, T. (2010). Uniform convergence rates for nonparametric regression and principal component analysis in functional/longitudinal data.

---

> ### Author Response · Authors · 2025-11-23
> **Response to Reviewer 63go**
>
> **Response to Question 1:**
>
> IDAT differs from existing attention architectures through its joint modeling of temporal and inter-sample dependencies within a single encoder. This is fundamentally different from hierarchical or two-stage architectures (e.g., TabNet processes features sequentially, Set Transformers apply attention within sets separately, hierarchical temporal attention models process temporal structure first then aggregate): in these methods, temporal and relational modeling are decoupled—either processed sequentially or in separate stages. This decoupling is problematic in sparse and irregular setting as sparse data makes decoupled modeling unstable.  In IDAT, the temporal patterns can benefit from similar subjects learned from the inter-sample attention; whereas the inter-sample attention better capture subject similarity using temporal relationship learned in the time-point attention.
>
> **Response to Question 2:**
>
> We appreciate the chance to clarify. We’re not claiming inter-sample attention is architecturally novel in a generic sense; our contribution is showing that this mechanism, which looks like cross-sample pooling in the i.i.d. case, implements the classical ``borrowing strength'' principle from longitudinal analysis within a transformer encoder. In sparse functional settings, individual trajectories alone are too noisy, so statisticians routinely use random effects, empirical Bayes, or hierarchical models to shrink estimates toward population structure. IDAT’s inter-sample attention is motivated accordingly: it weights subjects by functional similarity so that only the most informative neighbors contribute, rather than simply averaging over a mini-batch. To our knowledge, this statistically motivated use of attention to stabilize embeddings under irregular sparsity hasn’t been explored in prior transformer work on tabular or time-series data. Moreover, we show numerically when the data is clustered, this inter-sample attention is also able to detect the heterogeneity of the subjects.
>
>
> **Response to Question 3:**
>
> **Please refer to A.3.1 where we provide an overview of our theoretical results.**
> - In Lemma 3 we bound the discretization error so that masking and measurement noise can be controlled via the smoothness constants and mesh size. The subsequent lemmas propagate those bounds through both attention blocks, showing that the entire dual-attention architecture remains Lipschitz with respect to the underlying functional signal.
> - Lemma 9 then isolates the variance-reduction effect of inter-sample attention under the i.i.d. assumption.
> - As mentioned before, Theorem 1 shows that despite sparse, masked, noisy inputs, the IDAT encoder converges to the oracle embedding we would have obtained had we have complete, noiseless trajectories on the grid.
> These are not standard transformer results because existing attention theory **presumes fully observed sequences and sidesteps the identifiability problem inherent in sparse functional data.** Our proof demonstrates that the dual-attention stack actually resolves that identifiability issue by jointly using smoothness (time-point attention) and cross-subject borrowing strength (inter-sample attention).
>
> **Response to Question 4:**
>
> We acknowledge that the variance bound assumes the attention weights concentrate on genuinely similar subjects. Borrowing information works when data share a common model or structure. Rather than hard-coding a parametric cluster condition, which indeed would be dataset-specific, we rely on diagnostics that the model already produces. A practical way to guard against pooling bias is to use the model’s own embeddings and attention weights as diagnostics.
>
> After running IDAT, as suggested in line 353-355, we can:
> 1. cluster the learned embeddings or attention-weight vectors (as we do for the BMI data) to check that heavily weighted neighbors truly share functional structure,
> 2. perform FPCA [1] on the learned embeddings and cluster the leading FPC scores (e.g., via k-means).
>
> Both procedures reveal whether subjects receiving high mutual attention are from coherent groups; if not, inter-sample attention may introduce bias and the time-point–only variant is preferable. We also mention in the paper (Lines 477–481) that the inter-sample and time-point attentions can be blended via a learnable gate.
>
> [1] Yao, F., Müller, H. G., & Wang, J. L. (2005). Functional data analysis for sparse longitudinal data. Journal of the American statistical association, 100(470), 577-590.

---

> ### Author Response · Authors · 2025-11-23
> **Response to Reviewer 63go**
>
> **Response to Question 5:**
>
> Regarding the variance reduction claims and consistency in practice, **we address the train-test mismatch around the Y-token in Lemma 12**. Specifically, Lemma 12 provides theoretical bounds on the perturbation effect of the Y-token, characterizing how deviations from the training setup (such as train-test mismatch or random masking of Y-tokens during training) affect the model's output consistency. The lemma establishes that the perturbation on the Y-token is bounded, ensuring that small variations in Y-token usage between training and testing do not lead to unbounded changes in predictions. To bridge the gap between training and testing in practice, we specifically implemented random masking of the Y-token during the training stage in our IDAT implementation. This design choice helps better characterize the relationship from training to a scenario closer to the testing stage, reducing the practical impact of train-test mismatch by exposing the model to varying Y-token configurations during training. This approach aligns with standard regularization techniques (e.g., dropout, masked language modeling) that improve model robustness to variations between training and testing.
>
> **Response to Question 6:**
>
> We demonstrate that IDAT attention weights enable quantitative interpretation:
>
> 1. On the NCDS BMI dataset, we apply IDAT attention weights to unsupervised k-means clustering with the number of clusters $k$ determined by cross-validated k-nearest neighbors (kNN). The kNN cross-validation method quantitatively determines that there are exactly two clusters in the data based on attention-weighted embeddings, demonstrating that attention weights capture meaningful quantitative structure. We then validate that these two clusters correspond to scientifically meaningful BMI trajectory patterns: as shown in Figure 8(b), the clusters correspond to distinct BMI trajectory groups (lower vs. higher BMI at age 62), where at baseline (age 7) the two groups have very similar BMI values, but as age progresses they follow quantitatively different BMI trajectories. **The attention-weighted clustering successfully captures this longitudinal divergence that is not apparent at baseline**, demonstrating quantitative interpretability through numerical cluster identification and statistically significant trajectory differences.
>
> 2. Next, we demonstrate quantitative interpretability by comparing IDAT attention weights with empirically computed time-domain importance based on functional derivatives. For the nonlinear functional model in case II, we compute the functional derivative (Gateaux derivative) as it measures how much a scalar functional $F(X(\cdot))$ changes when you perturb $X(t)$ at time $t$, which measures the actual quantitative contribution of each time point to the response. **The mean attention weights (normalized by absolute value) among testing set matches the importance of the region** correctly identify and quantify critical time regions where functional coefficients are non-zero. This part will be added in the revised version.
>
>
> **Response to Question 7:**
>
> We agree with the reviewer's concern about the extrapolation issue with sinusoidal positional encodings. This limitation has been addressed in our **Limitations section (Lines 2246-2254)**, we also propose possible adjustments there.

---

> > ### Comment · Reviewer_63go · 2025-11-26
> >
> > Thank you for your answer.
> >
> > **Weaknesses**:
> >
> > **Weakness 1 and 2:**
> >
> > Thank you for the clarification. The contribution regarding the **theoretical analysis is sound**, and I appreciate the clarification. Still, to me it seems that the core idea lies in **combining two existing frameworks**, and applying them to a new setting (as acknowledged by the authors themselves).
> >
> > **Weaknesses 3 and 7:**
> >
> > Thank you. Given the amount of time since the rebuttal started, I would have appreciated an updated version of the paper to see how the revised paper would look like.
> >
> > **Weaknesses 4, 5 and 6:**
> >
> > Thank you for the clarification.
> >
> > **Questions**:
> >
> > My questions have been addressed. Regarding **Question 7**, again, I would have appreciated a revised version of the paper. The limitations should be included in the main body, not in the supplements.
> >
> > ____
> >
> > Overall, I think this is a neat paper, with a **sound theoretical investigation**.
> >
> > Still, I think that the idea lies in **combining two existing approaches**, and there is **very limited novelty** beyond the theoretical investigation. I also question why a single model architecture should generally be preferred over a decoupled model. Further, I would have appreciated a revised version of the paper given the amount of time since the rebuttal period started; it is difficult for me to assess what a final camera-ready version would look like, as,  in my opinion, there would be much formatting required to move.
> >
> > **I will raise my score to 4**, and would like to reiterate that i) the paper is a sensible theoretical analysis of ii) combining two existing approaches, limiting the overall novelty of the paper. *I would not mind if the paper gets accepted, but personally, I do **not** think the key idea of the work is strong and novel enough to make it for a top conference like ICLR.*

---

### Official Review · Reviewer_ZLuK · 2025-10-31

**Soundness:** 2
**Presentation:** 3
**Contribution:** 3
**Rating:** 6
**Confidence:** 3

**Summary:**

This paper proposes a new method, the Interpretable Dual-Attention Transformer (IDAT), for scalar-on-function regression where the functional covariate is observed sparsely and irregularly over time. IDAT discretizes the time axis into a grid and uses explicit missing-value masks, avoiding imputation and enabling end-to-end training. It introduces two attention mechanisms: (1) Time-Point Attention, which encodes both local and long-range temporal structure within a subject's trajectory, and (2) Inter-Sample Attention, which borrows information across similar subjects in a batch. The learned attention weights are interpretable and reveal predictive time windows and cohort clusters. The paper also provides theoretical analysis of prediction error bounds and consistency. Experiments on both simulated and real-world datasets are conducted to evaluate the proposed method.

**Strengths:**

1. The motivation for addressing scalar-on-function regression with sparse, irregular, and noisy longitudinal data is clearly described.
2. A new solution that combines time-point and inter-sample attention in a Transformer framework is proposed, directly targeting the challenges of missingness and heterogeneity.
3. Theoretical analysis for the error bound and consistency of the training and testing phases MSE is provided.
4. Experiments comparing the proposed method with more than 19 baselines on both simulated and real-world datasets are conducted to evaluate its effectiveness.

**Weaknesses:**

1. The experiments lack an ablation study to illustrate the importance of each module in the model, such as time-point attention and inter-sample attention, respectively.
2. The hyperparameters used in the experiments should be described clearly. In addition, a hyperparameter sensitivity analysis should be provided for important hyperparameters, such as batch size and grid size.
3. The method for simulating different levels of sparsity should be described clearly. It is recommended to also simulate different missing patterns, if not already done.

**Questions:**

Please refer to the Weaknesses.

---

> ### Author Response · Authors · 2025-11-23
> **Response to Reviewer ZLuK**
>
> Thank you for the insightful review and the clear summary of IDAT’s strengths and remaining gaps. We appreciate the chance to clarify the current design choices and outline the revisions we will include in the camera-ready version.
>
> **Response to Weakness 1:**
>
> We chose not to perform an ablation study removing the
>         time-point attention module because longitudinal data
>         fundamentally possesses natural temporal trajectory structure.
>         Temporal ordering and dependencies between observations are
>         inherent properties of functional data that distinguish it from
>         standard tabular data. When temporal information is neglected,
>         the model collapses to a standard tabular transformer that
>         treats observations as independent features without temporal
>         structure. As demonstrated throughout our experiments
>         (simulation studies and real data analysis), IDAT consistently
>         outperforms tabular transformer baselines, providing indirect
>         evidence for the importance of temporal attention. Removing
>         temporal attention would essentially reduce our method to a
>         baseline that we have already shown to be inferior, making such
>         an ablation redundant.
>
> We have conducted an ablation study demonstrating the importance
>         of the inter-sample attention module in all the numerical
>         comparison (the IDAT w/o $A_I$ model). To highlight, In Figure 6
>         (Simulation Case III), we compare test set embeddings learned by
>         IDAT models with and without inter-sample attention. When
>         inter-sample attention is removed, the learned embeddings
>         exhibit collapsed cluster separation: subject profiles become
>         substantially more similar to each other and differ only
>         marginally in early trajectory stages. This demonstrates that
>         inter-sample attention is essential for identifying and
>         preserving the underlying cluster structure in functional
>         representations, validating its critical role in the model
>         architecture.
>
>
> **Response to Weakness 2:**
>
>  We report the hyperparameters in lines 770-776 for the simulation
>     and lines 967-971 for the data implementation. The following is an
>     added hyperparameter sensitivity analysis we plan to add in the
>     revise version:
>
>                  Hyperparameters/ Sparsity  ssparse (10%)   vsparse (20%)   sparse (50%)   dense (80%)   full(100%)
>       ------------------------------------ --------------- --------------- -------------- ------------- ------------
>           Larger Mesh ($T= 50$, $B = 128$)     0.0353          0.0253          0.0062        0.0012        0.0004
>             Baseline ($T= 100$, $B = 128$)     0.0165          0.0063          0.0024        0.0006        0.0003
>         Smaller Mesh ($T= 200$, $B = 128$)     0.0112          0.0038          0.0018        0.0004        0.0002
>         Smaller Batch ($T= 100$, $B = 64$)     0.0232          0.0088          0.0033        0.0007        0.0003
>             Baseline ($T= 100$, $B = 128$)     0.0165          0.0063          0.0024        0.0006        0.0003
>         Bigger Batch ($T= 100$, $B = 256$)     0.0119          0.0042          0.0019        0.0006        0.0003
>
>       Hyperparameter sensitivity analysis on Simulation Case I (i.i.d. functional linear regression setting) illustrating how grid size (mesh resolution) and batch size affect performance.
>
> Finer grids (smaller mesh) mitigate discretization bias and reduce MSE, matching our theoretical analysis; whereas coarser grids inflate bias and hence MSE. Larger batch sizes, under this i.i.d. setting, yield stronger variance reduction through inter-sample attention, further lowering MSE.
>
> **Response to Weakness 3:**
>
>  We thank the reviewers for highlighting the need to describe our sparsity construction, we will add the following clarification in the revision.
> - Case I / I\* (non-informative sparsity): Start with the full latent trajectory $X_i(t)$ on the length-100 grid and then subsample grid indices uniformly without replacement to achieve the desired sparsity level.
> - Case II / II\* (informative sparsity): Start with the full latent trajectory $X_i(\tau)$ on the length-100 grid, assign sampling weights proportional to $w_{i,j}\propto1+ \gamma(|\beta_2(\tau_j)|+|\beta_3(\tau_j)|)$, so time points where the response actually depends on the trajectory are more likely to be observed.
> - Case III/III\* (informative sparsity): The sampling weight is proportional to $w_{i,j} \propto 1+ \gamma\times[1(\tau_j\ge 0.5,g=1)+1(\tau_j\ge 0.5,g=1)]$, so each cluster is sampled more densely in the region where its mean shift differs.
>
> Across these cases we demonstrate both non-informative and informative observation schemes. Note that as long as training and testing follow the same sampling plan, IDAT handles both regimes without additional assumptions.

---

### Official Review · Reviewer_jqgT · 2025-11-04

**Soundness:** 2
**Presentation:** 2
**Contribution:** 2
**Rating:** 4
**Confidence:** 2

**Summary:**

The paper proposes IDAT (Interpretable Dual-Attention Transformer), a method for a scalar-on-function regression over time. IDAT uses a dual attention module (time-point attention and inter-sample attention), and, thus, it can handle sparsely and irregularly measured longitudinal data. Furthermore, the learned attention weights are interpretable. The authors also provided the theoretical guarantees for the consistency of  IDAT. Finally, the paper compares the new method against multiple other baselines and demonstrates its effectiveness for the scalar-on-function regression.

**Strengths:**

The paper provides a comprehensive theory on the consistency of the proposed method. Also, the experimental evaluation is very extensive (it includes 19 baselines and multiple datasets).

**Weaknesses:**

I have outlined a couple of main concerns for the paper:
- **Lack of assumptions for missingness**. I lacked the assumptions on the missigness (or measurement intensity), namely, whether the observation times are informative of the outcome. If yes, shouldn’t the missingness mask be included as an input itself?
- **Limited contribution**. I my opinion, the main method of the paper is a simple combination of multiple standard Transformer-based approaches. Yes, technically, IDAT might be new in this specific setting of scalar-on-function regression. Yet, overall, I cannot pinpoint a single non-trivial or unique approach that was used in this setting.

I am curious to hear the authors’ response, and I am open to further discussion.

Other minor concerns include the following:
- I found the abstract hard to read, as it is overloaded with very specific, tiny details and misses the broader picture.
- Same for the theoretical part, the implications for the theoretical statements were missing.

**Questions:**

- What is the main motivation to use the inter-sample attention (especially given that the data is i.i.d.)? I understand that they might help to reveal cluster patterns in data, but don’t they increase the variance of the scalar-on-function regression?
- Line 23. “The learned attention weights are interpretable...” Isn’t it true for all the Transformers?
- I wonder whether the stream of literature on marked point processes is relevant or could be adapted to this work (e.g., [1])?

References:
- [1] Panos, Aristeidis. "Decomposable transformer point processes." Advances in Neural Information Processing Systems 37 (2024): 88932-88955.

---

> ### Author Response · Authors · 2025-11-23
> **Response to Reviewer jqgT**
>
> We thank the reviewer for their constructive feedback and thoughtful questions, which have helped us clarify key aspects of our work. We hope that the explanations provided above address the reviewer's concerns and clarify any confusion.  We look forward to further discussion and are grateful for the opportunity to improve our paper based on this feedback.
>
> **Response to Weakness 1:**
>
> It is important to clarify that **there is no \``missing\'' data in our set up as the longitudinal data were only planned  to be collected at these scheduled times**. Thus, each observed data point $(t_{i,j}, X_i(t_{i,j}))$ represents a real measurement collected at time $t_{i,j}$, and the irregular sampling pattern reflects the natural structure of the data collection process rather than missing observations. However, when longitudinal data with irregular sampling times is transformed into a fixed-size tabular format, entries corresponding to grid points where no actual measurements were collected are treated as  \``masked data\'' in the implementation. We apologize for the confusion and will **replace the term \``missing\'' with \``masked value/masked data\'' in the revised version**. Therefore this type of missingness is fundamentally different from traditional missing data scenarios, where values are missing due to measurement failures, dropouts, or other data collection issues. Instead, the missing entries in the discretized representation are artifacts of the transformation from  continuous or irregularly sampled functional domain to a discrete fixed-size vector format. The mask is deterministic based on the sampling schedule with respect to the grid (masked = no nearby measurement), not random missingness, so traditional missing-data assumptions are unnecessary.  That being said, on the time grid point of view, this type of missingness would be missing completely at random as the missingness is unrelated to both the observed and unobserved data.
>
> To summarize, there is no missing data in the data collection process but the conversion of the data into discrete time grid for the implementation of the algorithm treats the empty cells as missing data that are missing completely at random.
>
> **Response to Weakness 2:**
>
> Longitudinal data, even after discretization, retains an intrinsic temporal structure that tabular or high-dimensional feature vectors do not. **Existing longitudinal transformer models adapt only temporal attention, without leveraging information shared between subjects, thereby neglecting the statistical principle of "borrowing strength" that is especially necessary in sparse regimes.**
> While attention mechanisms themselves are established concept, the way IDAT couples time-point (temporal) attention with inter-sample (borrowing-strength) attention is statistically motivated and tailored to the challenges of sparse, irregular longitudinal trajectories. These two attention pathways reinforce each other: time-point attention captures subject-specific dynamics, while inter-sample attention leverages population structure to stabilize predictions when per-subject data are scarce. Empirically, this design yields the first truly end-to-end transformer for scalar-on-function prediction task. IDAT is shown to consistently outperforms imputation-prediction pipelines, including TabPFN, a state-of-the-art method for tabular data that recently appeared in Nature. IDAT's strong performance across the various sparsity spectrum is critical in practice because individual subjects can have vastly different observation densities. Finally, our theoretical analysis shows that, under a standard random-effects model for longitudinal data, IDAT enjoys discretization-error control, consistency, and identifiability guarantees theoretical properties that existing transformer-based baselines lack.

---

> ### Author Response · Authors · 2025-11-23
> **Response to Reviewer jqgT**
>
> **Response to Weakness 3:**
> We appreciate this feedback and will revise the abstract to highlight the broader contributions of IDAT more clearly. Below is the revised abstract that we will use for the final version:
>
> > Predicting scalar outcomes from functional data is challenging when measurements are sparse, irregular, and noisy, as in many scientific and clinical longitudinal studies. We propose IDAT, a dual-attention Transformer that operates directly on masked sampling schedules and avoiding ad-hoc imputation. IDAT couples (i) time-point attention, which captures local and long-range temporal dynamics together with the response relationship nonparametrically, with (ii) inter-sample attention, which adaptively shares information across subjects with similar trajectories to stabilize estimation under sparsity. These pathways complement one another: time-point attention captures subject-specific dynamics, whereas inter-sample attention leverages population structure to ``borrow strength'', echoing principles from random-effects model in longitudinal analysis. Under a random-effects framework that accounts for irregular sampling and measurement noise, we prove prediction-error bounds and consistency. Across both simulations and real-world applications, IDAT achieves the best overall performance among 19 baselines. Only in the extremely dense case (>80\% observations) does TabPFN (a recent Nature method) achieve a slight advantage, while IDAT still significantly outperforms all other baselines in this scenario. The learned attention weights are interpretable, revealing predictive time domains and potential clusters. Overall, IDAT demonstrates that an end-to-end sparsity-aware Transformer improves both accuracy and interpretability for scalar-on-function prediction.
>
>
> **Response to Weakness 4:**
>
> We will strengthen the theory section by explicitly discussing the implications of our results. In the revision, we’ll highlight how the discretization error bound connects the irregular functional data to time grids, explain how the variance-reduction effect of inter-sample attention translates into more stable predictions, elaborate on how the consistency results for IDAT establish identifiability of the underlying functional relationship, and add the key insights we gain from these theoretical properties to make their practical relevance clear.

---

> ### Author Response · Authors · 2025-11-23
> **Response to Reviewer jqgT**
>
> **Response to Question 1:**
>
> The inter-sample attention module is motivated by the principle of "borrowing strength", a well-established concept in longitudinal data analysis (e.g., random effects models and hierarchical modeling). In the i.i.d. case, the learned embeddings are similar across subjects but theoretically the inter-sample attention reduces prediction variance rather than increasing it. The prediction for subject $\hat{Y} = \frac{1}{|\mathcal{B}|}\sum_j f(\mathbf{X}_j)$, where $\mathcal{B}$ is the batch. **The variance of this averaged prediction is $\text{Var}(\hat{Y}) = \frac{\sigma^2}{|\mathcal{B}|}$, which is smaller than the variance $\sigma^2$ from using only subject $i$'s data, provided $|\mathcal{B}| > 1$ (See Lemma 9 for details).** This variance reduction translates to improved prediction accuracy. Similar variance reduction effects can be found in shrinkage estimation, a well-known statistical principle (James-Stein effect, empirical Bayes) for longitudinal data analysis.
>
> **Response to Question 2:**
>
> No, not all transformers are interpretable. IDAT incorporates explicit temporal information, enabling interpretable domain selection. For general transformer, since the input positional encoding is only capturing the ordering relationship the tokens, not with clear temporal information, we could not have such interprability on the domain selection. IDAT's interpretability is specific to address challenges in functional data analysis. **Time-point attention identifies which time points are predictive and inter-sample attention reveals which subjects inform each target subject's prediction.**
> 1. The time-point attention weights within each subject reveal which time points in that subject's trajectory are most informative for the prediction (e.g., early disease progression markers). This temporal pattern detection is analogous to domain detection in functional data analysis, where methods identify informative time regions in longitudinal trajectories [1,2,3]. The detection ability of our temporal attention is demonstrated in Simulation Case II, and we further validate this interpretability on the NCDS data by showing that time regions with low attention weights are indeed less informative to the prediction (see Figure 8), confirming that the learned attention patterns correspond to scientifically meaningful temporal structure.
> 2. The inter-sample attention weights reveal which other subjects inform the prediction for each target subject.
> - Specifically, in i.i.d. settings (Simulation Case I), attention weights are similar across subjects. This indicating that no particular subjects are preferentially informative, so the model learns to average appropriately.
> - In clustered settings (Simulation III, real data), non-uniform attention weights reveal which subjects share similar functional patterns with the target subject, effectively identifying latent cluster memberships and population-level patterns. In Simulation Case III (Figure 6), we compare test set embeddings for IDAT models with and without inter-sample attention. When inter-sample attention is removed, the learned embeddings show collapsed cluster separation: subject profiles become much more similar and differ only slightly in early trajectory stages, demonstrating that inter-sample attention is essential for identifying and preserving the underlying cluster structure in functional representations.
>
> **Response to Question 3:**
>
> Marked point processes (MPPs) are a related area. Longitudinal observations can be viewed through two complementary lenses:
> 1. The functional data perspective (IDAT's approach) assumes an underlying continuous function $X_i(t)$ exists for all $t \in \mathcal{T}$, treats observations as noisy snapshots
> 2. The MPP perspective treats observation times $\{t_{i,j}\}$ as random events from a point process with marks $\{X_i(t_{i,j})\}$, models the joint distribution of event times and marks, and explicitly handles observation schedules through intensity functions.
>
> MPPs model the joint distribution of times, which differs from our approach. In the reference, the distribution of marks represents functional trajectory values, and log-normal mixture models are applied for inter-event times. While we could potentially extend IDAT by applying this marginally to model measurement times through conditional intensity functions (to improve positional embeddings), the Transformer-based mark distribution in this work does not incorporate response ($Y$) information, making it less directly applicable to our prediction task.
>
> [1] James, G. M., Wang, J., & Zhu, J. (2009). Functional linear regression that’s interpretable.
>
> [2] Ramsay, J. O., & Silverman, B. W. (2005). Functional data analysis. New York, NY: Springer New York.
>
> [3] Matsui, H., & Konishi, S. (2011). Variable selection for functional regression models via the L1 regularization. Computational Statistics & Data Analysis, 55(12), 3304-3310.

---

### Author Response · Authors · 2025-11-28
**Rebuttal Revision Update**

Thank you all for the thoughtful review and for guiding the revisions. We have updated the manuscript accordingly and would very much appreciate your advice on any remaining issues you’d like us to address. Please let us know if any additional clarifications or refinements would be helpful.

---

### Meta-Review · Area_Chair_M38H · 2025-12-27

**Summary:**

The paper proposes IDAT, a dual-attention Transformer for scalar-on-function regression, specifically designed for sparse and irregular longitudinal data. All reviewers generally agreed that the problem setting is important and the theoretical analysis regarding consistency and error bounds is sound.

However, multiple reviewers noted that the architecture essentially combines two established mechanisms (time-point attention and inter-sample pooling) without offering significant algorithmic innovation beyond this combination. Additionally, there were concerns about the placement of critical results in the appendix and the diminishing performance advantage of the method in denser data regimes compared to baselines like TabPFN. While the authors provided clarifications on their contributions, the core concern that the work represents a sensible application of existing tools rather than a fundamental advance for top ML conferences remains.

**Reviewer Concerns:**

Addressed Concerns:

- The authors successfully clarified that the missingness assumption is actually deterministic masking due to grid discretization, rather than random missing data requiring specific statistical assumptions.

- The authors addressed Reviewer ZLuK's request for more experimental details by providing hyperparameter sensitivity tables and clarifying the generation of informative vs. non-informative sparsity.

- The authors clarified that inter-sample attention is motivated by the statistical principle of "borrowing strength" (variance reduction) akin to random effects models, which addressed Reviewer jqgT's question about why this wouldn't increase variance.


Outstanding Concerns:

- This remains the primary outstanding concern. Reviewer jqgT stated they "cannot pinpoint a single non-trivial or unique approach", and Reviewer 63go maintained that the core idea is "combining two existing approaches, limiting the overall novelty". The rebuttal did not convince them that the application to longitudinal data constituted sufficient architectural novelty.

- Reviewer 63go pointed out that the method's advantage narrows significantly or disappears outside of very sparse regimes.

**Reviewer Scores:**

Reviewer jqgT and Reviewer ZLuK will likely keep their scores.

Reviewer 63go explicitly stated in his/her final comment: "I will raise my score to 4". He/she acknowledged the sound theoretical analysis; however, he/she reiterated that the novelty is limited and the paper might not be strong enough for ICLR.

---

### Decision · Program_Chairs · 2026-01-26

Reject